METHODS AND RESOURCES

# Community-based reconstruction and simulation of a full-scale model of the rat hippocampus CA1 region

**Armando Romani** [1]*, **Alberto Antonietti** [1], **Davide Bella** [1], **Julian Budd** [1,2], **Elisabetta Giacalone** [3], **Kerem Kurban** [1], **Sára Sáray** [2,4], **Marwan Abdellah** [1], **Alexis Arnaudon** [1], **Elvis Boci** [1], **Cristina Colangelo** [1], **Jean-Denis Courcol** [1], **Thomas Delemontex** [1], **András Ecker** [1], **Joanne Falck** [5], **Cyrille Favreau** [1], **Michael Gevaert** [1], **Juan B. Hernando** [1], **Joni Herttuainen** [1], **Genrich Ivaska** [1], **Lida Kanari** [1], **Anna-Kristin Kaufmann** [1], **James Gonzalo King** [1], **Pramod Kumbhar** [1], **Sigrun Lange** [5,6], **Huanxiang Lu** [1], **Carmen Alina Lupascu** [3], **Rosanna Migliore** [3], **Fabien Petitjean** [1], **Judit Planas** [1], **Pranav Rai** [1], **Srikanth Ramaswamy** [1,7], **Michael W. Reimann** [1], **Juan Luis Riquelme** [1], **Nadir Román Guerrero** [1], **Ying Shi** [1], **Vishal Sood** [1], **Mohameth François Sy** [1], **Werner Van Geit** [1], **Liesbeth Vanherpe** [1], **Tamás F. Freund** [2,4‡], **Audrey Mercer** [5‡], **Eilif Muller** [1,8,9,10‡], **Felix Schürmann** [1‡], **Alex M. Thomson** [5‡], **Michele Migliore** [3‡]*, **Szabolcs Káli** [2,4‡]*, **Henry Markram** [1‡]*

**1** Blue Brain Project, École polytechnique fédérale de Lausanne (EPFL), Campus Biotech, Geneva, Switzerland, **2** HUN-REN Institute of Experimental Medicine (KOKI), Budapest, Hungary, **3** Institute of Biophysics, National Research Council (CNR), Palermo, Italy, **4** Faculty of Information Technology and Bionics, Pázmány Péter Catholic University, Budapest, Hungary, **5** UCL School of Pharmacy, University College London (UCL), London, United Kingdom, **6** School of Life Sciences, University of Westminster, London, United Kingdom, **7** Neural Circuits Laboratory, Biosciences Institute, Newcastle University, Newcastle upon Tyne, United Kingdom, **8** Department of Neurosciences, Faculty of Medicine, Université de Montréal, Montréal, Canada, **9** Centre Hospitalier Universitaire (CHU) Sainte-Justine Research Center, Montréal, Canada, **10** Mila Quebec AI Institute, Montréal, Canada

‡ These senior authors also contributed equally to this work.
* romani.armando@gmail.com (AR); michele.migliore@cnr.it (MM); kali@koki.hu (SK); henry.markram@epfl.ch (HM)

**Data Availability Statement:** All relevant data, tools, software, and models are available on the Hippocampus Hub (https://www.hippocampushub.eu). The full network model, along with the code

## Abstract

The CA1 region of the hippocampus is one of the most studied regions of the rodent brain, thought to play an important role in cognitive functions such as memory and spatial navigation. Despite a wealth of experimental data on its structure and function, it has been challenging to integrate information obtained from diverse experimental approaches. To address this challenge, we present a community-based, full-scale in silico model of the rat CA1 that integrates a broad range of experimental data, from synapse to network, including the reconstruction of its principal afferents, the Schaffer collaterals, and a model of the effects that acetylcholine has on the system. We tested and validated each model component and the final network model, and made input data, assumptions, and strategies explicit and transparent. The unique flexibility of the model allows scientists to potentially address a range of scientific questions. In this article, we describe the methods used to set up simulations to reproduce in vitro and in vivo experiments. Among several applications in the article, we focus on theta rhythm, a prominent hippocampal oscillation associated with various behavioral correlates and use our computer model to reproduce experimental findings.

and data needed to reproduce all the figures, are also accessible on the Harvard Dataverse. Model: https://dataverse.harvard.edu/dataset.xhtml?persistentId=doi:10.7910/DVN/TN3DUI Figures: https://dataverse.harvard.edu/dataset.xhtml?persistentId=doi:10.7910/DVN/UGOQWE.

**Funding:** This study was supported by allocated funding to the Blue Brain Project (www.epfl.ch/research/domains/bluebrain/), a research center of the École Polytechnique Fédérale de Lausanne (EPFL) (www.epfl.ch/), from the Swiss government's Board of the Swiss Federal Institutes of Technology (ethrat.ch/) to A.A., A.Ar., A.E., A-K. K., A.R., C.C., C.F., D.B., E.B., E.M., F.P., F.S., G.I., H.L., H.M., J.B., J.B.H., J-D.C., J.G.K., J.H., J.L.R., J.P., K.K., L.K., L.V., M.A., M.F.S., M.G., M.W.R., N.R.G., P.K., P.R., S.R., T.D., Y.S., V.S., W.V.G. Funding was also provided by The Human Brain Project (www.humanbrainproject.eu/) through the European Union Seventh Framework (FP7/2007-2013) under grant agreement no. 604102 (HBP) and from the Horizon 2020 Framework Programme for Research and Innovation (https://www.humanbrainproject.eu/en/about-hbp/project-structure/human-brain-project-ec-grants/) under the Specific Grant Agreements No. 720270 (Human Brain Project SGA1) and No. 785907 (Human Brain Project SGA2) to to H.M., S.K., M.M., A.M., and A.M.T.. M.M. also acknowledge funding for this work from the European Union, NextGenerationEU (Project IR0000011, CUP B51E22000150006, "EBRAINS-Italy"), the EU Horizon Europe Program under the specific Grant Agreement 101147319, EBRAINS 2.0 project, the Fenix computing and storage resources under the Specific Grant Agreement No. 800858 (Human Brain Project ICEI, https://fenix-ri.eu/about-fenix), and a grant from the Centro Svizzero di Calcolo Scientifico (CSCS) under project ID ich002 and ich011 (https://www.cscs.ch/publications/news/2019/fenix-consortium-partner-eth-zurich/cscs-e-infrastructure-enabling-nine-projects-from-human-brain-project). S.K. and S.S. were supported by the European Union Instrument for Recovery and Resilience project RRF-2.3.1-21-2022-00004 within the framework of the Hungarian Artificial Intelligence National Laboratory (https://mi.nemzetilabor.hu). The Wellcome Trust (https://wellcome.org/), Medical Research Council (https://www.ukri.org/councils/mrc/), Novartis Pharma (https://www.novartis.com/research-and-development/research-collaborations) and the Human Brain Project funded A.M. and A.M.T.. The funders had no role in study design, data collection and analysis, decision to publish, or preparation of the manuscript.

Finally, we make data, code, and model available through the hippocampushub.eu portal, which also provides an extensive set of analyses of the model and a user-friendly interface to facilitate adoption and usage. This community-based model represents a valuable tool for integrating diverse experimental data and provides a foundation for further research into the complex workings of the hippocampal CA1 region.

## Introduction

The hippocampus is thought to play a fundamental role in cognitive functions such as learning, memory, and spatial navigation [1,2]. It consists of three subfields of *cornu ammonis* (CA), CA1, CA2, and CA3 (see [3]). CA1, for instance, one of the most studied, provides the major hippocampal output to the neocortex and many other brain regions (e.g., [4]). Therefore, understanding the function of CA1 represents a significant step towards explaining the role of hippocampus in cognition.

Each year, the large neuroscientific community studying hippocampus contributes thousands of papers to an existing mass of empirical data collected over many decades of research (see S1 Fig). Recent reviews have, however, highlighted gaps and inconsistencies in the existing literature [5–8]. Currently, the community lacks a unifying, multiscale model of hippocampal structure and function with which to integrate new and existing data.

Computational models and simulations have emerged as crucial tools in neuroscience for consolidating diverse multiscale data into unified, consistent, and quantitative frameworks that can be used to validate and predict dynamic behavior [9]. However, constructing such models requires assigning values to model parameters, which often involves resolving conflicts in the data, filling gaps in knowledge, and making explicit assumptions to compensate for any incomplete data. In order to validate the model, it must be tested under specific experimental conditions using independent sources of empirical evidence before the model can be used to generate experimentally testable predictions. Therefore, the curation of a vast range of experimental data is a fundamental step in constructing and parametrizing any data-driven model of hippocampus.

The challenge of incorporating these data into a comprehensive reference model of hippocampus, however, is considerable and calls for a community effort. While community-wide projects are common in other disciplines (e.g., Human Genome Project in bioinformatics, CERN in particle physics, NASA's Great Observatories program in astronomy—[10–12]), they are a relatively recent development in neuroscience. OpenWorm, for example, is a successful, decade-long community project to create and simulate a realistic, data-driven reference model of the roundworm *Caenorhabditis elegans* (*C. elegans*) including its neural circuitry of approximately 302 neurons to study the behavior of this relatively simple organism in silico [13,14]. By contrast, for the hippocampus, with a circuit many orders of magnitude larger than *C. elegans*, models have typically been constructed with a minimal circuit structure on a relatively small scale and often their model parameters have been tuned with the goal of reproducing a single empirical phenomenon (see [15]). Comparing the results from a variety of circuit models is problematic because they vary in their degree of realism and frequently rely on one or a few single neuron models making generalization of their findings difficult (see [15]). While these focused models have led to valuable insights (see [16]), this piecemeal approach fails to demonstrate whether these separate phenomena can be reproduced in a full circuit model without the need to adjust parameters.

**Competing interests:** The authors have declared that no competing interests exist.

**Abbreviations:** ACh, acetylcholine; aCSF, artificial cerebrospinal fluid; ADF, augmented Dickey–Fuller; AIS, axon initial segment; bAC, bursting accommodating; BBP, Blue Brain Project; BC, basket cell; BPAP, back-propagating action potential; CA, cornu ammonis; cAC, classical accommodating; cNAC, classical non-accommodating; CSD, current source density; CV, coefficient of variation; DG, dentate gyrus; eFEL, Electrophys Feature Extraction Library; LFP, local field potential; MOOC, massive online open course; MS, medial septum; MVS, median over visible spread; OVS, overall visible spread; PC, pyramidal cell; PCA, principal component analysis; PP, perforant pathway; PSC, postsynaptic current; PSD, power spectral density; PSP, postsynaptic potential; SC, Schaffer collaterals; STD, standard deviation; STP, short-term plasticity; STTC, spike time tiling coefficient; TMD, topological morphological descriptor.

Large-scale circuit models of hippocampus using realistic multi-compartment spiking neuronal models pioneered by Traub and colleagues [17–20] have been used to explain key characteristics of oscillatory activity observed in hippocampal slices and to examine the origins of epilepsy in region CA3. More recently, with significant increases in high-performance computing resources, [21] in a microcircuit model of CA1 and notably [22] in a full-scale CA1 model, have examined the contribution of diverse types of interneurons to the generation of prominent theta (4 to 12 Hz) oscillations. While these large-scale circuit models provide a more holistic approach, they still need to incorporate other features to improve their realism. For example, to better reflect the highly curved shape of the hippocampus, an atlas-based structure that more closely mimics anatomy is required. Additionally, models need to employ pathway-specific short-term synaptic plasticity known to regulate circuit dynamics and neural coding [23]. While [24] have constructed a down-scaled, atlas-based model of the rat dentate gyrus (DG) to CA3 pathway, there has to date been no atlas-based, full-scale model of rat CA1 (for a more detailed comparison of these models, see S2 Table).

To initiate a community effort of this magnitude requires an approach that standardizes data curation and integration of diverse data sets from different labs and uses these curated data to construct and simulate a scalable and reproducible circuit automatically (for recent discussion on the benefits of community-based standards and workflow, see [25,26]). A reconstruction and simulation methodology was introduced and applied at the microcircuit scale, for the neocortex [27] and the thalamus [28] and at full-scale for a whole neocortical area [29,30]. However, these models relied primarily on data sets collected specifically for the purpose rather than data sought from and curated with the help of the scientific community.

In this paper, we describe a community-based reconstruction and simulation of a full-scale, atlas-based multiscale structural and functional model of the area CA1 of the hippocampus that extends and improves upon the approach described in [27]. Here, a community of 5 different labs (BBP, CNR, KOKI, LNMC, and UCL), with different expertise and having different roles, collaborated to create a "first draft" reference CA1 circuit model that could be shared with the wider hippocampus community. Specifically, to model stimuli originating from beyond the intrinsic circuitry, we included the synaptic input from the Schaffer collaterals (SC) from CA3, which is the largest afferent pathway to CA1 and the most commonly stimulated in experiments. Furthermore, we also added the neuromodulatory influence of cholinergic inputs, perhaps the most studied neuromodulator in the hippocampus [31]. We constrained all model parameters and data using available experimental data from different labs or explicit assumptions made when data were lacking. We extensively tested and validated each model component and the final network to assess its quality. To maximize realism of the simulations, we set up simulation experiments to represent as closely as possible the experimental conditions of each empirical validation. We demonstrated the broad applicability of the model by studying the generation of neuronal oscillations, with a specific focus on theta rhythm, in response to a variety of different stimulus conditions. Over time and with the help of the community, limitations of the model revealed by these processes can be addressed to improve upon it. To facilitate a widespread adoption by the community, we have developed a web-based resource to share the model and its components, open sourcing extensive analyses, validations, and predictions that can be accessed as a complement to direct interaction with the model (hippocampushub.eu). Finally, we have developed a massive online open course (MOOC) to introduce users to the building, analysis, and simulation of a rat CA1 microcircuit (https://www.edx.org/course/simulating-a-hippocampus-microcircuit) providing a smaller version of the full-scale model for education purposes.

## Results

We divide the Results section into 2 parts: how the model was constructed and validated (Model reconstruction) and the simulation and analysis of activity in the reconstructed circuit model (Model simulations). For a list of abbreviations and acronyms used in the paper, see S1 Table.

### Model reconstruction

In this section, we describe how we reconstructed the main components of the model: the cornu Ammonis 1 (CA1), the Schaffer collaterals (SC), and the effect of acetylcholine (ACh) on CA1. Each of these main components is itself a compound model of several circuit "building blocks" (Fig 1 and S4 Fig). For each of these subcomponents, we show how, from the sparse data available in the literature (see S3–S24 Tables) and a list of assumptions (section List of assumptions), we arrived at the dense data necessary to ascribe a value for each model parameter. For each "building block" subcomponent, we validate it against available experiment data.

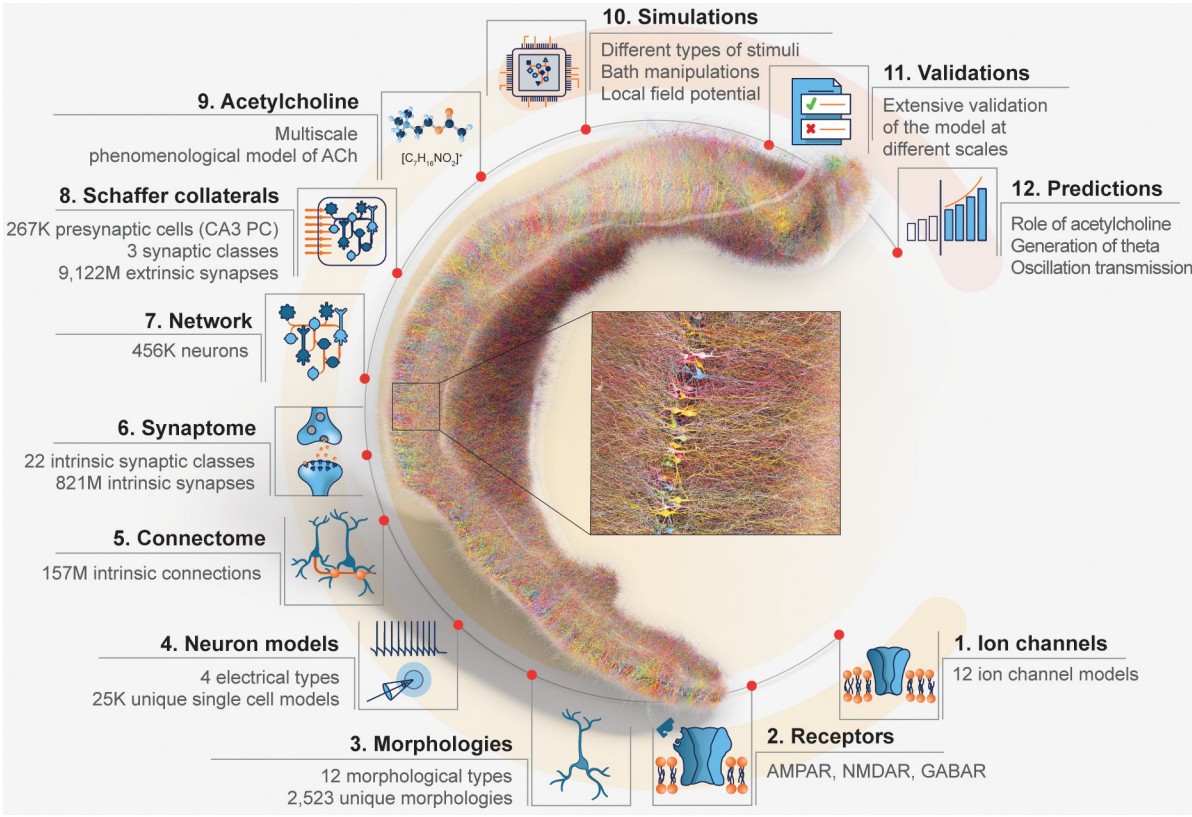

**Fig 1. Overview of the model.** A visualization of a full-scale, right-hemisphere reconstruction of rat CA1 region and its components. The number of cells is reduced to 1% for clarity, and neurons are randomly colored. The CA1 network model integrates entities of different spatial and temporal scales. The different scales also reflect our bottom-up approach to reconstruct the model. Ion channels (1) were inserted into the different morphological types (3) to reproduce electrophysiological characteristics and obtain neuron models (4). Neurons were then connected by synapses to generate an intrinsic CA1 connectome (5). For each intrinsic pathway, synaptic receptors (2) and transmission dynamics were assigned based on single neuron paired recording data (6) to create a functional intrinsic CA1 network model (7). The intrinsic CA1 circuit received synaptic input from CA3 via Schaffer collateral (SC) axons (8). The neuromodulatory influence of cholinergic release on the response of CA1 neurons and synapses was modeled phenomenologically (9). The dynamic response of the CA1 network was simulated with a variety of manipulations to model in vitro and in vivo, intrinsic and extrinsic stimulus protocols while recording intracellularly and extracellularly (10) to validate the circuit at different spatial scales against specific experimental studies (11) and to make experimentally testable predictions (12).

## Building CA1

To reconstruct a full-scale model of rat CA1, we created biophysical models of its neurons, defined an atlas volume of the region for one hemisphere, placed these neurons in the volume, and connected them together by following and adapting the method described in [27] (S4 Fig).

## CA1 neurons

To achieve a full-scale version of CA1, we needed to populate the model with approximately 456,000 cells. We started by curating 43 morphological reconstructions of neurons belonging to 12 morphological types: pyramidal cell (PC), axo-axonic cell (AA), 2 subtypes of bistratified cell (BS), back-projecting cell (BP), cholecystokinin (CCK) positive basket cell (CCKBC), ivy cell (Ivy), oriens lacunosum-moleculare cell (OLM), perforant pathway associated cell (PPA), parvalbumin positive basket cell (PVBC), Schaffer collateral associated cell (SCA), and trilaminar cell (Tri). To increase the morphological variability, we scaled and cloned them producing an initial morphology library of 2,592 reconstructions.

To validate the resulting morphology library, we compared them morphometrically and topologically to the original morphologies. The similarity scores for the distribution of morphological features were statistically similar (S5 Fig, all values $R > 0.98$, $p < 10^{-25}$). Using the topological morphology descriptor (TMD) [32], the persistence diagrams (S6 Fig) show an increase in morphological variability introduced by the cloning process (details per m-type in S7 and S8 Figs).

To produce electrical models (e-models), we began by taking 154 single-cell recordings and classifying traces into 4 electrical types (e-types) using Petilla nomenclature [33]: classical accommodating for pyramidal cells and interneurons (cACpyr, cAC), bursting accommodating (bAC), and classical non-accommodating (cNAC). From each trace, we extracted electrical features (e-features) which we used in combination with the curated morphologies to produce and validate 36 single-cell e-models [34,35]. In the case of pyramidal cells e-models, they qualitatively reproduced experimental findings in terms of back-propagating action potential (BPAP) ([36], $R = 0.878$, $p = 0.121$) and postsynaptic potential (PSP) attenuation ([37], $R = 0.846$, $p = 0.001$, S9 Fig). However, since we constrained the models with somatic not dendritic features, we expected some degree of difference compared with experimental results (see [38]).

To match the proportions of the morpho-electrical type (me-type) composition of the CA1 (S4 Table), we combined the 36 e-models with 2,592 curated morphologies to obtain an initial library of 26,112 unique me-type models. This is the pool of biophysical cell models available to populate the full-scale version of CA1.

## Defining the spatial framework

To represent the CA1 spatial volume, we started with a publicly available atlas reconstruction of the hippocampus [39]. Our aim was to create a continuous coordinate system to represent the 3 axes of the hippocampus (longitudinal, transverse, and radial) and a precise vector field for cell placement and orientation (Fig 2). For our building and analysis algorithms to work effectively, we applied a series of post-processing steps (S10 Fig, see section Atlas). Within this process, we redefined the layers parametrically to be consistent with the layer thicknesses in our data sets (section Layers).

## Placing neurons in the volume

After defining the volume, we wanted the model to match the neuronal density and proportion of cell types in rat CA1. We compiled available data to derive the cellular composition (section

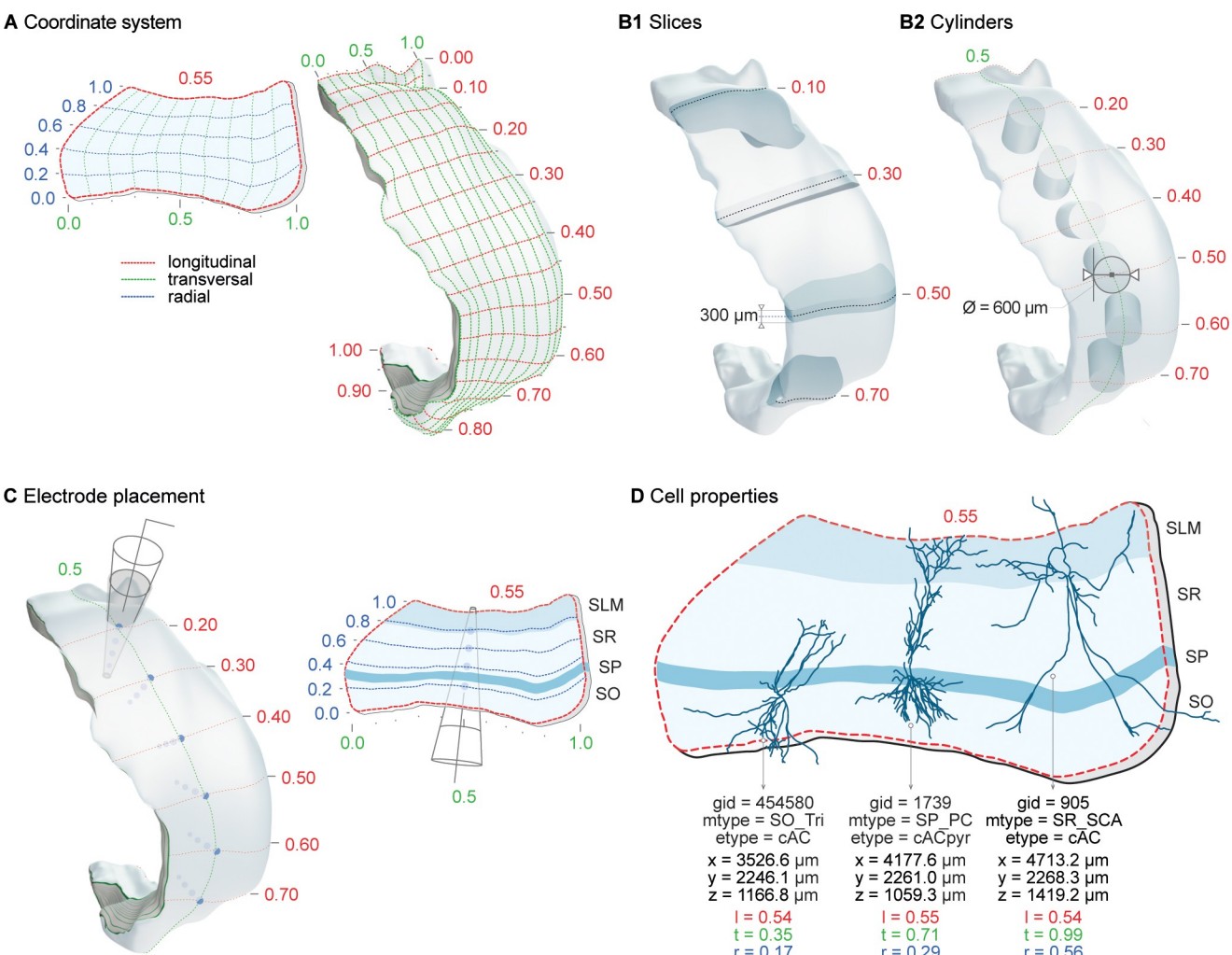

**Fig 2. Coordinate system.** Custom parametric coordinates system used as spatial reference for circuit building, circuit segmentation, and for simulation experiments. (A) Longitudinal (l, red), transverse (t, green) and radial (r, blue) axes of the CA1 volume are defined parametrically in range [0,1]. Left: Slice from volume shows radial depth from SO/alveus (r = 0) to SLM/pial (r = 1) and transverse extent from CA3/proximal CA1 (t = 1) to distal CA1/subiculum (t = 0) boundaries. Right: Full volume shows surface grid of transverse vs. longitudinal axes. Longitudinal axis extends from dorsal (l = 0) to ventral (l = 1) CA1. (B) Circuit segmentation for analysis and simulation. Coordinates system used to select CA1 slices of a given thickness (B1) or a cylinders of a given diameter (B2) at specific locations along longitudinal axis. (C) Extracellular electrode placed at a given surface position (left) and channels at selected laminar depth (right) in CA1 volume. (D) Each neuron in the circuit is defined by a unique general identifier (gid), its morphological type (m-type), electrical type (e-type), spatial xyz-coordinates and parameterized ltr-coordinates.

Cell composition, see S3 Table) and used it to populate the atlas with soma locations (S11A and S11B Fig). To check the consistency, we validated the resulting cell composition ($R = 0.999$, $p = 1.31\times10^{-28}$) and cell density ($R = 0.999$, $p = 0.0001$) (S12B and S12C Fig and S5 Table). At each soma location, we needed to select from the e-model library the neuron that best fits in the space of the layers. To this purpose, we oriented its morphology according to the vector fields (S11C and S11D Fig) and evaluated it against a set of rules that describe the target distribution of neurites per layer (S6 Table). Visually, cells in our model follow the curvature of the hippocampus and the different parts of the cells target the expected layers (S12A Fig). Subject to the multiple constraints of the cell placement algorithm, we placed 456,380 neurons in the volume, utilizing 2,523 unique morphologies, and 25,355 unique neuron models.

## Connecting CA1 neurons

To connect the placed neurons, we used the connectome algorithm previously described in [40]. In brief, the algorithm searches for co-localization of axon and postsynaptic neurons within a certain distance to identify a potential synapse (or apposition). After identifying all potential synapses, a subsequent pathway-specific pruning step discards some to match the known bouton densities (S7 Table) and number of synapses per single axon connection (S9 Table). This algorithm has been demonstrated to accurately recreate local connectivity [27–29] as well as higher-order topological features [41]. The resulting intrinsic connectome consisted of about 821 million synapses.

Given the importance of the connectome, we wanted to validate it as widely as possible to mitigate the uncertainty in our assumptions and literature data (S14–S17 Figs). First, we verified the bouton density and number of synapses per connection used in the pruning step was preserved in the generated connectome (bouton density: $R = 0.909$, $p = 0.0120$; number of synapses per connection: $R = 0.992$, $p = 2.41{\times}10^{-9}$, S14 Fig and S7 and S9 Tables). Next, we observed that the shape of the distributions for connection probability (S15A Fig), convergence (S16A Fig) and divergence (S17A Fig) were positively skewed as reported experimentally [42]. In the case of mean connection probability, experimental data did not allow a direct comparison because the distance between the neuron pairs tested was typically missing (S15C Fig and S8 Table). For convergence, we found that the subcellular distribution of synapses on different compartments of pyramidal cells in our model was consistent with [43] ($R = 0.988$, $p = 0.012$, S16D Fig). For divergence, the model did not always closely match the experimental data for the total number of synapses per axon formed by certain m-types ($R = 0.524$, $p = 0.286$, S17C Fig and S10 Table). We compared divergence also in terms of the percentage of synapses formed with PCs or INTs (S17D Fig and S11 Table) and validated the distribution of efferent synapses in the different layers (SO: $R = 0.798$, $p = 0.057$; SP: $R = 0.905$, $p = 0.013$; SR: $R = 0813$, $p = 0.049$; SLM: $R = 0.999$, $p = 4.11{\times}10^{-8}$, S17E Fig and S12 Table). Overall, this suggests the model connectome provides a reasonable approximation based on available data, while the discrepancies can be due, for example, to the small sample size and high variability in axon length recovered from in vitro slices.

To provide functional dynamics for synaptic connections with stochastic neurotransmitter release and short-term plasticity (STP) (see S18 Fig), we used the optimized parameters we previously derived in [34] for the 22 intrinsic synaptic classes of pathways we have identified (S13 and S14 Tables). The model was able to reproduce the PSP amplitudes (S18D Fig, R = 0.999, $p = 1.65{\times}10^{-19}$) and postsynaptic current (PSC) coefficient of variation (CV) of the first peak (S18E Fig, $R = 0.840$, $p = 0.018$) for the pathways with available electrophysiological recordings. After having constrained and validated the synapses, the reconstruction process of the intrinsic CA1 circuit is complete.

## Reconstruction of Schaffer collaterals (SC)

An isolated CA1 does not have substantial background activity [44], while normally the network is driven by external inputs. The Schaffer collaterals from CA3 pyramidal cells are the most prominent afferent input to the CA1 and the most studied pathway in the hippocampus [45,46]. Their inclusion allows us to deliver synaptic activity input patterns to the CA1.

To reconstruct the anatomy of Schaffer collaterals, we constrained the number of CA3 fibers and their average convergence on CA1 neurons with literature data (S15 and S16 Tables). Due to scarce topographical information, we distributed synapses uniformly along the transverse and longitudinal axes, while along the radial axis we followed a layer-wise distribution as reported by [5]. The resulting Schaffer collaterals added more than 9 billion synapses to

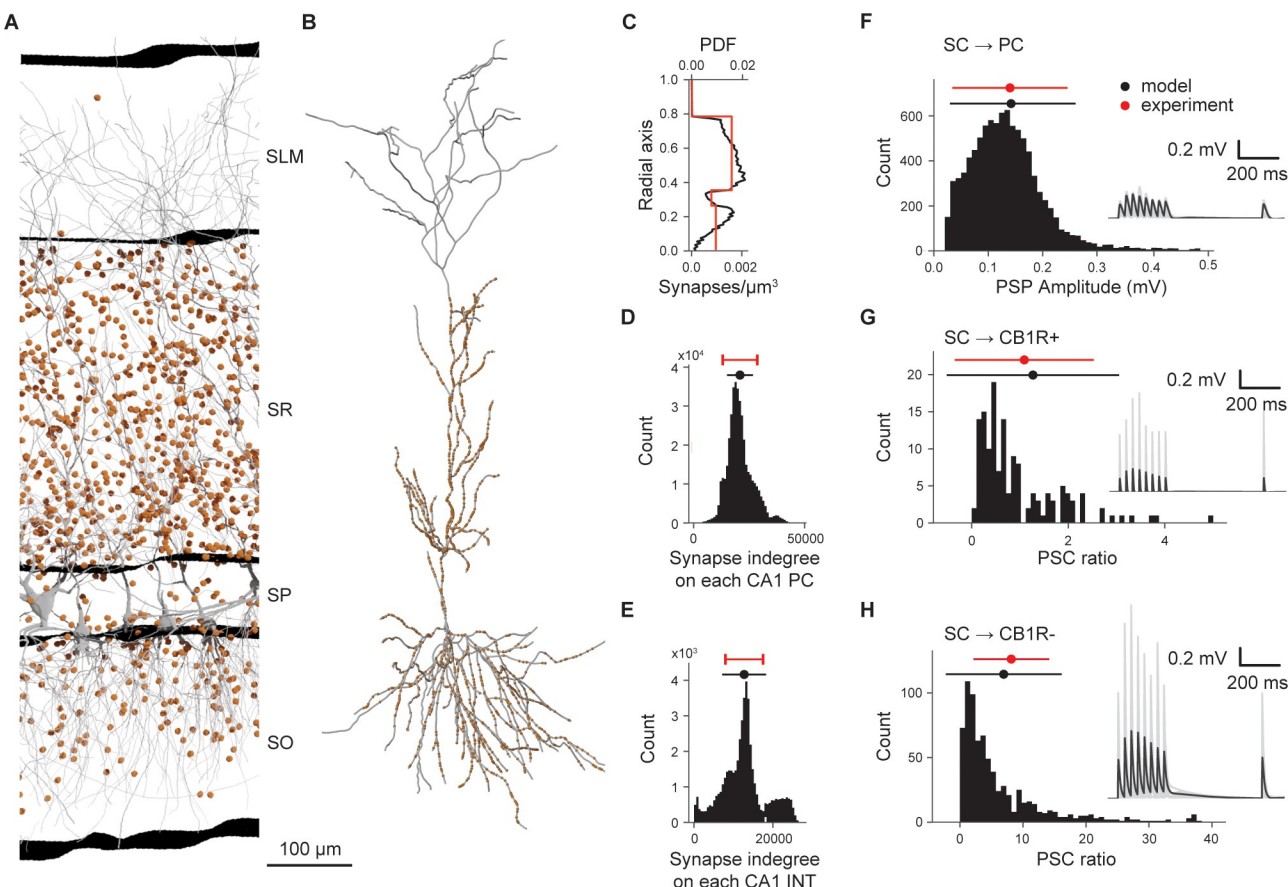

**Fig 3. Schaffer collaterals anatomy and physiology.** (A) Section of a slice of the dorsal CA1 showing neurons in gray and SC synapses in orange (10% of the existing ones). (B) Example of SC synapse placement (orange dots) on one reconstructed PC (in gray). (C–H) Validation of the anatomy (C–E) and physiology (F–H) of the SC. Experimental values can be found in S15–S18 Tables. Density of SC synapses (lower x axis) and PDF (upper x axis) at different depths (radial axis percentage) (C). Distributions of afferent synapses from SC to PC (D) and INT (E). Distribution of PSP amplitudes for SC → PC synapses (F). Distribution of PSC ratio (see text) for SC → CB1R+ (G) and SC → CB1R- (H). Insets in panels F–H report voltage membrane traces of 10 randomly selected pairs of SC → PC, SC → CB1R+, and SC → CB1R- interneurons, respectively. The presynaptic SC is stimulated to fire 8 times at 30 Hz, plus a recovery pulse after 500 ms from the last spike of the train. Solid black lines represent mean values and shaded gray areas the standard deviation. PC, pyramidal cell; PDF, probability density function; PSP, postsynaptic potential; SC, Schaffer collaterals.

CA1, representing 92% of total modeled synapses. As expected, the synapse laminar distribution (Fig 3A–3C) and mean convergence on PCs and interneurons (Fig 3D and 3E) match experimental values (one-sample $t$ test, $p = 0.957$ for PCs and $p = 0.990$ for INTs). Interestingly, other unconstrained properties also match experimental data. The convergence variability is comparable with the upper and lower limits identified by [5] (model PC: 20,878 ± 5,867 synapses and experimental PC: 13,059–28,697, model INT: 12,714 ± 5,541 and experimental INT 7,952–17,476, Fig 3D and 3E). In addition, the axonal divergence from a single CA3 PC is 34,135 ± 185 synapses (S19A Fig), close to the higher end of the ranges measured by [47–49] (15,295–27,440, S19B Fig). Finally, most of the connections formed a single synapse per neuron (1.0 ± 0.2 synapses/connection, S19A Fig), consistent with what has been previously reported [5].

The CA3 afferent pathway is sparsely connected to the CA1, so the chance of obtaining paired CA3-CA1 neuronal recordings is small between PCs and much smaller from PC to interneurons [50–53]. So, to constrain SC physiology we did not have enough data to follow the parametrization used for intrinsic synapses [34]. Instead, we used the available data

(S17 and S18 Tables) and optimized the missing parameters as shown in S19C Fig. The resulting SC→PC synapses match the distribution of EPSP amplitudes as measured by [50] (Fig 3F, experiment: 0.14 ± 0.11 mV, CV = 0.76, model: 0.15 ± 0.12 mV, CV = 0.80, z-test $p$ = 0.709), giving a peak synaptic conductance of 0.85 ± 0.05 nS and $N_{RRP}$ of 12. The rise and decay time constants of AMPA receptors (respectively 0.4 ms and 12.0 ± 0.5 ms) were obtained by matching [50] EPSP dynamics (S19C Fig, 10% to 90% rise time model: 5.4 ± 0.9 ms and experiment: 3.9 ± 1.8 ms, half-width model: 20.3 ± 2.9 ms and experiment: 19.5 ± 8.0 ms, decay time constant model: 19.5 ± 2.5 ms and experiment: 22.6 ± 11.0 ms). In the case of SC→INT synapses, we distinguished between cannabinoid receptor type 1 negative (CB1R-) and positive (CB1R +) interneurons [54]. SC→CB1R- synapses match $EPSC_{CBR1−}/EPSC_{PC}$ experimental ratio [54] (model 6.95 ± 9.20 and experiment 8.15 ± 6.00, z-test $p$ = 0.18, Fig 3G), resulting in a peak conductance of 15.0 ± 1.0 nS and $N_{RRP}$ of 2. SC→CB1R+ synapses match the $EPSC_{CBR+}/EPSC_{PC}$ experimental ratio (model 1.27 ± 1.78 and experiment 1.09 ± 1.44, z-test $p$ = 0.06, Fig 3H), giving peak conductance of 1.5 ± 0.1 nS and $N_{RRP}$ of 8. SC→INT synapses (all) match the timing in the EPSP-IPSP sequence of [55] (model: 2.69 ± 1.18 ms, experiment: 1.9 ± 0.6 ms, S19 Fig E), yielding a rise and decay constants for AMPA receptors of 0.1 ms and 1.0 ± 0.1 ms, respectively. This short latency gives effective feedforward inhibition, which is a key aspect for the transmission of oscillations from CA3 to CA1 (see below).

We functionally validated SC projections reproducing [56], where the authors examined the basic input–output (I-O) characteristics of SC projections in vitro. The SC pathway is thought to be dominated by feedforward inhibition, which increases the dynamic range of the CA1 network and linearizes the I-O curve [56,57]. Blocking gamma-aminobutyric acid receptor (GABA$_A$R) drastically reduces the dynamic range of the network resulting in an I-O curve that saturates very quickly. To match the methodology of [56], we set up the simulations to be as close as possible to the experimental conditions (slice of 300 μm, $Ca^{2+}$ 2.4 mM, $Mg^{2+}$ 1.4 mM, 32˚C) (Fig 4A) and we used the same sampling strategy: randomly sampling 101 neurons in the slice to find how many SC axons were required to make all of them fire (representing respectively 100% of the input and 100% of output, Fig 4B). To assess the role of feedforward inhibition, we mimic the effect of gabazine by disabling the connections from interneurons. The model captured the quasi-linearization of the I-O response in control conditions (Pearson test on linearity R = 0.992, p = 2.56×10$^{-9}$) and the rapid saturation of the CA1 network with the simulated "no GABA" condition (Fig 4B). In control conditions, at 50% of input intensity (Fig 4C) the spiking activity of CA1 SP neurons is rather weak and rapidly suppressed by the feedforward inhibition, while without inhibition CA1 neurons fire for more than 50 ms at high frequency (up to 200 Hz). Taken together, these results suggest the SC projection represents a valid model given the available empirical data.

## Cholinergic modulation

The behavior of the hippocampus is shaped by several neuromodulators, with acetylcholine (ACh) among the most studied. Cholinergic fibers originate mainly from the medial septum (MS) and have been correlated with phenomena such as theta rhythm, plasticity, memory retrieval, and encoding, as well as pathological conditions such as Alzheimer's disease [58]. This section describes the reconstruction of a phenomenological model of ACh, quantifying the effects of ACh on neurons and synapses, and developing a novel method to integrate available experimental data (S19 and S20 Tables and Fig 5A and 5B). The data used to build the model was obtained from in vitro application of various cholinergic agonists such as ACh and carbachol (CCh); here, we assume that their effects are comparable [59]. We modeled the effect of ACh on neurons and synapses. The effect on neurons results in an increase in resting

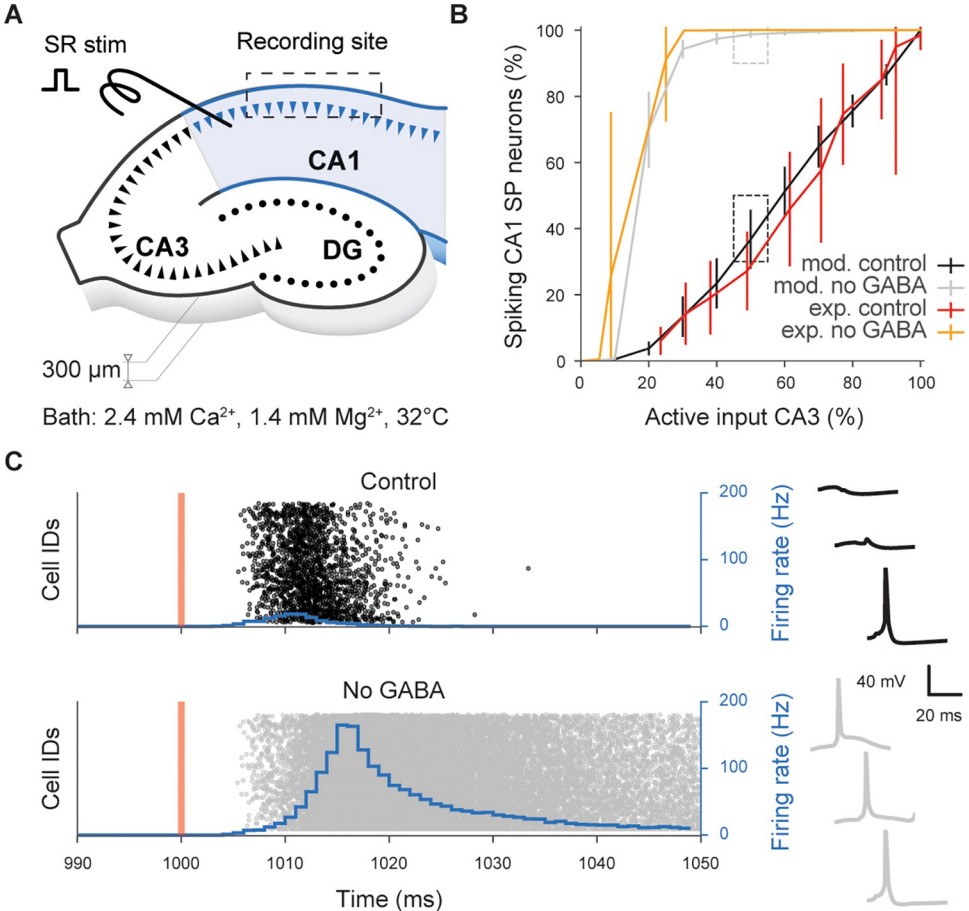

**Fig 4. Schaffer collaterals validation.** Effect of the feedforward inhibition on the input–output relationship of the network illustrated in a slice experiment. (A) The illustration (redrawn from Fig 1A of [56]) shows the in silico experimental setup. (B) Response of 101 selected neurons to an increasing number of stimulated SC fibers, with and without GABA. The dashed boxes identify the condition (50% of active SC) that is used to show the model's results in panel C. (C) Raster plots of SP neurons in response to the SC stimulation (orange vertical line) with the overlaying firing rate (blue). On the right, membrane voltage traces of three randomly selected SP neurons in control (black) and no GABA (gray) conditions. SC, Schaffer collaterals.

membrane potential or firing frequency. We were able to integrate both types of experimental data by estimating the net current that is required to evoke the corresponding changes for a given concentration of ACh (see Eq 9 in Methods). ACh affects synaptic transmission acting principally at the level of release probability [23,60,61] (see Eq 10 in Methods).

After modeling the effect of ACh on neuron excitability and synaptic transmission, we validated the effect of ACh at the network level against available data (see S21 Table). To accomplish this, we simulated bath application of ACh for a wide range of concentrations (from 0 µM (i.e., control condition) to 1,000 µM) (Fig 5E and 5F). We observed a subthreshold increase in the membrane potential of all neurons for values of ACh lower than 50 µM, without any significant change in spiking activity. At intermediate doses (i.e., 100 µM and 200 µM), the network shifted to a more sustained activity regime. Here, we observed a generalized increase in firing rate as ACh concentration increased and a progressive build-up of coherent oscillations whose frequencies ranged from 8 to 16 Hz (from high theta to low beta frequency bands). The correlation peak between CA1 neurons occurred at 200 µM ACh (Fig 5G and 5H). At very high concentrations (i.e., 500 µM and 1,000 µM), we observed a decrease in the

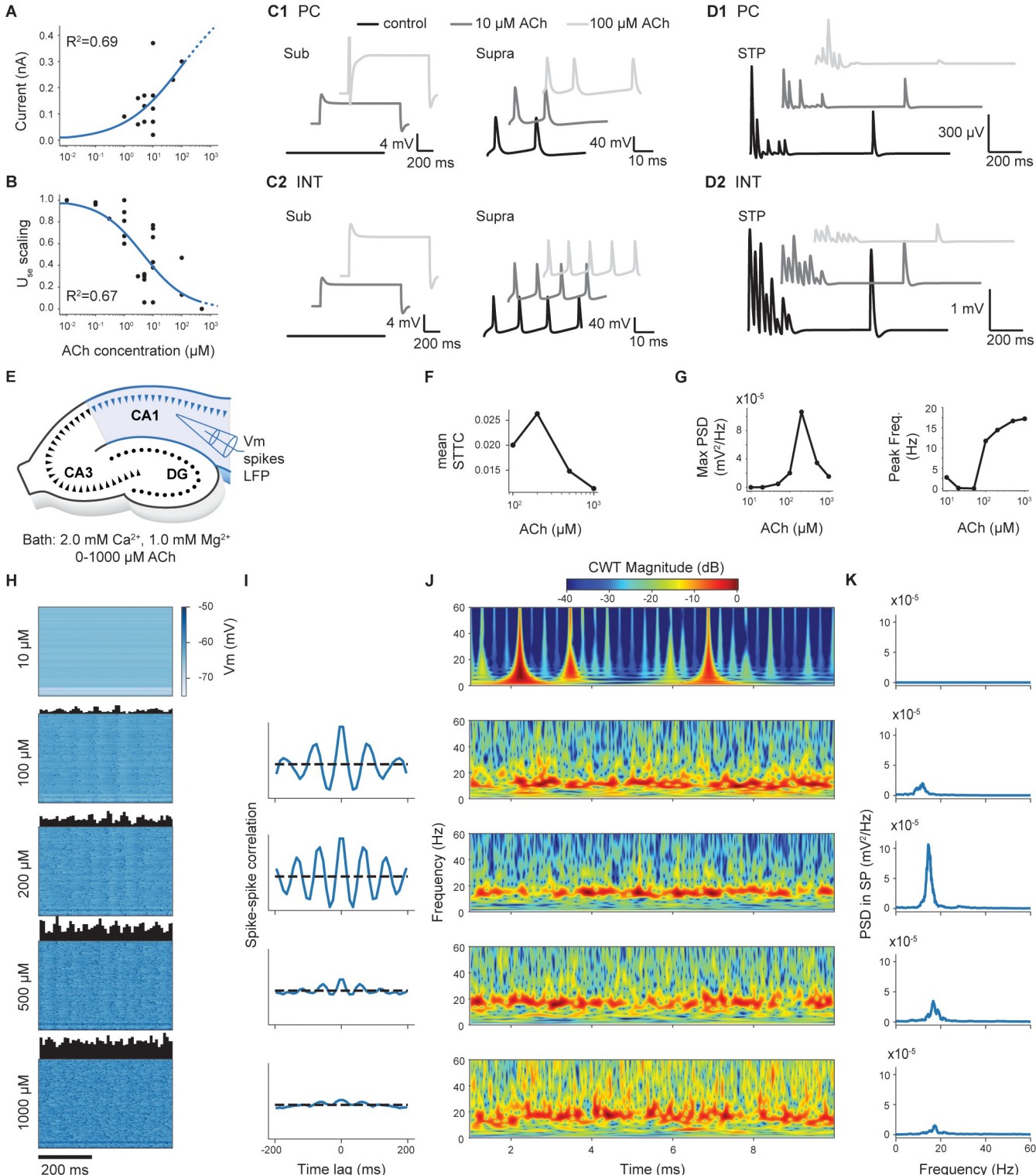

**Fig 5. Effects of Acetylcholine on neurons, synapses, and network.** (A) Dose-response modulation of neuronal excitability caused by ACh. Black dots are experimental data points; blue curve represents the fitted equation. The dashed part of the curve indicates regions outside available experimental data. (B) Dose-response modulation of synaptic release. Same color code as in A. (C) Example traces for PC (C1) and interneurons (C2) in sub-threshold and supra-threshold conditions, with different concentrations of ACh. (D) Example traces showing the STP dynamics for PC (D1) and interneurons (D2) at different concentrations of ACh. (E) The illustration shows the in silico experimental setup to analyze network effects of ACh. Different concentrations of CCh are applied to the circuit and multiple types of recordings made in the CA1 (membrane voltage, spike times, LFPs). (F) Mean STTC as a function of ACh

concentration. (G) Maximum of the LFP PSD and peak frequency as a function of ACh concentration. (H) The voltage of 100 randomly selected neurons at different levels of ACh. The upper histograms show the instantaneous firing rate. (I) Spike-spike correlation histograms. (J) Spectrogram of the LFP measured in SP at different ACh levels. (K) PSD of the LFP recorded in SP. ACh, acetylcholine; LFP, local field potential; PC, pyramidal cell; PSD, power spectral density; STTC, spike time tiling coefficient.

power of network oscillations, which was further confirmed by analysis of local field potential (LFP) (Fig 5I). Power spectral density (PSD) analysis showed a maximum absolute amplitude for 200 μM ACh with a peak frequency of approximately 15 Hz (Fig 5J and 5K). Higher concentrations decreased the maximum amplitude of the PSD while the peak frequency converged toward 17 Hz (Fig 5K). Thus, we observe the emergence of 3 different activity regimes at low, intermediate, and high levels of cholinergic stimulation. It is unclear whether the network behavior we observe is consistent with all experimental findings because of their different methodologies. Nevertheless, this phenomenological model of ACh allows the reproduction of other experiments in which ACh or its receptor agonists are necessary (see, for example, sections Medial septum input and Other frequencies).

## Model simulations

By following a data-driven approach and independently validating each model component, we have arrived at a candidate reference model. It can serve as a basis for investigating several scientific questions where parameters are only adjusted to reflect the different experimental setups, rather than tuned to achieve a specific network behavior. A simulation experiment is essentially a model of the experimental setup that is reproduced with as much accuracy as possible within the limits of the circuit model. As presented in the Model reconstruction section we can, for example, simulate slices of a certain thickness, change the extracellular concentration of ions, change temperature, and enable spontaneous synaptic events.

Because a central interest in hippocampus has been its oscillatory activity, here, we show several simulations with particular emphasis on theta oscillations, a prominent network phenomenon observed in the hippocampus in vivo and related to many behavioral correlates [62]. Then, we examine the frequency response band-pass properties of CA1 circuit for a wider range of SC input frequencies.

## Theta oscillations

During locomotion and REM sleep, CA1 generates a characteristic rhythmic theta-band (4 to 12 Hz) extracellular field potential [63–67]. Neurons in many other brain regions such as neocortex are phase-locked to these theta oscillations [68,69] suggesting hippocampal theta plays a crucial coordinating role in the encoding and retrieval of episodic memory during spatial navigation [62,70]. Yet, despite more than 80 years of research, the trigger that generates theta oscillations in CA1 remains unclear because of conflicting evidence. This represents an opportunity to use our reference model to investigate these inconsistencies and gain an improved understanding of theta generation. As a first step, we wanted to reproduce a series of experimental data and investigate potential mechanisms proposed in literature [71], namely, intrinsic CA1 generation and extrinsic pacemaker oscillations from CA3 or from MS.

## CA1 generation

To investigate possible intrinsic mechanisms of theta rhythm generation in CA1 [72,73], we examined 3 candidate sources of excitation that might induce oscillations: (1) spontaneous synaptic release or miniature postsynaptic potentials (minis or mPSPs); (2) homogeneous

random spiking of SC afferent inputs; and (3) varying bath concentrations of extracellular calcium and potassium to induce tonic circuit depolarization. While in their CA1 circuit model [22] reported random synaptic activity was sufficient to induce robust theta rhythms, in our model we found none of these candidates generated robust theta rhythms. For minis, we found setting release probabilities to match empirically reported mPSP rates (S22 Table) led to irregular, wide-band activity in CA1 (see S20 Fig). For random synaptic barrage, varying presynaptic rate to match the mean postsynaptic firing rate of pyramidal cells in [22] resulted in irregular beta-band, not regular theta-band oscillations (see S21 and S22 Figs). For tonic depolarization, within a restricted parameter range it was possible to generate theta oscillations around 10 to 12 Hz, but their intensity was variable and episodic (see S23 and S24 Figs). Therefore, we could not find any intrinsic mechanism capable of generating regular theta activity in our circuit model.

## CA3 input

To mimic the transmission of CA3 theta oscillations to CA1, we generated SC spike trains across a range of theta-modulated sinusoidal rate functions (signal frequency) at different mean individual rates (cell frequency) (Fig 6A). We delivered these stimuli at 3 different circuit scales (whole circuit, thick slice, and cylinder circuit; see Fig 2) and measured extracellular LFP and intracellular membrane potential. To mimic recordings performed under in vivo and in vitro conditions, we did this using different calcium levels: 1 mM for "in vivo-like" and 2 mM for "in vitro-like." For LFP, we found CA1 faithfully followed the theta-modulation input frequency at both in vivo-like and in vitro-like calcium levels and at the different circuit scales tested (e.g., 8 Hz, see Fig 6C–6F). However, for the same stimulus, the LFP at in vivo-like calcium levels was around 3 orders of magnitude less powerful than the one at in vitro-like calcium levels (Fig 6B, 6C and 6F) due to CA1 spiking rates being far lower (e.g., in full circuit, pyramidal cell mean firing rate of 0.00018±0.0067 (1 mM) versus 0.25±0.50 Hz (2 mM)). Therefore, due to this very low firing rate, we decided not to analyze results from in vivo-like conditions further and focused on those from the in vitro-like condition only (Fig 6C–6E).

We analyzed the properties of the model LFP and compared them to experimental data. First, we examined whether the circuit size is critical for theta generation. We found that all the 3 circuit scales generated theta oscillations, but slice and cylinder circuits had reduced magnitude of the modulation and LFP power (Fig 6C–6E). Second, across all the scales, the LFP waveform (Fig 6C–6E, left columns) was more asymmetrical with a fast rise and slower decay (mean asymmetry index = −1.34±0.23; see S25 Fig) with respect to what was reported during rat locomotion (asymmetry index = −0.27, [74]) or REM sleep periods (asymmetry index = −0.13, [75]). Third, we observed a strong narrow-band peak of power at the same frequency as the signal that was maintained throughout the entire period of stimulation (Fig 6A–6C, middle left columns). Fourth, consistent with experimental evidence (e.g., Fig 1B in [72]), first- (16 Hz) and second-order (24 Hz) harmonics of the theta modulation frequency were also present (Fig 6A–6C, middle right columns). Fifth, current source density (CSD) showed a highly regular alternating current dipole between layers with a phase reversal between SP and SR (Fig 6A–6C, far right column). This is similar to in vivo LFP recordings in the absence of perforant pathway input [70], which we did not model. Therefore, since the results did not depend qualitatively on circuit size, for the sake of simplicity, we further analyzed only the cylinder circuit.

Next, we looked at how the spiking of different neuronal classes in the model are modulated by theta oscillations ([76–79]; see S23 Table). When the spike times of neurons close to the extracellular recording electrode in SP were compared with the phases of theta-band LFP

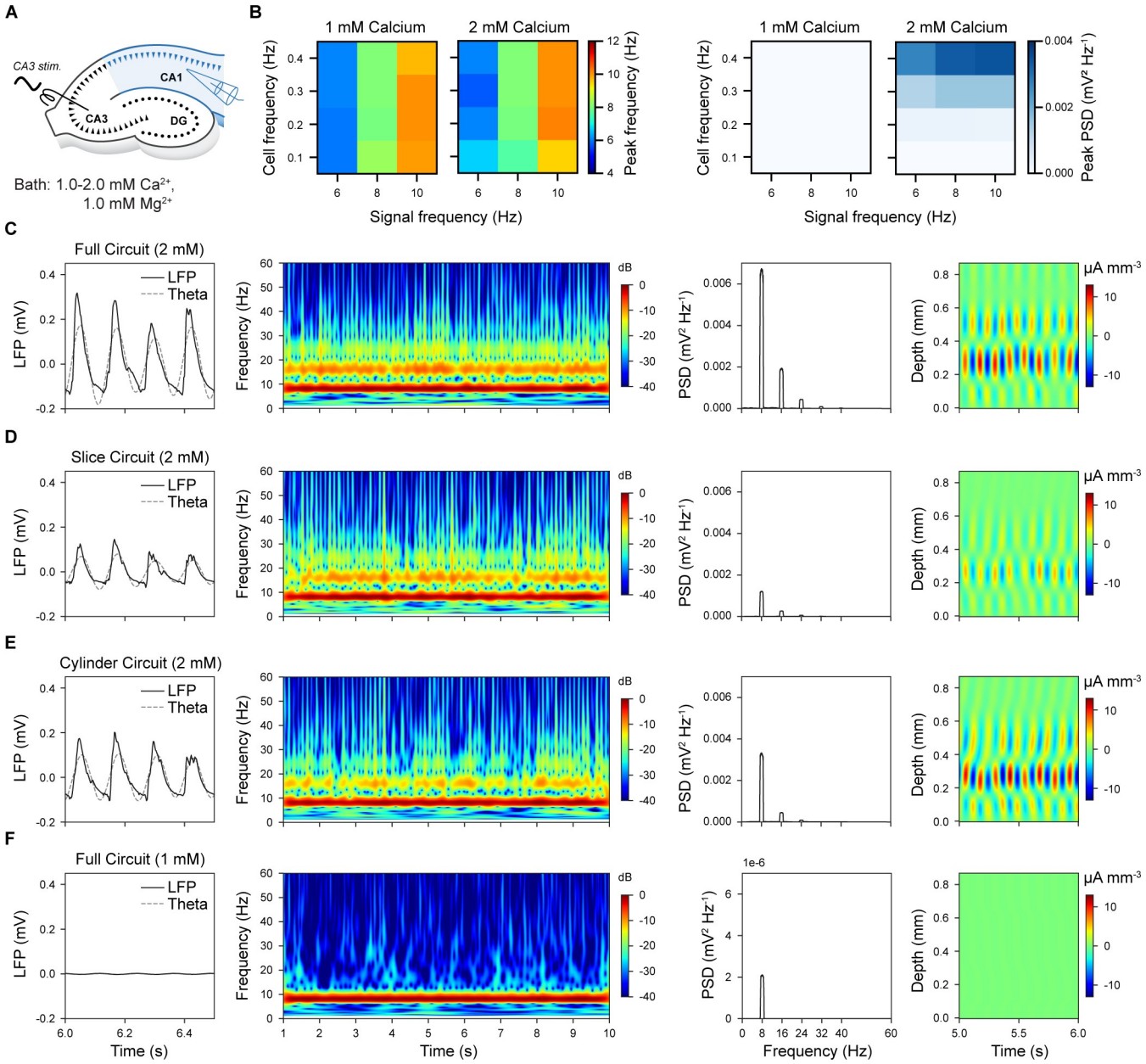

**Fig 6. CA3 theta (8 Hz) oscillatory input entrains CA1 to matched theta oscillation across different scales of circuit.** (A) Schema showing the in silico experimental setup. (B) Dependency of peak frequency (left) and PSD (right) from calcium level, cell and signal frequencies during simulations of a cylinder circuit. (C) Full circuit model (2 mM calcium). LFP recordings from SP (far left), spectrogram (left middle), PSD (right middle), CSD (far right). (D) Slice circuit model (thickness of 300 μm, 2 mM calcium). (E) Cylinder circuit model (radius of 300 μm, 2 mM calcium). (F) Full circuit model (1 mM calcium). Note that PSD has 1,000 times smaller y-axis scaling than the ones in panels A–C. CSD, current source density; LFP, local field potential; PSD, power spectral density.

rhythm (theta trough = 0˚), all neuron types were found to respond at roughly the same phase of the theta cycle (Fig 7). As the mean rate of SC afferent spiking increased, more neurons became phase-locked yielding a denser mono-phase distribution for higher signal modulation frequencies (Fig 7A). For example, under stimuli with a 0.4 Hz SC mean spiking frequency and 8 Hz signal modulation, CA1 PCs fired first during the mid-rising phase of theta and were followed by all types of interneurons, whose spiking mostly ended before peak theta, with BS

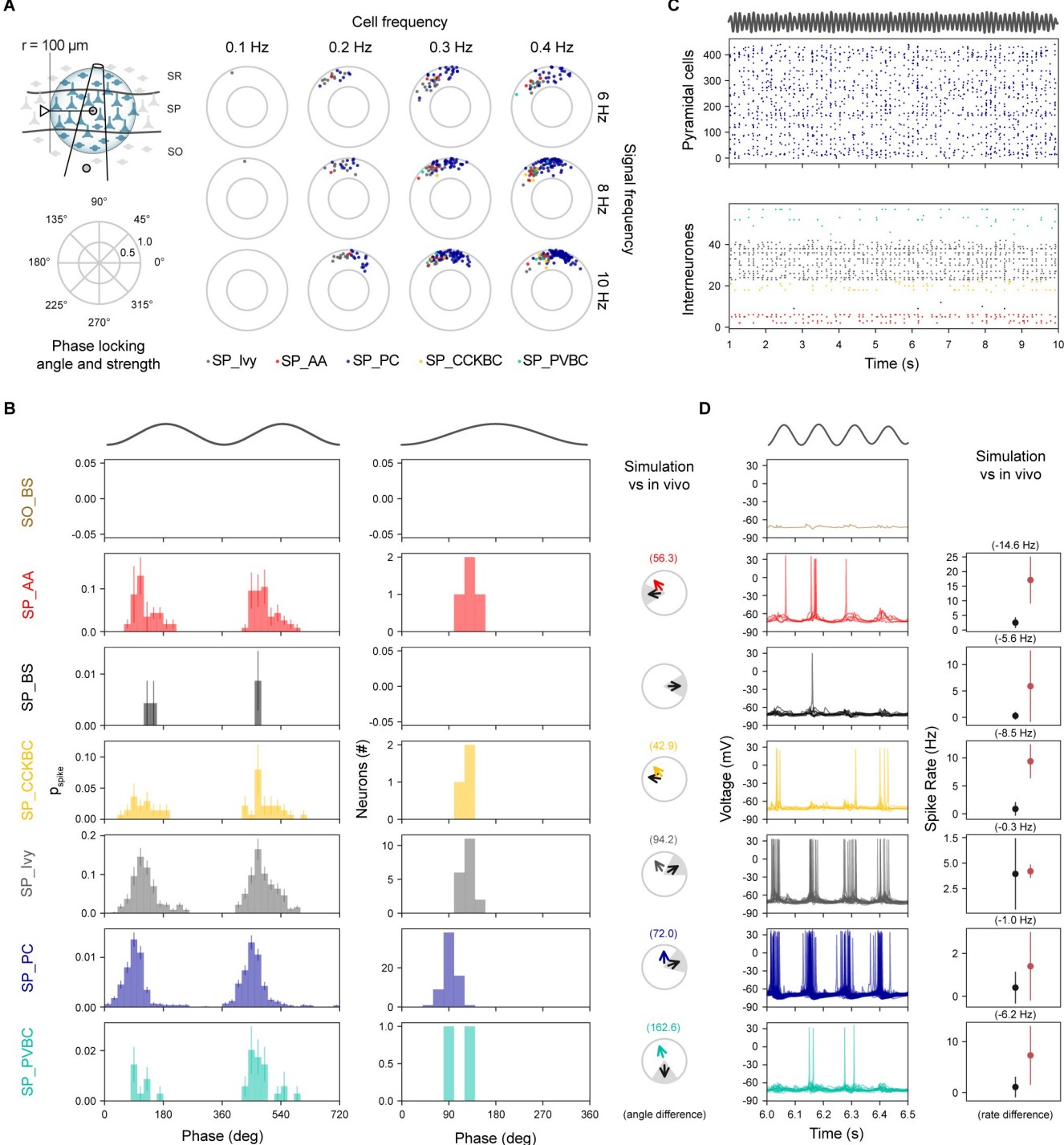

**Fig 7. CA1 morphological types are homogeneously tuned to CA3 theta oscillatory input.** (A) Neurons for analysis were selected within 100 μm radius of the stratum pyramidale electrode location (top left), shown as shaded region with the arrow indicating the radius of this region. Phase locking angle and strength for a range of individual (CA3) cell frequencies (columns) and modulation frequencies (rows). (B) Phase modulation. Spike discharge probability of all neurons grouped by morphological type (left). Phase locked neurons tuning over theta cycle for each morphological class over a single theta cycle (middle). Experimental validation of phase-locking against in vivo recordings (right) with an arrow representing the mean phase angle (experimental data = black arrow; simulated = colored arrow) and gray shaded sector indicating, where known, the experimental angular deviation. (C) Spiking raster plots. SP_PC cell spiking (top panel); LFP theta rhythm (trace above plot); interneuron spiking (bottom panel). (D) Intracellular traces from morphological cell types (left) and validation against in vivo recordings. Stimulus panels B–D: 0.4 Hz SC mean spiking frequency and 8 Hz signal modulation. All simulations shown for 2 mM calcium. Experimental values in panels B and D can be found respectively in S23 and S24 Tables. LFP, local field potential; PC, pyramidal cell; SC, Schaffer collaterals.

neurons emitting few or no spikes (Fig 7B, left). Phase-locked neurons had a tighter tuning than in vivo, with pyramidal neurons typically firing before interneurons (Fig 7B, middle). When compared with in vivo recordings of phase-locked neurons (see S23 Table), the mean phase angle of model spiking was closely matched for SP_CCKBC but substantially out of phase for SP_AA (Fig 7B, right). Although the angular deviation of phase-locking was generally tighter than observed in vivo (e.g., model versus in vivo for SP_AA 8.9˚ ($n = 4$) versus 55.0˚ ($n = 2$), SP_PVBC 12.0˚ ($n = 2$) versus 68.0˚ ($n = 5$), and SP_Ivy 10.9˚ ($n = 19$) versus 63.1˚ ($n = 4$)).

For the same neuronal classes in the model, we compared their somatic membrane potentials and mean firing rates to experimental data. Pyramidal cell spiking was closely aligned to theta LFP rhythm although individual neurons did not spike at every cycle (Fig 7C, top). SP_Ivy showed a similar pattern to SP_PC while other types of interneuron participated more sporadically (Fig 7C, bottom). Intracellular voltage traces for pyramidal and ivy cells were also similar albeit with ivy cell firing slightly later and overlapping with other types of interneurons (Fig 7D, left). Mean firing rates during theta were generally lower than observed in vivo except for ivy cells, which were a close match; SP_AA, BS, and BC (CCK+ and PV+) were well below empirical expectations (Fig 7D, right). Theta modulated the amplitude of pyramidal cell membrane potential by 1.57–7.34 mV (for 0.1–0.4 Hz SC axon frequency), consistent with the in vivo range (2–6 mV, [80]). When we compared model population synchrony during theta oscillations with in vivo data [81], we found that the percentage of SP_PC spiking was a poor match around the theta trough (0˚) but was a better match around theta peak (180˚), while fast-spiking SP_PVBC and to a lesser degree SP_AA were under-recruited (S26 Fig). Overall, for this stimulus the pyramidal-interneuron theta phase order suggests that intrinsic inhibition was activated more powerfully by recurrent than by afferent excitation. Altogether, for the CA3 input, the model does not generate theta at in vivo-like calcium levels but does reliably at in vitro-like ones. However, the phase analysis suggests that the mechanism is different from what is observed in vivo.

## Medial septum input

In vivo evidence points to a fundamental role of the MS in theta generation [71]. To model the possible role of MS-mediated disinhibition to CA1 in theta oscillations, we (i) set an in vivo extracellular calcium concentration (1 mM); (ii) applied a tonic depolarizing current (% of rheobase current) to all neurons to represent in vivo background activity; (iii) introduced an additional current to mimic the depolarizing effect of an arhythmic release of ACh from the cholinergic projection (see ACh section); and (iv) applied a theta frequency sinusoidal hyperpolarizing current stimulus only to PV+ CA1 neurons to represent the rhythmic disinhibitory action of the GABAergic projection ([82–84]; see Fig 8A). Due to uncertainty regarding some of these factors, we examined the response over a wide range of physiological conditions (see Methods). Since this required a high number of simulations, we decided to use the cylinder circuit.

Prior to the onset of the disinhibitory stimulus ("MS OFF"), the global tonic depolarization resulted in weak, irregular beta-band LFP activity in CA1 but after its onset ("MS ON"), it induced a strong and sustained, regular theta oscillation matching the frequency of the hyperpolarizing stimulus (Fig 8B). For disinhibitory modulation amplitude of 0.2 nA, the LFP waveforms generated were close to symmetrical (mean asymmetry index = 0.25±0.11; see S27 Fig). Over a range of ACh concentrations and tonic depolarization levels, this theta rhythm was robust, narrow banded (Fig 8B and 8C), and generated by a highly regular current source restricted to SP (Fig 8E). Higher ACh concentrations, while slightly reducing theta-band power, reduced the level of beta-band activity (Fig 8C). Increased levels of tonic depolarization

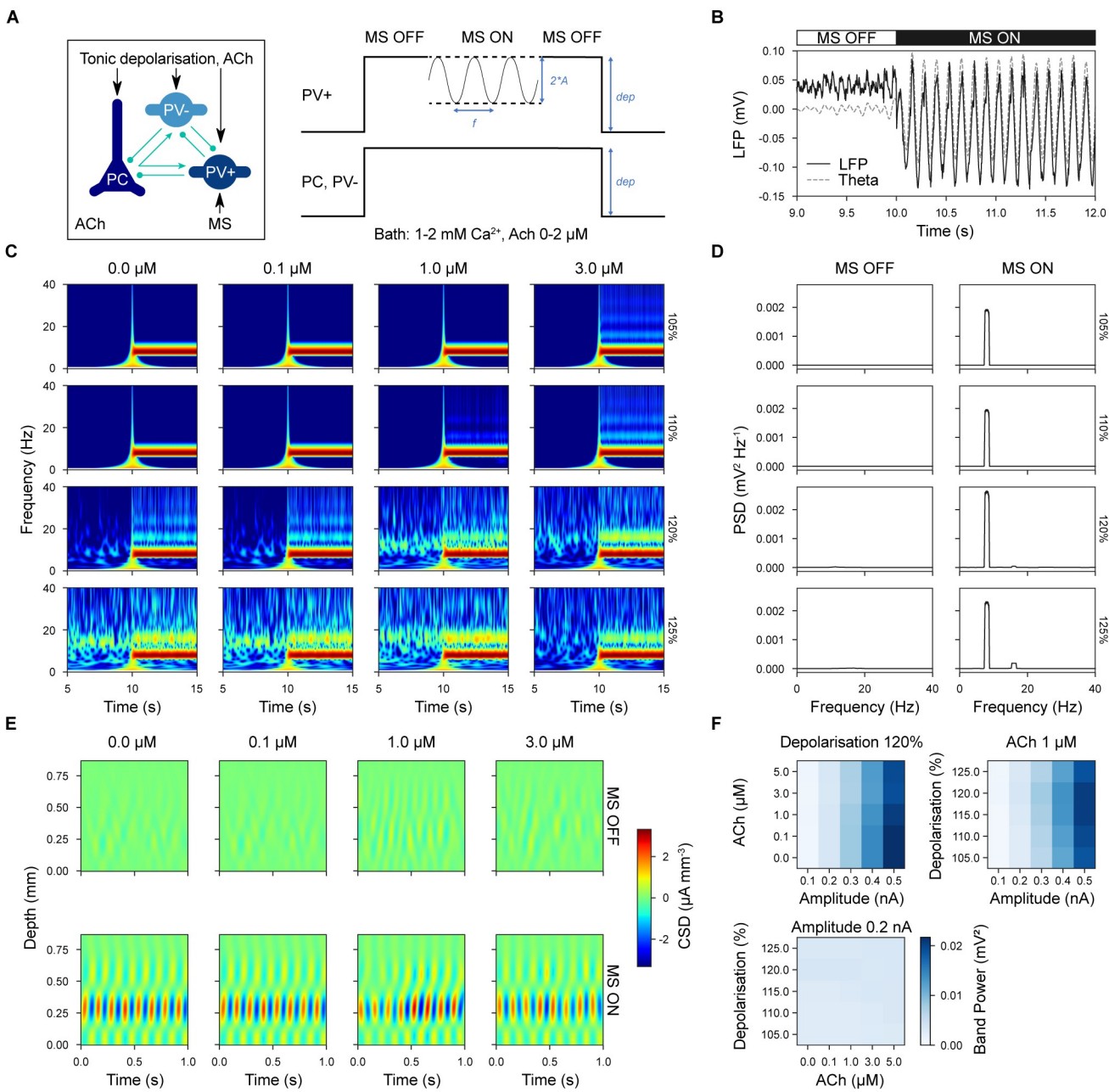

**Fig 8. MS disinhibition of parvalbumin-positive (PV+) interneurons induced theta oscillations in CA1.** (A) Simulation setup. All neurons received a tonic depolarizing current in the presence of ACh ("MS OFF" condition). For a given period, an oscillatory hyperpolarizing current was injected into PV+ interneurons only ("MS ON" condition). (B) Example of simulation before and after the onset of disinhibition. LFP in black and theta-filtered LFP in gray. (C) Spectrogram for a range of ACh concentrations (top labels) and tonic depolarization levels (right labels). (D) PSD across different levels of tonic depolarization (right labels) with and without disinhibition. (E) CSD analysis across different ACh concentrations (top labels), with and without disinhibition. (F) Theta band power as a function of the amplitude of oscillatory hyperpolarizing current, ACh concentration, and level of tonic depolarization. B–E: Stimulus disinhibitory modulation amplitude of 0.2 nA. ACh, acetylcholine; CSD, current source density; LFP, local field potential; MS, medial septum; PSD, power spectral density.

enhanced both theta harmonics and higher frequency components (Fig 8C and 8D). We observed that theta-band power was more dependent on the amplitude of the disinhibitory oscillation than on either ACh concentration or tonic depolarization level (Fig 8B and 8F). Therefore, we found MS-mediated disinhibition could strongly induce CA1 theta oscillations.

During theta oscillations, the phase of spiking of different neuronal classes here divided into 2 main groups that were in anti-phase with each other (see Fig 9). As the level of tonic depolarization increased, more phase-locked cells were detected (Fig 9A) and only above 110% depolarization (where 100% represents the depolarization necessary to reach spike threshold) were there a sufficient number of active interneurons to discern this dual grouping. Increasing ACh concentration tended to weaken pyramidal phase locking (Fig 9A). For example, at 120% depolarization and 1 μM ACh, the firing of SP_PC, SP_Ivy, and SP_CCKBC cells was broadly tuned around the theta trough and rising phase, while the firing of SP_AA, BS and SP_PVBC neuron was more narrowly tuned around the peak of the theta rhythm (Fig 9B, left). Neurons with significant phase locking matched this pattern but were even more narrowly tuned (Fig 9B, middle). The phase locking of SP_AA, SP_Ivy and PC closely matched in vivo recordings (see S23 Table) but SP_BS and BC (CCK+ and PV+) were by comparison more than 90 degrees out of phase (Fig 9B, right).

The voltage traces and rate of spiking during theta oscillations for different neuronal types was also grouped. While SP_PC did not spike on every theta cycle their firing appeared weakly modulated by theta (Fig 9C, top). Whereas for interneurons, SP_AA, BS, and SP_PVBC spiked tightly for most cycles, ivy cells spiked more rarely and SP_CCKBC more tonically (Fig 9C, bottom). Intracellular voltage traces for SP_AA, BS, and SP_PVBC showed they spiked tightly on the rebound from the release of the hyperpolarizing stimulus, whereas SP_PC and other interneurons lacking this were less reactive to theta (Fig 9D, left). Notably, all neurons spiked at a lower average rate than in vivo recordings [76–79] (Fig 9D, right). The population synchrony of SP_PC with theta trough was consistent with in vivo data [81] for a range of disinhibitory stimulus amplitudes, whereas for fast-spiking interneurons like SP_AA and SP_PVBC, synchronization with theta peak only occurred with lower stimulus amplitudes (S28 Fig). Taken together, the MS-mediated disinhibition entrained theta oscillations under in vivo-like conditions, creating a diversity of firing phases between interneuron classes, close to what has been observed experimentally.

In summary, we used the reference model to investigate several possible mechanisms for theta oscillations. For intrinsic mechanisms, we found that spontaneous synaptic release and random afferent synaptic barrage did not induce detectable theta oscillations in the model, while tonic depolarization could induce a variable and unstable theta oscillation at 10 to 12 Hz. For extrinsic mechanisms, both CA3 and MS input induced a stable and stronger theta oscillation but in different ways, where MS disinhibition was more compatible with in vivo data.

## Other frequencies

Next we asked whether the reference circuit was capable of propagating gamma oscillation using the commonly used in vitro experimental paradigm of inducing them using bath carbachol (CCh) [85–88]. Specifically, we replicated the setup of [89], where the authors added CCh to generate oscillations in CA3, which were transmitted to CA1 via SC. The simulation conditions were tailored to these in vitro experiments (i.e., 300 μm-thick slices, 2 mM extracellular $Ca^{2+}$, 2 mM extracellular $Mg^{2+}$, 10 μM ACh). We matched the shape and frequency of the input stimulus reported by [89] and followed the same LFP analysis methodology. We observed that gamma oscillatory SC input could entrain the entire CA1 network of the model slice to oscillate at the driving frequency (31 Hz) (S29A Fig). As well as inducing oscillations in

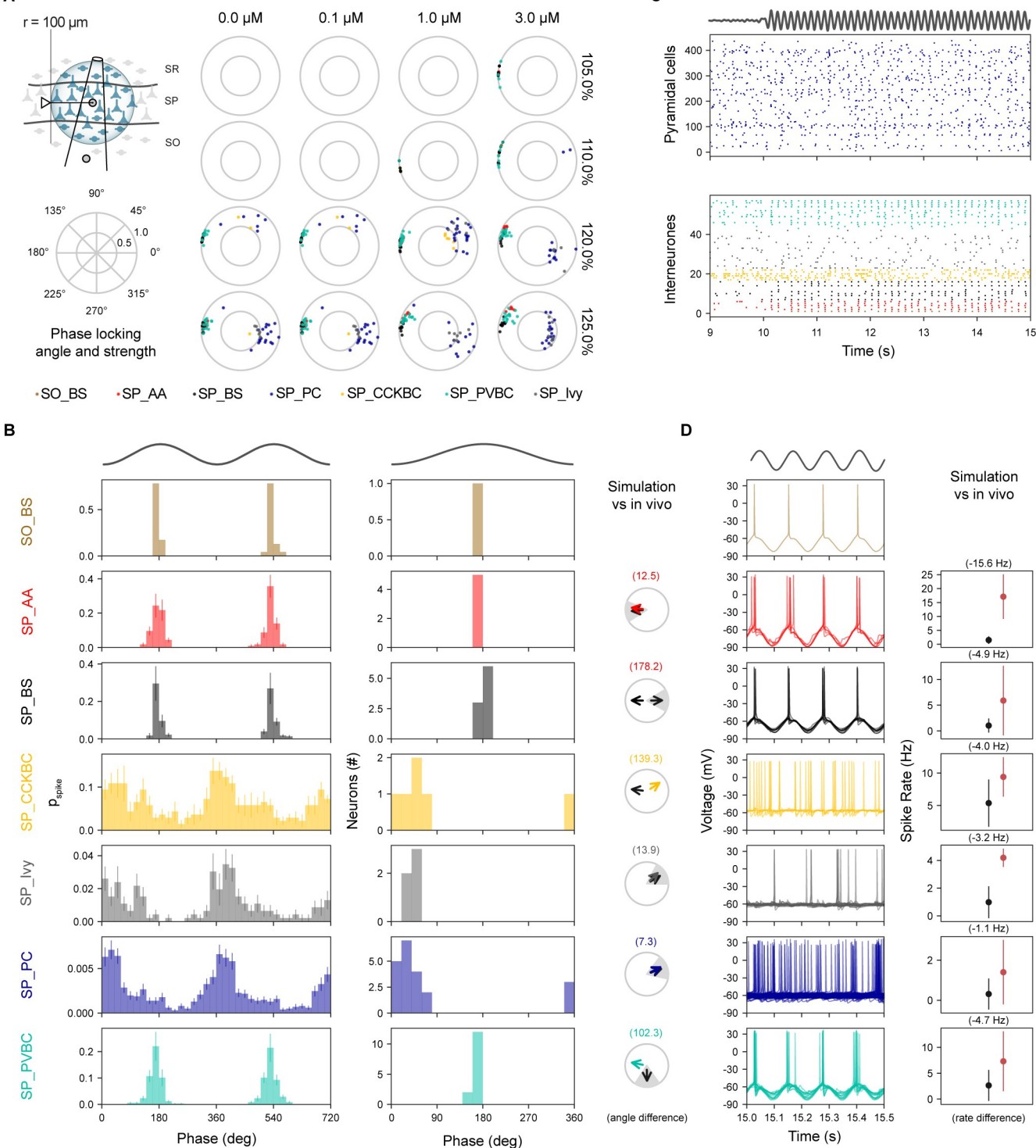

**Fig 9. MS disinhibition induced anti-phase modulation of CA1 neurons during theta cycles.** (A) Neurons for analysis were selected within 100 μm radius of the stratum pyramidale electrode location (top left). Phase locking angle and strength for range of ACh concentration (columns) and levels of tonic depolarization (rows, where 100% represents the spike threshold) for modulation amplitude of A = 0.2 nA. (B) Phase modulation. Spike discharge probability of all neurons grouped by morphological type (left). Phase-locked neurons tuning over theta cycle for each morphological class over a single theta cycle (middle). Experimental validation of phase locking against in vivo recordings (right) with an arrow representing the mean phase angle (experimental data = black arrow; simulated = colored arrow) and gray shaded sector indicating, where known, the experimental angular deviation. (C) Spiking raster plots. SP_PC cell spiking (top panel); LFP theta rhythm (trace above plot); interneuron spiking (bottom panel). Disinhibition is switched on ("MS ON") at

time 10 s. (D) Intracellular traces from morphological cell types (left) and validation against in in vivo recordings. B–D: Stimulus disinhibitory modulation amplitude of A = 0.2 nA, 1 μM ACh and tonic depolarization 120%. Experimental values in panels B and D can be found respectively in S23 and S24 Tables. ACh, acetylcholine; LFP, local field potential; MS, medial septum.

CA3, in vitro CCh could alter the response of CA1 neurons to the SC input. Thus, to quantify the effect of CCh on CA1, we repeated the simulation without the influence of CCh. SC inputs, ranging from 15,000 to 100,000 stimulated fibers, were able to induce strong gamma oscillation in the absence of CCh. However, CCh increased the number of inputs needed for stable gamma oscillation, probably due to its weakening effect on synapses (S29B Fig). Therefore, at least for this experimental setup, the circuit was capable of propagating gamma oscillations.

Finally, we investigated how the reference circuit behaved across a much wider range of SC input frequencies. Because the nature of the input was less clear across this range, we used a sinusoidal modulated stimulus. In this case, we isolated the role of SC input and we excluded the influence of modulators as ACh. In particular, we extended SC input for a wider range of cell (0.1 to 0.8 Hz) and modulation frequencies (0.5 to 200 Hz) and measured the corresponding input–output (I-O) gain and spike-spike correlation (Fig 10A). We found I-O gain was not uniform but depended on both cell and signal input frequencies (Fig 10B). The I-O responses of PC and interneurons were different, with interneuron gain greater at lower cell frequencies compared to PCs (Fig 10B). The strongest overall CA1 gain was obtained with a mean CA3 frequency of 0.4 Hz. The spike-spike correlation also depends on both cell and signal input frequencies (Fig 10C–10E, upper). In the case of input cell frequency of 0.4 Hz, we found CA1 spiking activity was strongly correlated for delta- to lower gamma-band input signal frequencies (i.e., between 1 and 30 Hz) but weaker outside this range. Internal CA1-CA1 spike correlation was similar but spanned higher frequencies of the gamma-band (Fig 10C–10E, lower). Therefore, the model predicts that oscillations propagate better within the delta- to low gamma-band range.

## Discussion

### Main summary

This study presents the reconstruction and simulation of a full-scale, atlas-based reference model of the rat hippocampal CA1 region based on community data and collaboration. We extended and improved the framework of [27] to curate and integrate a wide variety of anatomical and physiological experimental data from synaptic to network levels. We then systematically applied multiple validations for each level of the model. We augmented the resulting highly detailed intrinsic CA1 circuit with a reconstruction of its main input from CA3 and a phenomenological model of neuromodulation by acetylcholine. Importantly, the reference circuit model is, by definition, general. It is capable of addressing a range of research questions because its parameters are adjusted only to different experimental setups, not tuned each time to specifically reproduce individual experimental results. For example, to help explain network activity observed in both health and disease, with the model one can block different neurotransmitters or specific groups of synapses, change the proportions of different cell types, modify synaptic strength or short-term plasticity, or alter the properties of ion channels in specific cell types to study their consequences on network dynamics. However, the range of the research questions it can address is necessarily limited by the current extent of the model and its inputs (see later). To demonstrate its general utility, we were able to simulate different scales of circuits and investigate the generation and transmission of neuronal oscillations, with particular emphasis on theta rhythm, for a variety of stimulus conditions.

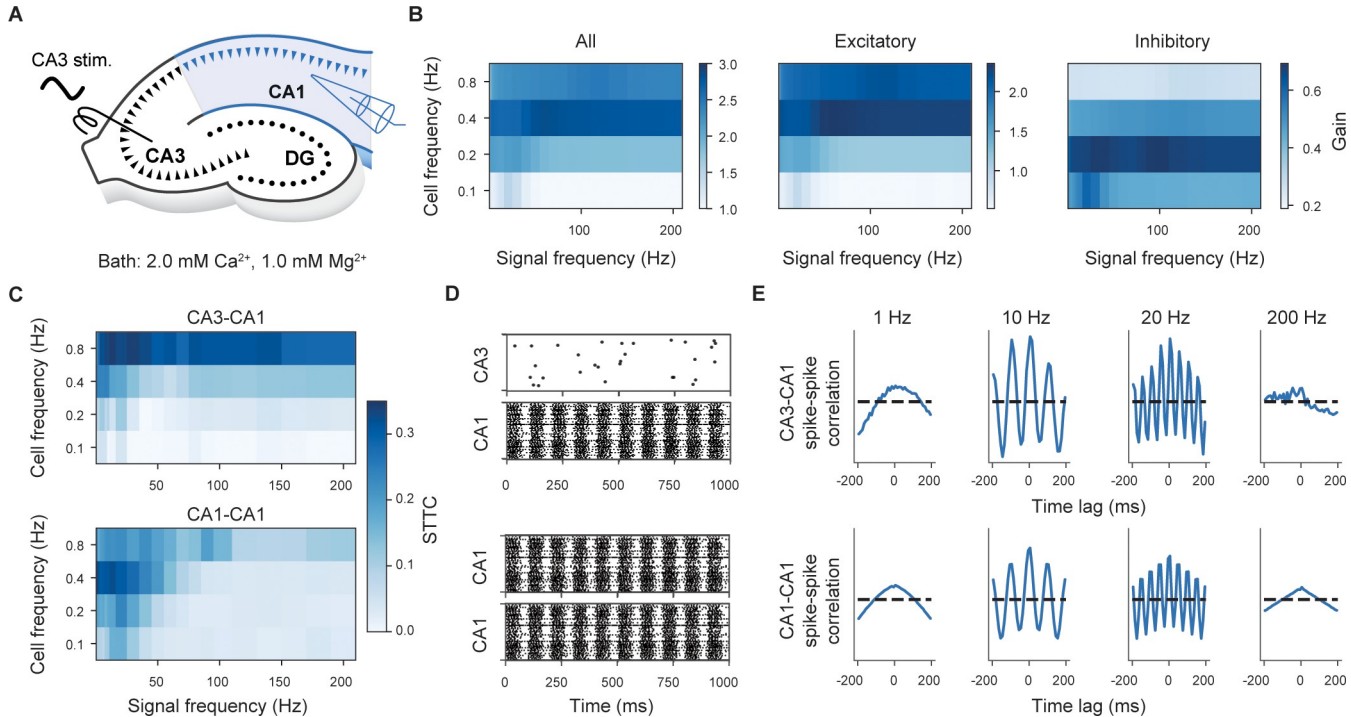

**Fig 10. Intermediate frequencies propagate more efficiently through SC.** (A) In silico experimental setup. (B) Ratio between the number of CA1 and CA3 spikes as a function of input cell and signal frequencies. We considered all CA1 neurons (left), CA1 PCs (center) and CA1 interneurons (right). (C) Heatmaps representing the computed STTC values as a function of input cell and signal frequencies, for CA3-CA1 and CA1-CA1 neurons. (D) Examples of CA3 and CA1 spike train (cell frequency of 0.4 Hz, signal frequency of 10 Hz); 100 random CA3 neurons (i.e., SC fibers) and CA1 neurons are selected for clarity. The same neurons are used to compute the STTC and spike-spike correlations (panels D and E). (E) Spike-spike normalized correlation histograms for 4 signal frequencies (cell frequency of 0.4 Hz) for CA3-CA1 and CA1-CA1 neurons. SC, Schaffer collaterals; STTC, spike time tiling coefficient.

## Previous work and limitations

While a full review of the many hippocampal circuit models is beyond the scope of the paper, we focus on the progression in both the size and level of detail of large, multiscale models of the rat hippocampus during the last 3 decades (for a comparison of their key features with the present model, see S2 Table). These biologically realistic models aim to explain the complex dynamics of hippocampal activity, in particular the generation and control of rhythmic responses. However, all these models, including the one reported here, are incomplete descriptions of a hippocampal region or regions because of the paucity or even absence of some types of data necessary to constrain them. Moreover, the results of these models are difficult to compare because (1) there is no commonly agreed-upon set of validations with which to benchmark a circuit model against experimental data; and (2) there are fundamental differences in their composition, organization, and underlying assumptions.

In terms of model validation, circuit models of isolated CA1 have, ever since the reports of [72], been expected to generate theta oscillations intrinsically (see [22]). Later research by the same lab, however, raises some doubts about this interpretation of the results. In an isolated whole mouse hippocampal preparation, as used by [72], [90] found that inactivating the subiculum abolished theta (5 Hz) activity in CA1 and the remaining 2.5 Hz oscillations in CA1 matched that of intact CA3 (see their S6 Fig). So the theta oscillations in isolated CA1 observed in [72] might be explained if the subiculum was not removed when isolating CA1 from CA3 (see S10 Fig of [72]). Therefore, these later results imply that CA1 may not be able to generate robust theta oscillations intrinsically but instead its oscillations are induced by subiculum and

CA3 inputs, which is what we observed here for CA1 circuit model for CA3 input simulations. Yet, while it is clear from the current model that different mechanisms can generate CA1 theta oscillations (degeneracy; for a review of this concept, see [91]), the cell-class specific phasing of spiking of circuit neurons during theta LFP oscillations does not offer a good overall match to in vivo experimental results [77–79,92].

In terms of model construction, the current model stands out by realistically constraining neurons and their connectivity by the highly curved shape of CA1 rather than by relying on an artificial space as found in other contemporary models. Additionally, it reflects both STP and spontaneous synaptic release, well-established characteristics of central nervous system synapses. In addition, the morphologies and electrical properties of model neurons here are not just copies of the same class exemplars but their properties have been systematically varied to better capture the diverse nature of neuronal circuits and their responses to stimulation. However, compared with [22], some elements are still missing from the current model such as neuroglia-form cells, which did not exist in our available data set, and $GABA_B$ R which are not included in our simulations. Yet, neither the current model nor [22] incorporates other established features of CA1 such as dorsal-ventral differences in neuronal and synaptic properties [93] or burst firing of pyramidal neurons [94].

Nonetheless, the current model includes the perforant path-associated and trilaminar interneurons, which were absent from all previous models. In addition, we modeled the NMDA synaptic currents observed in both hippocampal pyramidal cells and interneurons (with specific NMDAR conductance, rise and decay time constants for each pathway) that are absent in the model of [22]. Furthermore, the connectivity algorithm used for the current model generates an intrinsic connectome with more realistic high-order statistics than the more prescriptive approach used in the [22] model (see [42] for the analysis). Unlike [24], we did not replicate the topography of the afferent projections, which may play a role in patterning the circuit response, but did model the projections and circuit at a full rather than reduced scale. Overall, further improvement to our model requires additional experimental data.

While we incorporated key features distributed among previous models into a single, general model (see S2 Table), it is important to recognize that our aims and approach were different, representing a step change in hippocampal modeling. The intention of the framework was first to curate and integrate community data into the model, preserving provenance for reproducibility, in a way that would allow the addition of new datasets from the large hippocampal community. Re-using these data sets and then making them publicly available through the hippocampushub.eu supports the 3R principles (replace, reduce, refine) for the reduction of animal experiments. Each circuit component and the final model was then systematically validated in an open and transparent way to a degree not previously attempted by other research groups. To increase the realism and utility of simulation experiments, we sought to approximate experimental conditions (e.g., slice thickness and location, bath calcium, magnesium and acetylcholine concentrations, and recording temperature) and to increase the capability to manipulate and record from the model (e.g., enable spontaneous synaptic release, alter connectivity, estimate extracellular LFP signals, and apply a variety of stimuli). In short, the aim was to offer a more realistic yet scalable and sustainable approach to model brain regions at full scale.

## Lessons learned

In the context of a community effort, the process of curating and integrating available data to reconstruct a brain region and replicating the experimental conditions in silico proved instructive in a number of ways. First, assembling the components to reconstruct a brain region

reveals surprising gaps in the available data and knowledge. Notably, for example, while Schaffer collateral input to CA1 has received many decades of attention, especially in terms of long-term plasticity, we found the basic information needed to model this pathway quantitatively, was limited. To address this gap, we devised a multi-step algorithm constrained by the data that were available to parameterize these connections. Second, when data was available it often required further work before it could be used in the model. While an open-source rat hippocampal atlas [39] was crucial to reconstruct CA1, the original volumetric reconstruction was too noisy for our purposes and required additional processing to give smooth layering. This smoothness was necessary to place and orient morphologies accurately in the atlas in relation to the layers. If the morphology was incorrectly placed or oriented, this had a knock-on effect for how the circuit was connected. Similarly, the completeness of morphological reconstructions also affected connectivity. For these reasons, some cell types in our available data set could not be used in the circuit model, sacrificing a small amount of cell type diversity in favor of completeness. Third, setting up simulations to reproduce the desired experimental conditions required careful attention. We offer 2 examples from our research. When reproducing the I-O gain of SC afferent input reported in [56], we initially sampled all neurons in the model slice to plot to the I-O curve. However, the result was poor. We later resolved this by reproducing how the neurons were sampled in the experiments with which we could closely match the empirical curve. When reproducing MS-induced theta oscillations, we initially simulated under default conditions of extracellular calcium concentration at 2 mM, resulting in theta oscillations that occurred episodically and only for a restricted parameter regime. However, when we lowered the extracellular calcium to in vivo levels (1 mM), sparser activity led to more robust and stable theta oscillations.

## A community-based modeling approach

The model was built and simulations run through the cooperation of several laboratories each with different expertise. Since we could not access a standardized core set of data made for the purpose of modeling as done previously [27–29], instead, we curated and integrated data from collaborating labs which followed different protocols. The majority of single neuron morphologies and recordings, for example, came from University College London (UCL) [76,95–102]. From these data, single neuron models were created between the Blue Brain Project (BBP) and Italian National Research Council (CNR) [35] and these were then validated by a computational lab at Institute of Experimental Medicine, Budapest (KOKI) [103]. Similarly, physiological data from paired recordings that characterized individual synaptic pathways were provided by an experimental lab in KOKI and then curated and integrated together with BBP [34]. Subsequently, BBP used these single neuron and synapse models to build and share the circuit model so computational labs at BBP, CNR, and KOKI could simulate various hippocampal use cases, only some of which have been presented here. Combining the framework of [27] with community data and collaboration resulted in the generalization and improvement of data curation and integration methods for more varied data, improvements in tools like BluePyOpt for optimizing neurons, and the development of a new tools such as HippoUnit to systematically validate and compare different single neuron models.

This approach offers important features that make an attractive case for adoption by the wider hippocampus community. First, anchoring a circuit model in the volumetric space of a brain region atlas makes mapping experimental data for data integration, validation, and prediction easier than for more abstract spaces and permits investigations at different scales, e.g., a brain slice cut at an arbitrary angle. The Allen Brain Atlas has demonstrated the advantages of registering community experimental data in a common reference atlas [104]. Extending this

principle, a common framework appears advantageous for modeling as well. Second, the model components, validations, and circuit are openly available through a dedicated portal (hippocampushub.eu) to maximize transparency and reproducibility. We allow the community to examine how the reference circuit model was built, validated, simulated, and analyzed. We conceived the model to be adopted by the community, extended and improved. Third, while the traditional approach of constructing circuits for a specific use case has short-term advantages for demonstrating proof-of-concept, in the long-term a community reference model must be valid across a range of use cases.

To illustrate its utility, we suggest how the circuit model could be developed by the community to help address key scientific questions concerning structure and function of hippocampus. An example use case under in vitro-like conditions could be to use slice circuit models to study network responses to different stimuli based on observed difference in neuronal and synaptic properties between dorsal and ventral CA1 [93]. Under in vivo-like conditions, the full-scale circuit model could be employed to investigate the factors necessary for the emergence of traveling and standing wave phenomena reported from CA1 LFPs [105]. To increase the level of biological accuracy at the cellular level, for example, a new class of neurons might be added to the circuit (for practical steps required, see S1 Appendix). To achieve greater accuracy at the circuit level, particularly in investigating place cells, significant additions would be necessary to match the current level of biological accuracy in the model. In particular, the anatomy and function of the perforant pathway (PP) would need to be reconstructed using available empirical data as we did here for the SC pathway. This would involve understanding quantitatively the synaptic connectivity and unitary monosynaptic EPSPs from individual PP axons onto CA1 neurons, ideally from single neuron paired recordings. Additionally, the simulation patterns of these PP and SC projection axons would need to reflect the activity of EC and CA3 neurons during rat locomotion to activate place cells realistically. Finally, to model the formation and remapping of place cells, model synaptic connections would need to be endowed with long-term plasticity (e.g., [106]). While the level of effort required is significant, the potential benefits of creating a model-experiment loop, where experimental data are incorporated in the model (experiments to model) and model results used to generate experimentally testable predictions (model to experiments) should not be underestimated in minimizing the cost in terms of animals and materials.

## Methods

### Experimental procedures

In this study, we used 2 data sets for single neuron morphological reconstructions and electrophysiological recordings from the region CA1: Sprague Dawley rat and Wistar rat. Both data sets also include the reconstruction of the layers that were used to estimate the layer thicknesses (see section Layers) and guide the cell placement (see section Soma placement). Note that, while the Wistar rat data set also includes electrophysiological recordings, we did not use them for this model. The experimental procedures have been previously described in [35], but are summarized below.

### Sprague Dawley rat data set

Procedures involving animal care and use were approved by the University of London or University College London ethical committees and by the British Home Office with regard to the Animal Scientific Procedures Act 1986 (PPL 80/2131, 80/2598). Hippocampal slices were obtained from young adult rats (Sprague Dawley, 90 to 180 g) as previously described [76,95–102]. Briefly, following deep anesthesia with euthatal solution, young adult rats (Sprague

Dawley, 90 to 180 g) were perfused transcardially with ice-cold modified artificial cerebrospinal fluid containing in mM: 248 sucrose, 25.5 $NaHcO_3$, 3.3 KCl, 1.2 $KH_2PO_4$, 1 $MgSO_4$, 2.5 $CaCl_2$, 15 D-glucose, equilibrated with 95% $O_2$/5% $CO_2$; 450 to 500 μm coronal sections were cut, transferred to an interface recording chamber and maintained at 34−36°C in modified ACSF solution for 1 h and then in standard ASCF (in mM: 124 NaCl, 25.5 $NaHCO_3$, 3.3 KCl, 1.2 $KH_2PO_4$, 1 $MgSO_4$, 2.5 $CaCl_2$, and 15 D-glucose, equilibrated with 95% $O_2$/5% $CO_2$) for another hour prior to starting electrophysiological recordings. Single intracellular recordings were made using sharp microelectrodes (tip resistance, 90 to 190 MΩ) filled with 2% biocytin in 2M $KMeSO_4$ under current-clamp (Axoprobe; Molecular Devices, Palo Alto, California, United States of America). Electrophysiological characteristics of the recorded neurons were obtained from voltage responses to 400 ms current pulses between −1 and +0.8 nA and recorded with pClamp software (Axon Instruments, USA).

Following recording and biocytin filling, slices of 450 to 500 μm were fixed overnight [4% paraformaldehyde (PFA), 0.2% saturated picric acid solution, 0.025% glutaraldehyde solution in 0.1 M phosphate buffer] as previously described [97,107]. Slices were then cut in 50 to 60 μm sections, cryoprotected with sucrose, freeze-thawed, incubated in 1% $H_2O_2$ and then in 1% sodium borohydride (NaBH4). Slices were incubated overnight in ABC (Vector Laboratories) and then in DAB (3, 3′ diaminobenzidine, Sigma) to reveal the morphology of the recorded neurons. Following washes, sections were postfixed in osmium tetroxide ($O_sO_4$), dehydrated, cleared with propylene oxide, mounted on slides (Durcupan epoxy resin, Sigma), and cured for 48 h at 56°C. For immunofluorescence staining, slices were incubated in 10% normal goat serum after incubation with $NaBH_4$. Sections were then incubated overnight in a primary antibody solution containing mouse anti-PV (Sigma), rabbit anti-PV [108] or mouse anti-gastrin/CCK (CURE, UCLA) made up in ABC and then in a solution of secondary antibodies (Avidin-AMCA, FITC and anti-rabbit Texas Red) for 3 h. Following fluorescence visualization, slices were incubated in ABC, DAB, and $OsO_4$ prior to dehydration and mounting as described above. All neurons were then reconstructed in 3D using a Neurolucida software (MBF Bioscience) and a 100× objective as previously described [107].

## Wistar rat data set

The project was approved by the Swiss Cantonal Veterinary Office following its ethical review by the State Committee for Animal Experimentation. All procedures were conducted in conformity with the Swiss Welfare Act and the Swiss National Institutional Guidelines on Animal Experimentation for the ethical use of animals (license VD3389). Hippocampal slices were obtained from young adult rats (Wistar, postnatal 14 to 23 days) as previously described [27,35]. In brief, ex vivo coronal preparations (300 μm thick) were cut in ice-cold aCSF (artificial cerebrospinal fluid) with low $Ca^{2+}$ and high $Mg^{2+}$. The intracellular pipette solution contained (in mM) 110 potassium gluconate, 10 KCl, 4 ATP-Mg, 10 phosphocreatine, 0.3 GTP, 10 HEPES and 13 biocytin, adjusted to 290±300 mOsm/Lt with D-mannitol (2±35 mM) at pH 7.3. Chemicals were from Sigma Aldrich (Stenheim, Germany) or Merck (Darmstadt, Germany). A few somatic whole cell recordings (not used for this model) were performed with Axopatch 200B amplifiers in current clamp mode at 34±1°C bath temperature.

The 3D reconstructions of biocytin-stained cell morphologies were obtained from whole-cell patch-clamp experiments on 300 μm thick brain slices, following experimental and post-processing procedures as previously described [109]. The neurons that were chosen for 3D reconstruction were high contrast, completely stained, and had few cut arbors. Reconstruction used the Neurolucida system (MicroBrightField, USA) and a bright-field light microscope (DM-6B, Olympus, Germany) at a magnification of 100× (oil immersion objective, 1.4 to 0.7

NA). The finest line traced at the 100× magnification with the Neurolucida program was 0.15 μm. The slice shrinkage due to the staining procedure was approximately 25% in thickness (Z-axis). Only the shrinkage of thickness was corrected at the time of reconstruction.

## Morphological classification

We classified the morphologies into one of 12 different morphological types (m-types) based on the layer containing their somata and their morphological features. For the classification, we adopted 3 main assumptions. (1) Several subtypes of PCs have been described [108,110–116], but for simplicity we consider the pyramidal cells as a uniform class (SP_PC or simply PC). (2) SP_PVBC and SP_CCKBC were classified as 2 separate m-types. The 2 types of basket cells can be distinguished by biochemical markers and electrical properties, and they show different densities and connectivity [5]. On the contrary, there is no strong evidence for differences in their morphologies and the small number of examples in our possession (3 SP_PVBCs, 1 SP_CCKBCs) prevented us from conducting any systematic classification. While we could have pulled the 2 cell types into one m-type, we kept them separated for the sake of simplicity. (3) We created supersets of m-types based on their principal biochemical marker. In particular, we defined PV+ cells as SP_PVBC, SP_BS, SP_AA, CCK+ cells as SP_CCKBC, SR_SCA, SLM_PPA and SOM+ (somatostatin) cells as SO_OLM, SO_BS, SO_BP, SO_Tri. When the layer is not specified (e.g., BS), we meant all the neurons of this type regardless their soma location.

## Morphology curation

We curated the morphological reconstructions extensively prior to insertion in the network model. First, we translated the reconstructions to have their somas centered at coordinates (0,0,0). We reoriented the reconstructions so that their x, y, and z axes coincided respectively with the transverse, radial, and longitudinal axes they followed in the tissue. We considered the cells to be substantially complete, with only a few cuts, so we have not applied any corrections for cuts [27]. The only exception is the PC axon. One particular reconstruction has a relatively long axon (3,646 μm) that spans 1,325 μm along the transverse axis. We assumed this axon to be relatively complete within the CA1 and we used it as a prototype for all the PCs. Validation of the number of synapses per connection, bouton density, and connection probability showed that this assumption is reasonable (see section Building CA1). A subsequent cloning procedure (see section Morphology library) guaranteed that all the PC axons are unique. We removed 2 pyramidal cells due to the complete absence of an axon since at least the axon initial segment (AIS) needs to be present to create electrical models.

We applied a series of corrections as described in [27]. In brief, to comply with the NEURON and CoreNEURON standards, we repaired the reconstructed morphologies using NeuroR [117] (Table 1). In particular, we used this tool to remove unifurcations of dendrites, fix neurites whose type changes along its main branch, fix invalid formats of soma, and remove segments with close to zero length. We used this tool also to correct tissue shrinkage. The correction increased the neurite reach approximately 25% along the z-axis and approximately 10% along the x- and y-axes.

## Morphology library

To produce morphology variants that fit well in different locations of the space, we applied scaling ±15% of the original reconstructions. This range was a good compromise between the need to change the size but yet to not introduce too much distortion [27]. To increase morphological diversity, we applied a cloning procedure as described in [27]. Some clones showed a

**Table 1. List of software used.**

| Software name | Source | Identifier |
|---|---|---|
| atlas-direction-vectors | BBP/EPFL software package | https://github.com/BlueBrain/atlas-direction-vectors |
| bbp-workflow | BBP/EPFL software package | https://github.com/BlueBrain/bbp-workflow |
| BluePySNAP | BBP/EPFL software package | https://github.com/BlueBrain/snap |
| BluePyEfe | BBP/EPFL software package | https://github.com/BlueBrain/BluePyEfe |
| BluePyOpt | BBP/EPFL software package | https://github.com/BlueBrain/BluePyOpt |
| BluePyMM | BBP/EPFL software package | https://github.com/BlueBrain/BluePyMM |
| Brainbuilder | BBP/EPFL software package | https://github.com/BlueBrain/brainbuilder |
| Brayns | BBP/EPFL software package | https://github.com/BlueBrain/Brayns |
| circuit-build | BBP/EPFL software package | not yet open source |
| CoreNEURON | BBP/EPFL software package | https://github.com/BlueBrain/CoreNeuron |
| eFEL | BBP/EPFL software package | https://github.com/BlueBrain/eFEL |
| Elephant | Elephant authors and contributors | https://doi.org/10.5281/zenodo.1186602 |
| EMSim | BBP/EPFL software package | https://github.com/BlueBrain/EMSim |
| Hippounit | KOKI software package | https://github.com/KaliLab/hippounit |
| ITK-SNAP | University of Pennsylvania | http://www.itksnap.org/ |
| morphology-workflows | BBP/EPFL software package | https://github.com/BlueBrain/morphology-workflows |
| mtspec | pypi python package | https://pypi.org/project/mtspec/ |
| neo | The NeuralEnsemble Initiative | https://github.com/NeuralEnsemble/python-neo |
| Neurodamus | BBP/EPFL software package | https://github.com/BlueBrain/neurodamus |
| NeuroMorphoVis | BBP/EPFL software package | https://github.com/BlueBrain/NeuroMorphoVis |
| NeuroM | BBP/EPFL software package | https://github.com/BlueBrain/NeuroM |
| NeuroR | BBP/EPFL software package | https://github.com/BlueBrain/NeuroR |
| projectionizer | BBP/EPFL software package | https://github.com/BlueBrain/projectionizer |
| psp-validation | BBP/EPFL software package | https://github.com/BlueBrain/psp-validation |
| regiodesics | BBP/EPFL software package | https://github.com/BlueBrain/regiodesics |
| TMD | BBP/EPFL software package | https://github.com/BlueBrain/TMD |
| voxcell | BBP/EPFL software package | https://github.com/BlueBrain/voxcell |

wrong distribution of axons and dendrites, and we removed them. In particular, we annotated how different parts of the morphology were positioned within the layers (see section Layers). We used these annotations to guide the cell placement, but it was also useful to discard cells that were too distorted by cloning and had an incorrect distribution of neurons within the layers. Clones were further validated by visual inspections. For OLM cells, a scaling of ±15% was not sufficient to make sure that the axon correctly targeted SLM in all dorso-ventral positions. In this case, we introduced a synthetic axon. In the reconstruction of OLM cell, a chunk of 60 μm axon, which is placed at 85 to 145 μm from the soma, falls in SR and it is relatively simple. To produce morphologies with about 90% to 150% (step of 5%) of the height of the original morphology, we removed the aforementioned chunk of 60 μm axon (93%), only 45 μm of it (95%), or added a multiple of 45 μm (corresponding to a step of 5%) synthetic axon within SR until we reached a 150% scaling.

After we corrected the orientation of the initial set of reconstructions, the cloning procedure changed the branch lengths and rotations, which may have resulted in a change in the main orientation of the cells. Based on visual inspection, the effect on short neurites was negligible. In the case of PCs, the axon is relatively longer and has a clear orientation [118]. In this case, the cloning may significantly alter the orientation of the axon, and this may result in a wrong placement in the circuit (see section Soma placement). To avoid this problem, we

renormalized the orientation of the cloned PCs. In particular, we used principal component analysis (PCA) to first determine the principal axis of all axonal points in each cloned morphology. Next, we used the direction of the principal axis to rotate the caudal portion of the arbor onto the x-axis so all pyramidal axon arbors were aligned in the same direction. Finally, to remove an unrealistic degree of variability, we filtered out cloned morphology outliers whose width (z-axis range) was 3.3 times greater than that of the original pyramidal axon arbor from which they were derived. This filtering step rejected about 14% of all pyramidal cell cloned morphologies.

## Morphology library validation

Neuronal morphologies were validated at the end of the processing stage (cloning) by comparison of their morphological properties, as well as their topological profiles. Around 100 morphological features from dendritic and axonal neurites were extracted using the NeuroM package [119]. The median over visible spread (MVS) statistical score, defined as the ratio of the absolute value of the difference between the medians of the feature ($\tilde{F}$) divided by the overall visible spread (OVS), was computed according to the following formula:

$$MV\ S(F) = \frac{|\tilde{F}_{P_r} - \tilde{F}_{P_c}|}{OVS(F_{P_c}, F_{P_r})} \tag{1}$$

to identify the difference between populations of reconstructed $P_r$ and cloned $P_c$ neurons for each feature $F$. This score is close to zero for similar populations and increases as the difference between the populations increases. Scores below 0.3 indicate good agreement between the 2 populations as their differences are contained within 3 standard deviations (STD). Note that this measurement depends on the sample size, due to the computation of the OVS, which remains small if the sample size is small and therefore the MVS score increases accordingly.

In addition to the traditional morphometrics, the topological profiles of the original reconstructions were compared to those of the processed morphologies (repaired and cloned). The TMD encodes the start and the end radial distances of each branch from the soma surface [120]. The pairs of distances (start, end) are represented in a 2D plane, a representation known as the persistence diagram of the neurons (S6 Fig). The Gaussian kernels of these points are averaged to generate the persistence images (S7 and S8 Figs). The persistence images of 2 populations can be subtracted to study the precise differences between data sets. Note that the robustness of average persistence images depends on the sample size of the population, when only few (less than 3) cells are available these images represent a small subset of the biological population and are therefore not quantitatively reliable.

## Electrical type classification

We classified the cells into different electrical types (e-types) based on their firing patterns as defined by the Petilla nomenclature [33]. Some of the firing patterns (i.e., irregular patterns) were too rare and we excluded them. For SP_CCKBC we do not have traces, and we considered them to be classical accommodating (cAC) [96].

## Morpho-electrical compositions

The 154 recordings were recorded from different m-types, and we observed that each m-type can show one or more firing patterns with different probabilities. We used this information to derive the morpho-electrical type (me-type) composition (S4 Table).

## Ion channels

We considered a set of active membrane properties which included a voltage-gated transient sodium current, 4 types of potassium current (DR-, A-, M-, and D-type), 3 calcium channels (CaN, CaL, and CaT), a nonspecific hyperpolarization-activated cation current (Ih), 2 Ca-dependent potassium channels (KCa and Cagk), and a simple $Ca^{2+}$-extrusion mechanism with a 100 ms time constant. We based channels' kinetics on previously implemented cell models of hippocampal neurons [121–124].

## Single-neuron modeling

Single-cell models are described in previous publications [34,35,125]. In brief, we optimized cell parameters to match electrophysiological features rather than the traces directly. We extracted features using the open-source Electrophys Feature Extraction Library (eFEL, Table 1) or the Blue Brain Python E-feature extraction (BluePyEfe, Table 1). Initial optimizations considered only features from somatic recordings [35], while a subsequent refinement included also the amplitude of the back-propagating action potential (in the apical trunk, 150 and 250 μm from the soma) for PCs [34].

We assumed channels were uniformly distributed in all dendritic compartments except $K_A$ and $I_h$, which in pyramidal cells are known to increase with distance from the soma [126,127]. Pyramidal cells include $K_M$ in the soma and axon [128], $K_A$ with different kinetics in dendrites, soma and axon [121,129], $K_M$ with a different kinetics in the soma and the axon, Na and $K_{DR}$, whereas they do not include $K_D$ since the delayed spiking is not a feature observed in PCs. Given the limited knowledge of the currents in interneurons, we applied the same currents of pyramidal cells with the following exceptions. Dendritic sodium channel densities decay exponentially with distance from the soma (with a length constant of 50 μm) based on [130]. We included $K_D$ since some interneurons show delayed firing. $K_A$ has the same kinetics in somas and dendrites because there is no experimental evidence of a different $K_A$ kinetics in the dendrites of interneurons. We distinguished 2 types of $K_A$ for proximal and distal dendrites [121]. For both pyramidal cells and interneurons, we optimized channel peak conductance independently in the different regions of a neuron (soma, axon, and dendrites).

We performed a multi-objective evolutionary optimization using the open source Blue Brain python optimization library (BluePyOpt, Table 1) [131] to obtain 39 single-cell models. From this set, we excluded 3 models because their me-types are not used in the network model (cAC_SP_Ivy, cAC_SP_PVBC, cNAC_SR_SCA).

## Single neuron model validations

We used HippoUnit [103] (Table 1) to validate the electrical models of pyramidal neurons, specifically considering the attenuation of PSP and BPAP.

## PSP attenuation

The PSP attenuation test evaluates how much PSP amplitude attenuates as it propagates from different dendritic locations to the soma. To get the behavior from the model, EPSC-like currents (i.e., double exponentials with rise time constant of 0.1 ms, decay time constant of 3 ms, and peak amplitude of 30 pA) were injected into the apical trunk of PCs at varying distances from the soma (100, 200, 300 ± 50 μm), and PSP amplitudes were simultaneously measured at the local site of the injection and in the soma. Finally, the experimental and model data points were fitted using a simple exponential function. The space constants resulting from the fitting were then reported and compared with experimentally observed data in [37].

## BPAP attenuation

The BPAP test evaluates the strength of AP back-propagation in the apical trunk at locations of different distances from the soma. The AP is triggered by a step-current of 1 s with an amplitude for which the soma fires at approximately 15 Hz. The values were then averaged over distances of 50, 150, 250, 350 ± 20 μm from the soma. We measured the amplitudes of the first AP at the 4 different dendritic locations and compared them with experimentally observed data in [36].

## Library of neuron models

Optimizing all the neurons in the morphology library is computationally expensive. Following [27], we minimized the problem by using Blue Brain Python Cell Model Management (Blue-PyMM, Table 1) which combines the morphology library with initial single-cell models to produce a library of cell models. The procedure accepts a new combination if the model and experimental features are within 5 STD of the experimental feature. If at least 1 feature is greater than this range, then the new combination is excluded. This strategy also has the advantage of increasing the model variability within the experimental data. In the case of bAC-SLM_PPA, the threshold for accepting new combinations had to be relaxed to 12 STD to obtain some valid models.

## Rheobase estimation

For all the single-cell model combinations, we estimated their rheobase with a bisection search until an accuracy of e-3 was reached. The spikes were recorded at the AIS, and the upper bound from the last step of the search was used, to ensure cells spiked in the axon at rheobase.

## Atlas

**Volume.** We based our annotated volume on a publicly available atlas of the CA1 region [39] (http://cng.gmu.edu/hippocampus3d/). From the original atlas, we took all voxels labeled as CA1 without maintaining their subdivisions in CA1a, CA1b, and CA1c or the 4 layers (S10A Fig). The original file was converted from a csv to a nrrd format using voxcell (Table 1).

We undertook a series of post-processing steps to augment the atlas with a coordinate system and vector fields that followed the 3 hippocampal axes (longitudinal, transverse, radial). First, a substantial smoothing operation was necessary to render the surfaces within the CA1 regular enough for subsequent manipulations. To obtain smoother surfaces, we applied a Gaussian filter together with a morphological closing filter. A few manual checks were needed to ensure isolated voxels were removed and holes from missing tissue closed. The resulting volume has a minor difference in voxel counts compared with the original atlas (−3.7%) (S10B Fig).

We created a mesh for the boundary surface of the CA1 using Ultraliser (Table 1) and separated it into an upper and lower shell (S10C Fig). In practice, the operation of separating the mesh into 2 portions is challenging and we were not able to automate it. The curved structure of the hippocampus makes automatic solutions appear incorrect upon visual inspection, particularly around the ridges. Taking the results into account, we used an OpenGL 3D graphical user interface tool (atlas-direction-vectors, see Table 1) that allows one to paint voxels on the surface of the atlas in different colors. It enabled us to manually correct the voxel selection until the shell division appeared satisfactory. Two surface meshes were then derived from the selected voxel masks.

We generated a polygonal centerline along the innermost voxels of the CA1 from the dorsal to the ventral extremity (S10D Fig). In detail, this process started by computing the distance

transform of the input volume by assigning each voxel the distance from its closest neighbor outside of the volume. The 2 extremity points were used as entry and exit locations to build a stochastic chain of points following the local maxima of the distance transform. A further step generated a graph of these possible points using proximity conditions and determined the overall shortest path between the extremities using a weighted Dijkstra algorithm [132]. The resulting skeleton of the centerline was then converted into a Bézier curve [133] and its continuous derivative was used to orient a series of planes. These planes are oriented perpendicularly to the curve and cut the volume of the hippocampus in slices at regular intervals (S10E Fig).

**Coordinates system.** To utilize the 3D volume fully, we created a set of parametric coordinates ranging from 0 to 1 and named them *l*, *t*, *r* as they follow the longitudinal, transverse, and radial axes of the hippocampus (S10F Fig). The longitudinal coordinate was assigned first using the derivative of the centerline to sample the volume with many cross-section planes and assigning each CA1 voxel the [0, 1] value of the plane closest to it.

The transverse coordinate was obtained by considering the intersection between each of the cross-section planes and the meshes assigned previously as the upper and lower shells of the volume. Each plane cuts the meshes creating 2 lines of points which were fitted to spline functions and re-sampled to yield the same number of points each having a [0, 1] *u* coordinate. Connecting the upper to the lower line points resulted in a field of vectors that represent the natural orientation of pyramidal neurons in CA1. The transverse coordinate *t* was assigned from the *u* value of the spline points to all voxels found on the plane according to which was their closest vector (i.e., the line segment joining the upper *usp(u)* and lower *lsp(u)* line points with the same *u*).

The radial coordinate was assigned following the previous step and represents the relative [0, 1] location of each voxel of the plane along the segment connecting the *usp(u)* and *lsp(u)* points. It is defined as the ratio:

$$\frac{||uss(p), voxel||}{||uss(p), lss(p)||} \tag{2}$$

In summary, all voxels lying on the same *l* plane are assigned a *t* coordinate from the upper and lower splines and later those voxels on the same *t* segment are assigned a relative *r* position along it. Taken together, these 3 coordinates provide a coherent system to slice and parse the volume which is more robust to surface irregularities compared to other methods. Finally, for each voxel we re-computed its direction vector from the partial derivative of the (*l*, *t*, *r*) coordinates with respect to *r* to ensure perfect correspondence between the vector field and coordinate space (S10G Fig).

**Layers.** Having achieved the necessary smoothing and orientations over the entire volume, we reintroduced the distinction into layers (S10H Fig) that are used to constrain cell density and the placement of specific portions of neural morphologies such as the axon and the apical dendritic tuft (S11 Fig). For this reason, we used a combined approach to determine layer position: extracting thicknesses from morphology reconstructions and comparing the final layer volumes with voxel counts from the original atlas. During morphology reconstruction, we traced layer thicknesses from 23 images of pyramidal cells in CA1 slices and averaged them: SLM: 146±27 μm, SR: 279±40 μm, SP: 59±15 μm, SO: 168±30 μm.

The available morphological reconstructions came from the dorsal CA1, and thus these layer thicknesses reflect the dorsal portion rather than the entire extent of the region. To reintroduce layer labels back into the atlas, we used the radial coordinate computed above to assign a uniform proportion of voxels to each layer (SLM: 0.224, SR: 0.42791, SP: 0.090, SO: 0.258). This assumption returns the same relative layer thickness throughout the volume,

automatically compensating for the different CA1 shape in each location thanks to the [0, 1] parametric range. As a final step, we computed the total volumes for each layer (SLM: 3.2143, SR: 6.8421, SP: 1.5789, SO: 4.8178 mm$^3$).

## Cell composition

We started by fixing the PC density in SP to $264 \pm 32.6 \times 10^3$ mm$^3$ [134]. By combining the PC density and SP volume as extracted from the atlas (1.5789 mm$^3$), we estimated the total number of PCs as 416,842. To estimate the number of cells or cell densities for interneurons, we used the ratio between PCs and the different interneurons as predicted by [5] with several assumptions. In some cases, a cell type has been described as present in several layers. If we only have reconstructions of cells from one layer, we consider all the expected cells to be placed in this layer. In the case of bistratified (BS) cells, we have reconstructions from SO and SP, but [5] also described a small percentage of cells in SR. In this case, we considered the cells from SR to be placed in SP; [5] gave a rough estimation of 1,400 cells for trilaminar cells and radiatum-retrohippocampal cells together in SO. Without any other information, we considered these 2 cell types to have the same proportion, so we estimated the number of trilaminar cells as 700. We ignored the m-types for which we do not have at least 1 reconstruction. The lack of some cell types led to a smaller number of interneurons. To maintain the same E/I ratio of 89:11 [5], we increased the number of interneurons accordingly. The resulting cell composition counts and densities are shown in S3 Table.

## Cell placement

**Soma placement.** We expanded the basic algorithm of [27] to take into account the more complex volume of the atlas and the particular constraints of the hippocampus. For each me-type, the cell positions are created given cell density volumetric data (using uniform distribution). The total cell count is calculated based on cell density values and the volume. Each voxel is populated with the desired count of cells, weighted by the contribution of that voxel to the total density so that the total count of cells is reached. For SR_SCA and SLM_PPA, we could not select morphologies for all the potential positions within the corresponding layers. For this reason, the soma placement was restricted to 2 narrow subvolumes within the layers. In the case of SCA, we placed the soma in the middle of SR ± 5% of the layer thickness. In the case of PPA, we placed the cells in the lower part of the layer, from 2% to 10% of the layer.

**Cell orientation.** Once we decided the soma positions, we oriented the neuronal morphologies to follow the curvature of the hippocampus and other additional constraints. For all the cells, we aligned their y-axis with the radial axis of the hippocampus, and this allowed us to compute the placement scores and select the morphologies that best fit the space. Cells may show a preferential orientation around the radial axis. For interneurons, the experimental evidence for a specific orientation in the transverse-longitudinal plane is scarce, and we applied a random rotation around the radial axis to avoid any bias in the orientation. On the contrary, literature has reported a particular orientation for pyramidal cell axons. Pyramidal cells normally have 2 main branches, roughly parallel to the transverse axis, one towards the subiculum and one towards the CA3. There is also a thinner branch that is roughly parallel to the longitudinal axis [118,135]. To take this into account, we rotated the pyramidal cells so that their axons were parallel to the transverse axis and the most complex branch points closer to subiculum.

**Morphology selection.** Once we identified the cell positions, we selected the morphologies that most closely matched a set of rules to ensure correct neurite targeting. We distinguished 2 types of rules: strict and optional ones (see later in this section). Optional rules are shown in S6 Table, while we used only one strict rule, i.e., dendrites and axons should be below the top of SLM.

For each soma position, we have a candidate pool of morphologies to be selected. The candidate pool is computed as follows. For each soma position, we consider the radial axis passing through it and compute the relative position of the soma and the layer boundaries. For each morphology, we associate a score $\hat{S}$ that describes how well the rules are matched. If $\hat{S} = 1$ we have a perfect match, $\hat{S} = 0$ the rules are not matched and the morphology is excluded from the pool. The score combines a score for "optional" rules $\hat{I}$ and "strict" rules $\hat{L}$.

$$\hat{S} = \hat{I} \bullet \hat{L} \tag{3}$$

$\hat{I}$ combined all the scores from the $n$ "optional" rules $I_j$ as follows:

$$\hat{I} = \left( \frac{\sum_j I_j^{-1}}{n} \right)^{-1} \tag{4}$$

The use of a harmonic mean allows us to penalize low scores for a particular rule heavier than a simple mean, but still "to give it a chance" if other interval scores are high. If some optional score is close to zero ($< +0.001$), the aggregated optional score would be zero. If there are no optional scores or if optional scores are ignored $\hat{I} = 1$.

Each rule $I_j$ is computed as follows:

$$I = \max \left( \frac{\min(a^{\uparrow}, r^{\uparrow}) - \max(a^{\downarrow}, r^{\downarrow})}{\max(a^{\uparrow} - a^{\downarrow}, r^{\uparrow} - r^{\downarrow})}, 0 \right) \tag{5}$$

Where $(a^{\uparrow}, a^{\downarrow})$ is the interval of the dendrites, axon or dendritic tuft (when applicable) corresponding to the rules. It corresponds to the interval as defined above that needs to be shifted by the soma position $y_0$ of the morphology in the circuit.

$$(a^{\uparrow}, a^{\downarrow}) = (a_0^{\uparrow}, a_0^{\downarrow}) + y_0 \tag{6}$$

$(r^{\uparrow}, r^{\downarrow})$ is the interval of the target region along the radial axis passing through $y_0$. This interval corresponds to the rules expressed as relative intervals in the table above, converted to µm.

The numerator represents the overlap of the 2 intervals, while the denominator normalizes the overlap by the largest interval among $(a^{\uparrow}, a^{\downarrow})$ and $(r^{\uparrow}, r^{\downarrow})$. In this way, only a perfect overlap receives a score of 1, while if one interval is larger than the other, the score is lower than 1.

$\hat{L}$ combines all the scores from the "strict" rules $L_k$ as follows:

$$\hat{L} = \min_k L_k \tag{7}$$

If there are no strict scores $L = 1$.

Given the multiple constraints described above, 2.6% soma positions could not have any associated morphologies, and they were ignored. This resulted in fewer neurons than expected. However, validation of the cell composition and densities reassured us that this discrepancy is not significant. In fact, lower densities occurred mainly at the border of the circuit where the particular distorted shape of the layers makes the cell placement more complicated.

## Circuit sections

We defined a series of sections of the circuit to be used in analyses and simulations. In particular, we defined nine cylinders equally spanned from position 0.1 to 0.9 along the longitudinal axis, in the middle of the transverse axis (0.5) and with a radius of 300 µm. In addition, we defined 47 consecutive nonoverlapping transverse slices of thickness of 300 µm.

## Synapses

**Local synapse anatomy.**   To derive the connectome, we adapted the algorithm described in [40]. The first part of the algorithm finds all potential synapses (appositions or touches) among cells based on colocalization of presynaptic axon and postsynaptic cells. We allowed all the possible connections between m-types with the exception of AA cells that can form contacts only with PC. We allow synapses to occur on soma and dendrites, with the following 2 exceptions: PC-PC connections only occur on dendrites [43] and AA cells make synapses only on AIS of PCs [136]. Initially, SCA, Ivy, and BS were found to make too many synapses on the PC soma compared to the data reported in literature [76,79,136,137]. Since the number of expected synapses on the soma is relatively low, and since the tool does not allow us to prune synapses on specific compartments, we decided not to allow SCA, Ivy, and BS to make synapses on PC somas.

An apposition occurs when the presynaptic axon is within a threshold distance (maximum touch distance) from the postsynaptic cell. In the somatosensory cortex microcircuit, this distance was set to 2.5 μm and 0.5 μm for synapses on PCs and INTs, respectively [27]. However, when we applied the same thresholds to the hippocampus, we could not match some of the experimental data. In particular, compared to experimental data there were too many connections among PCs and too few between PCs and INTs [138]. Using a grid search approach, we found that maximum touch distances of 1.0 μm and 6.0 μm for synapses on PCs and INTs respectively represent the minimum values that guarantee a sufficient number of synapses to match experimental data [138] and a certain room for the subsequent pruning step.

The second part of the algorithm discards synapses (pruning) in order to match experimental data on bouton densities (mean and standard deviation) and numbers of synapses per connection (mean). As experimental data were available for only a few pathways, we had to make additional assumptions for the uncharacterized pathways. According to [40], the number of appositions per connection can be used to predict the number of synapses per connection. We plotted the number of appositions as found by the model and the number of synapses for characterized pathways, and found that we can describe a good relationship among the 2 by splitting the data points into 2 groups (I-I connections and the rest) and fitting them separately ($y = 0.1096x$ for I-I, $R = 0.401, p = 0.325$; $y = 1.1690x$ for the rest, $R = 0.267, p = 0.828$) (S13 Fig). For mean bouton density, we applied an average bouton density from characterized pathways to uncharacterized ones. Finally, the standard deviation of bouton density is estimated from the mean bouton density for the given pathway and the coefficient of variation (CV) estimated from a well-characterized pathway which we can generalize to all the other pathways. For somatosensory cortex, [40] used a CV of 0.32. For the hippocampus, the best source is [139], which reported that the 64 PV postsynaptic cells receive 99 boutons from PVBC. They also reported that 51 neurons received 1 synapse, while 13 neurons received from 2 to 4 synapses, for a total of 99 to 51 = 48 synapses. So, the 13 neurons have on average 48/13 = 3.69 synapses per connection. We estimated the standard deviation to be 1.08 by considering 51 neurons with 1 synapse, and 13 neurons with 3.69 synapses. The resulting coefficient of variation is then 1.08/1.55 = 0.70. We scanned several CVs between these 2 extremes and checked how well the resulting connectomes matched experimental data on several parameters. We obtained the best results with a CV of 0.50, and we applied it to all the pathways.

As reported in [40], a bouton may form synapses onto 2 different postsynaptic neurons and we need to take this into account to better use data on bouton density to constrain the connectome. We used the value of 1.15 for synapses per bouton.

**Local synapse physiology.**   The methodology for the design of the synapse models, the extraction of the parameters, and their implementation is described in detail in [34] with the

only exception of increasing the sample size of the number of pairs (i.e., 10,000 pre-post neurons) to increase the robustness of the calibration (see below). In brief, synapses were modeled with a stochastic version of the Tsodyks–Markram model [23,140,141], featuring multivesicular release [142]. We used sparse data from the literature (S13 and S14 Tables) to parameterize the model in a pathway-specific manner. We aimed at characterizing the physiological properties of synapses like PSC rise and decay time constants, receptor ratios (NMDA/AMPA), and STP profiles, which can be directly input into the model after some corrections (e.g., for calcium concentration, temperature, and liquid junction potential). We set the synaptic reversal potential ($E_{rev}$) to 0 mV for AMPA and NMDA receptors, and to −80 mV for $GABA_A$ receptors; $\tau_{rise}$ = 0.2 ms for AMPA and $GABA_A$ and 2.93 ms for NMDA receptors [76,143–149].

We sampled 10,000 connected pairs from the circuit and replicated paired recordings in silico to calibrate $N_{RRP}$ (the number of vesicles in the release-ready pool) and peak synaptic conductance to match in vitro PSC CVs and PSP amplitudes, respectively ([34,142]). Because of the sparsity of experimental data, at the end of this exercise, we only had 6 and 14 pathways (out of 130) with calibrated $N_{RRP}$s and synaptic peak conductances, respectively. For the uncharacterized pathways, we had to generalize the values found.

As in a previous study, the release probability of the synapses scales nonlinearly with the extracellular calcium concentration, enriching their dynamical regime even further [27]. The scaling of the release probability is made using a Hill function with 3 possible coefficients (steep, intermediate, and shallow). NMDA/AMPA receptor ratios are pathway dependent. Physiological evidence about release probability scaling and NMDA/AMPA ratios exists only for a minimal number of pathways. Thus, as the last step, we grouped the characterized pathways into 22 classes based on neurochemical markers, STP profiles, and peak synaptic conductances and used class average values predicatively for the remaining uncharacterized pathways.

## Schaffer collaterals (SC)

**SC synapse anatomy.** In the case of Schaffer collaterals, we did not model presynaptic CA3 pyramidal cells, and their axons, but directly the synapses on the postsynaptic neurons. For this reason, we followed an approach different from the one used for CA1 internal synapses, and we used the tool Projectionizer (Table 1), already adopted to model thalamo-cortical and cortical-thalamic projections [27–29]. In brief, the generation of projections was a multi-step workflow as described below.

The number of CA3 PCs was constrained considering the physiological ratios between CA3 PCs and CA1 PCs, as reported by [5], an approach consistent with the rest of the cell composition. Having estimated the number of presynaptic neurons (i.e., 267,238), we determined the number of afferent synapses on CA1 PCs to be 20,878 (the average of the range reported in Table 22 of [5]), and the number of afferent synapses on interneurons to be 12,714 (the average of the range reported in Table 26 of [5]). Details on the experimental data used to constrain SC anatomy are reported in S15 Table.

To connect CA3 PC axons (i.e., SCs) with CA1 neurons, for each region in CA1, we gathered all the dendrite segment samples in the voxelized atlas. Neuron somas were not considered viable targets. Each region-wise candidate segment pool was then subsampled. Each region was assigned a number of synapses based on the synapse distribution (i.e., SLM: 0.3%, SR: 67.9%, SP: 7.1%, SO: 24.7%) and the total afferent synapses. The drawing from the pool was done with replacement, and sampling was weighted by the segment length not to oversample short segments. The sampled segments were considered as candidates for placing synapses. For each sampled segment, a random synapse position was chosen along the segment. Then, the candidate synapses were randomly assigned to each of the CA3 PCs presynaptic neurons.

Finally, synaptic physiology parameters were drawn from distributions with specific means and standard deviations (see section SC physiology). To match the average number of SC on PC and INT, the workflow was run separately for SC to PC and SC to INT.

**SC physiology.** To define the synaptic parameters, we distinguish 2 pathways: SC-PC and SC-INT. In both cases, we used the Tsodyks–Markram model parameters estimated by [53] (S17 and S18 Tables), since we did not have experimental traces to estimate the parameters (as done with the internal synapses). We also set the NMDA/AMPA ratio, rise and decay time constant for NMDAR according to available data (S17 and S18 Tables). The remaining parameters were not available, and we optimized them using two-steps procedure that is schematically reported in S19B Fig. In the first step, we set placeholder values for rise and decay time constant for AMPAR, and optimized maximum synaptic conductance and $N_{RRP}$, while in the second step, we optimized the AMPAR time constants. In each step, we aimed to match a set of experimental measures that depend on the considered pathway. We used a grid search and selected the model parameters that minimized a cost function defined as follows:

$$error = \sum \frac{|x_i^{mod} - x_i^{exp}|}{x_i^{exp}} \tag{8}$$

Where $x^{exp}$ is the experimental value we want to match, and $x^{mod}$ is the value produced by the model. Because of the strong parameter interdependence, after each step, the other step was rerun in order to verify if both held true. The cycle ended when the 2 steps converged toward a satisfactory solution (i.e., model and experiments are not statistically different). As done for the definition of the internal CA1 synaptic parameters, we ran all the steps selecting 10,000 random pairs of pre- and postsynaptic neurons.

**Definition of SC-PC synaptic parameters.** To constrain the peak synaptic conductance, $N_{RRP}$, rise and decay time of AMPA and NMDA receptors, we aimed to match mean, CV, rise and decay time constant, and half-width of the PSP as reported in [50] (S17 Table). We set up the simulations to best approximate the same experimental conditions (2.0 mM $Mg^{2+}$ and 2.0 mM $Ca^{2+}$).

**Definition of SC-INT synaptic parameters.** To constrain the peak synaptic conductance and $N_{RRP}$, we used [54] as a reference. The authors distinguished between cannabinoid receptor type 1 negative (CB1R-) basket cell (BC) and CB1R+ BC. Each cell type has a very distinctive response to SC stimulation, measured as the ratio between EPSC from SC to BC and the EPSC from SC to PC ($EPSC_{BC}/EPSC_{PC}$) (S18 Table). We considered the CB1R- and CB1R+ BC to be representative respectively of all the CB1R- and CB1R+ INTs. We considered CB1R+ INTs to include PPA, CCKBC, and SCA, while CB1R- INTs the rest, according to the molecular marker profiles reported on www.hippocampome.org [7,8].

Following this strategy, we treated the 2 populations, CB1R- and CB1R+ INTs, separately, and we optimized the peak synaptic conductance and $N_{RRP}$ to match the corresponding EPSC ratio. We set up the simulations to best reproduce the experimental conditions (i.e., 1.3 mM $Mg^{2+}$, 2.5 mM $Ca^{2+}$, and NMDAR blocked). For each combination of parameters, we simulated 1,000 voltage-clamp experiments (neurons clamped at −85 mV as in the experiments) where we stimulated with a single spike one SC connected to one PC and one interneuron. For each of the 1,000 triplets SC-PC-interneuron, we have simulated 100 trials to have a sufficient number of traces to compute robust statistics of EPSCs. PSC ratios were computed only when there was a PSC in both PC and INT of the triplet. For each triplet, the mean PSC values were computed.

To optimize rise and decay time constants of AMPAR, we aimed to match the EPSP-IPSP latency as reported by [55] (S17 Table). We set up the simulations to reproduce the same

experimental conditions (1.3 mM $Mg^{2+}$, 2.5 mM $Ca^{2+}$, and NMDAR blocked, with and without inhibition). For the sake of simplicity, we optimized the parameters of PVBC given their important role in feedforward inhibition [54,55] and then generalized the resulting parameters to the other INTs. To identify a feedforward loop SC-PV-PVBC, we proceeded as follows: (1) randomly select 1 PVBC; (2) select 1 PC connected to the PVBC; (3) select 200 SC fibers that innervate the PVBC. A simultaneous stimulation of 200 SC fibers is necessary to trigger an AP in PVBC; and (4) check whether at least one of these SC fibers also innervates the PC. If this is not the case, we repeat the procedure. We selected 1,000 SC-PC-PVBC triplets. For each triplet, we simulated 35 trials (different seeds). Each trial was 900 ms long, with spikes simultaneously delivered to the 200 SC fibers at 800 ms. We then made an average of the included voltage traces for each triplet and computed the EPSP with and without the inhibition. We derived the IPSP trace by subtracting the trace with inhibition from the trace without inhibition (S19E1 Fig). Traces with EPSP or IPSP failures, without PC or INT spikes, or more than 1 spike were excluded from the analysis. Since some parameter combinations led to few valuable traces and this could bias the result, we included the number of surviving traces in our cost function (Eq 8), where $x^{mod}$ becomes the number of usable traces and $x^{exp}$ the total number of traces (i.e., 35,000).

**SC validation.** The reconstruction of the SC were validated against the results of [56] on input-output (I-O) characteristics of SC projections in vitro. We set up the simulations to mimic the same experimental conditions (slice of 300 μm, $Ca^{2+}$ 2.4 mM, $Mg^{2+}$ 1.4 mM, 32˚C) (Fig 4A). As in the experiment, we randomly selected 101 SP neurons in the slice. We randomly chose 350 SC inputs to activate simultaneously, evoking an AP at t = 1,000 ms. This input was able to make all the 101 neurons spike corresponding to 100% of input/output (Fig 4B). We activated a different percentage of the 350 SC fibers, ranging from 5% to 100%, and quantified the number of spiking neurons. We repeated the protocol by blocking the GABAergic synapses and repeated each condition (with and without GABAR) in 5 different slices. We computed the Pearson correlation coefficient R in control conditions to assess the linearity of IO curve in control condition (with GABAR).

## Cholinergic modulation

To build a model of the effects of cholinergic release, we began by collecting and curating literature findings from bath application experiments in which ACh, carbachol (CCh), or muscarine were used. We extracted data on their effects on neuron excitability (membrane potential, firing rate, S19 Table), synaptic transmission (PSP, PSC, S20 Table), and network activity (extracellular recordings, S21 Table). We subsequently verified that the other experimental conditions (such as cell type, connection type and mouse versus rat provenance) did not produce further data stratification.

Since for each experiment, data was available for both control and drug-applied conditions, we estimated the relative amount of depolarizing current at different concentrations of ACh. For sub-threshold data, we computed the cell type-specific amount of current causing a change in the resting membrane potential equal to the mean value reported in the experiments. For supra-threshold data, we computed the amount of current needed to increase the firing rate from the baseline frequency to the frequency in ACh conditions.

The data points show a dose-dependent increase in depolarizing current. To describe this effect, we fitted a Hill function assuming the ACh-induced current at control condition is zero ($R^2 = 0.691$, $N = 28$).

$$I_{depol} = \frac{0.567 ACh^{0.436}}{100^{0.436} + ACh^{0.436}} \tag{9}$$

where $I_{depol}$ is the depolarizing current (in nA) and ACh is the neuromodulator concentration in μm (Fig 5A and 5C).At the synaptic level, the data points show a dose-dependent decrease in the amplitude of the voltage/current response to pre-synaptic stimulation for all pathways analyzed (S20 Table). Since ACh affects synaptic transmission principally at the level of release probability [23,60,61], we used the Tsodyks–Markram model [23,140,141] and introduced scaling factor to make the parameter $U_{SE}$ (neurotransmitter release probability) dependent on ACh concentration with a Hill function fitted to experimental data ($R^2 = 0.667$, $N = 27$).

$$U_{SE}^{ACh} = \frac{1.0ACh^{-0.576}}{4.541^{-0.576} + ACh^{-0.576}} U_{SE} \tag{10}$$

Where $U_{SE}$ is the release probability (without ACh), and $U_{SE}^{ACh}$ is the release probability with the dependency on ACh concentration (Fig 5B and 5D).

As a control, we verified that the values of PSP or PSC are proportional to $U_{SE}$.

For each network simulation, we applied these equations to compute the effect of ACh concentration on cells and synapses. In particular, we inject the same amount of depolarizing current (Eq 9) to all the cells and apply the same $U_{SE}$ scaling factor (Eq 10) to all pathways including Schaffer collaterals. All the simulations shown in Fig 5E–5K were run on the cylinder microcircuit with the extracellular calcium concentration set to 2 mM and a duration of 10 s. To disregard the initial ramping activity, we used the last 9 s of the simulation to compute correlation metrics.

## MOOC circuit

We built a CA1 microcircuit as part of an MOOC on edx platform (https://www.edx.org/learn/neuroscience/ecole-polytechnique-federale-de-lausanne-simulating-a-hippocampus-microcircuit). This microcircuit shared many features with the current full-scale model of CA1, but there are a series of differences which need be considered. (1) The model uses a simplified volume as described in [27]. In particular, we defined an hexagonal prism of side 243 μm, surrounded by 6 other equivalent prisms to minimize boundary effects. (2) The network consists of 18K neurons, 20M internal synapses, and 34M afferent synapses. (3) Minor differences could also be observed at the level of the connectome and synaptome due to differences in volume, size, and cell orientations.

## Simulation

**Model instantiation and execution.** For running the hippocampus simulations, the NEURON simulator is used and leverages the CoreNEURON optimized solver for improved efficiency [150]. The model is instantiated using existing scripting methods for NEURON to construct the components of the virtual tissue and add support constructs such as stimuli and reports. Users can use the introspection feature of NEURON to adapt parameters based on certain features and identifiers. Once all components have their final values, the data structures are serialized to disk so that they can be loaded by the CoreNEURON solver into optimized data structures. This results in a 4 to 7× reduction in memory utilization by CoreNEURON, allowing larger simulation on the same hardware. The memory layout can better utilize hardware features such as vectorized instructions and can have a 2 to 7× improvement to execution time.

**Simulation experiments.** For simulations in Section Model simulations, spike time, intracellular membrane potential, and extracellular voltage were recorded and analyzed.

**CA1 spontaneous synaptic release.** The aim of these simulations was to discover whether spontaneous synaptic release in the intrinsic circuit would be sufficient to induce rhythmic

activity in CA1. So for these simulations, the stochastic synaptic release was enabled for each model intrinsic synapse and the SC input disconnected for the entire period simulated, typically 10 s. In vitro CA1 spontaneous synaptic release rate mPSPs are generally estimated from recording postsynaptic events (S22 Table), which is the summation of presynaptic release from the many converging axons. To determine what presynaptic rate parameter value to assign to the synapse models, we ran single neuron simulations for a range of mean presynaptic release rates. We verified that the presynaptic rate range produced a postsynaptic rate range that included ranges reported in literature (S20A Fig). We ran circuit simulations using this presynaptic rate range with both in vitro- and in vivo-like calcium concentrations. All other parameters remained constant across simulations.

**CA1 random synaptic barrage.** The aim of these simulations was to discover whether random synaptic activity of the intrinsic circuit would induce oscillatory activity in CA1. Therefore, in these simulations, SC input was connected while stochastic synaptic release for intrinsic and extrinsic synapses was disabled for the entire period simulated, typically 10 s. SC spike times were generated independently for each input implemented using a Poisson random process with a constant (homogeneous) mean firing rate (Hz). To match the input firing rate of SC input that was able to generate regular theta activity in their CA1 model, we downloaded and analyzed example afferent input data provided with their Fig 5 of [22]. While [22] quoted 0.65 Hz as the critical input rate of synaptic barrage required to generate theta activity, we found their actual spike data had a much lower rate approximately 0.14 Hz (approximately 64k spikes, the summation of their "proximal" and "distal" afferent sources arrived per second from approximately 450k afferents). To cover this range, we ran simulations with constant random spiking rates of 0.05 to 0.60 Hz and repeated these simulations at in vitro- and in vivo-like calcium levels. All other parameters remained constant across simulations.

**CA1 bath concentration of extracellular calcium and potassium ions.** The aim of these simulations was to discover whether tonic depolarization of the intrinsic circuit led to the emergence of oscillatory activity in CA1. Hence, stochastic synaptic release was disabled for each model intrinsic synapse and SC input disconnected for the entire period simulated, typically 10 s. We modeled the bath effect of calcium as explained previously and the effect of extracellular potassium ion ($K^+$) concentration as an constant current injected into the somatic compartment of each neuron (115% to 140% relative to each neuron's rheobase current) [27]. Simulations for this range of injected currents were repeated for a range of calcium concentrations between in vitro- and in vivo-like levels. All other parameters remained constant across simulations.

**CA3 theta oscillatory input.** The aim of these simulations was to discover whether regular theta activity delivered via SC input to the intrinsic circuit would induce theta activity in CA1. So for these simulations, SC input was connected while stochastic synaptic release for intrinsic and extrinsic synapses was disabled for the entire period simulated, typically 10 s. SC spike times were generated independently for a subset of input axons (typically 15,000) using a Poisson random process with a sinusoidal (inhomogeneous) firing rate (Hz) using the Elephant software package functions [151]. Here, the individual mean rate of firing (cell frequency, 0.1 to 0.4 Hz) defines the offset and amplitude that was sinusoidally modulated over time to represent theta-band (4 to 10 Hz) oscillation (signal frequency). Simulations spanning the range of combinations of cell frequency and signal frequency were repeated at in vitro- and in vivo-like calcium levels. All other parameters remained constant across simulations.

**Medial septum input.** In this set of simulations, we aimed to study how MS input can induce theta oscillations in CA1. All the simulations run on the cylinder microcircuit with the extracellular calcium concentration set to 1 mM (in vivo-like level). All the neurons received a background depolarization, expressed as a percentage of the voltage threshold. Since the

background depolarization is not completely known during theta oscillation, we tested values from 100% (threshold) to 130%. Above 130%, we observed non-physiological behaviors of the cells. The sinusoidal hyperpolarizing current injected in PV+ interneurons can be described by 3 parameters: frequency, mean, and amplitude. To reduce the parameter space to be scanned, we considered only the frequency of 8 Hz, an intermediate value in the theta range (4 to 12 Hz). In addition, we set *mean = -amplitude*. The disinhibition amplitude is unknown and we tested values in a range that produces physiological hyperpolarization in the PV+ interneuron models (0.1 to 0.5 nA). We scanned also different acetycholine concentrations which are biological plausible (0 to 5 μM). The simulations ran for 20 s and disinhibition started at time 10 s.

**Propagation of oscillatory inputs.** To examine whether CA3 gamma oscillations propagated to CA1, we aimed to reproduce the in vitro slice experimental results in [89]. During these CCh-induced gamma oscillations, the time course of CA3 spiking activity did not follow a pure sinusoidal waveform, so we had to create a custom inhomogeneous rate function. We did this by first manually reconstructing the probability of discharge of CA3 pyramidal cells over time and then mapped a smoothed version of these values from phase (radians) to time (seconds) coordinates to create a temporal rate function modulated at the reported gamma frequency of 31 Hz. The spike trains were generated from this custom rate function using the Elephant software package [151]. Since the circuit did not model the topography of SC connections to CA1 neurons, it did not constrain how many active SCs project to the simulated slice. Therefore, we ran multiple simulations varying the number of activated SC axons with overall mean rate matching the reported average firing rate of CA3 PCs [89]. To mimic the same experimental conditions, we simulated a slice circuit with extracellular $Ca^{2+}$ 2 mM and $Mg^{2+}$ 2 mM. The effect of bath application of 10 μM CCh effect on CA1 neurons was modeled using the approach described earlier (see section Cholinergic modulation). These simulations ran for 5 s, during which stochastic synaptic release for intrinsic and extrinsic synapses was disabled, with SC input applied 2 s to 5 s. All other parameters remained constant across simulations.

In a final set of simulations, we studied how the CA1 circuit responds to CA3 oscillatory input covering a broad range of frequencies (0.5 to 200 Hz). We simulated an oscillatory input at CA3, with 4 different signal strengths (0.1, 0.2, 0.4, and 0.8 Hz, as mean firing rate) and spike trains were generated using Elephant [151]. We measured the spiking response of CA1, checking if it was constant regardless of the oscillation frequency of CA3. Then, we computed the correlation between CA3 and CA1 and within CA1 neurons using the metrics explained below in section Correlation, with 10,000 randomly selected cell pairs (CA3-CA1 or CA1-CA1, respectively). All spike train correlation measurements were repeated using standard covariance and cross-correlation functions. All the simulations run on the cylinder microcircuit with the extracellular calcium concentration set to 2 mM and a duration of 6 s. We used 1 s of activity (i.e., from 3 s to 4 s of simulation time) to compute gains and correlation metrics.

## Analysis

Simulations showed an initial transient due to variable initiation and it appeared as an initial high activity of the network. This transient lasted for few ms, but we normally discarded the first 1,000 ms in all the simulation analyses to be sure that the parameters converged into a stable regime.

**LFP analysis.** For all simulations where extracellular voltage was recorded, the LFP analysis was performed in the same way unless otherwise stated. The raw extracellular voltage signal was estimated from multiple locations (channels) in the circuit model to mimic experimental electrode positions of a linear probe (e.g., Fig 2C). Following a standard electrophysiological

processing protocol (e.g., [152]), the raw extracellular signal was divided into 2 components using 6th-order Butterworth filtering implemented in the Elephant package (doi:10.5281/zenodo.1186602; RRID:SCR_003833) [151]: a low-pass filtered signal (<400 Hz cutoff) referred to as the LFP and a high-pass filtered signal (>400 Hz). Here, we analyzed the low-pass filtered LFP signal only. In the case of simulations reproducing the results of [89], LFP was band-pass filtered between 10 and 45 Hz to match their analysis protocol focusing on CA1 low gamma-band oscillations. Typically, the first second of recording was discarded to eliminate onset transient artifacts. The remaining signal was detrended and tested for stationarity before being analyzed further (see section Statistical analysis). For time-frequency analysis, this detrended signal was then downsampled from 2,000 (1,000/0.5 ms) to 400 Hz. The downsampled LFP signal was next filtered into separate frequency bands: delta (1 to 3 Hz), theta (4 to 12 Hz), and gamma (30 to 120 Hz). A multitaper method was used to estimate PSD of the downsampled LFP signal with a frequency resolution of 1.5 Hz ([153], mtspec package: https://pypi.org/project/mtspec/). To estimate the spatiotemporal LFP spectral response, a complex Morlet wavelet transform (CWT) was applied with n_cycles = 7 for 1 to 150 Hz range in steps of 0.25 Hz using the Elephant package function and reported in decibel (dB) units [154]. To identify cross-laminar source-sink relationships, CSD analysis was applied to the theta-band filtered signal across all channels of the virtual electrode using the Elephant package function for KCSD1D method [155].

For simulations that generated regular theta oscillations, spike-LFP phase coupling and theta waveform analyses were performed. To quantify phase preference of neurons during stable periods of theta oscillations (defined as at least three 2 s periods of theta in [77]), the Hilbert analytic transform of single channel theta-band signal (unless otherwise stated in SP) was calculated to estimate the instantaneous phase using the Elephant package function. The instantaneous phase was used to assign a (theta) phase angle to individual spike times for a period over 2 consecutive theta cycles (0 to 720 degree period) with trough of theta cycle set as 0 degrees [77]. For each neuron, the resultant (summed) vector angle for its spike train was used to determine the preferred phase of firing and the vector norm estimated the degree of phase locking (see section Statistical analysis). To quantify waveform asymmetry, we measured the asymmetry index for a low-pass LFP signal filtered in 1 to 80 Hz range [75]. To identify peak and trough locations, we used z-score thresholding, as the Hilbert method yielded too many spurious locations, and from this we computed the duration of rise ($t_{rise}$) and decay ($t_{decay}$) for each oscillation. The asymmetry index per oscillation was calculated from $\log(t_{rise}/t_{decay})$.

**Correlation.** To have a quantitative measurement of correlations between spike trains, we mainly used the spike time tiling coefficient (STTC) and the mean spike-spike correlations, computed for a defined number of cell pairs (normally 10,000, for robustness) as the histogram of intervals between all spike times of 2 different cells. For both STTC and spike-spike correlations we used a bin size of 10 ms, considering 1 s of simulated activity.

To verify the results obtained with the STTC method, we computed with the same input data the correlation using standard covariance and cross-correlation functions. All correlation analyses have been done with the Elephant library [151].

## Statistical analysis

To calculate the slopes in linear fits to the appositions data, we used least-squares solution from the python toolbox (`numpy.linalg.lstsq` function).

For the fitting of experimental data points of cholinergic modulation, we used the nonlinear least squares solution (`scipy.optimize.curve_fit` function). Values are expressed as $R^2$ coefficients.

For LFP analysis, the augmented Dickey–Fuller (ADF) unit root test was used to determine whether a single channel continuous signal was stationary before analyzing its spectral properties (`statsmodels.tsa. stattools.adfuller` function). To determine whether single neurons were phase-locked, the Rayleigh Test of randomness of circular data was used to reject the null hypothesis that phase angles of a spike train were uniformly distributed [156]. To compare with empirical results (e.g., [77]), phase analysis results were described by mean ± angular deviation in degrees.

## Statistical comparison methodology

Our statistical comparisons between the model and experimental data were guided by the experimental data availability. To streamline the process, we performed statistical comparisons at an aggregate level rather than conducting comparisons for each individual cell type or pathway.

There are 2 main reasons to perform statistical comparisons at an aggregate level. Firstly, there may be a lack of data available at a granular level, such as for cell composition validation. Secondly, for certain parameters, such as bouton density, because we used a multi-objective optimization approach to match experimental data at an aggregate level, it is more appropriate to validate at an aggregate level as well. For these comparisons, we typically use Pearson's correlation coefficient (`scipy.stats.pearsonr`). Values are expressed as (Pearson correlation coefficient, *p*-value).

In cases where we needed to compare a large number of features for a metric, such as in morphology cloning validation, we used similarity scores in addition to correlation analysis. However, in other cases, when only a mean and standard deviation value were available for comparison, as in Schaffer collateral comparison, we used a z-test or *t* test. Values are expressed as *p*-value.

Finally, if the experimental data had a very small sample size, such as in the case of population synchrony, we did not perform any statistical and we relied only on qualitative assessments.

## Visualization

**Brayns.** Hippocampus circuit and simulations were visualized using Brayns software and its internally developed web interfaces: WebBrayns and Brayns Circuit Studio (Table 1).

Brayns is a visualization software based on ray tracing techniques. It allows rendering 3D scenes and producing high-quality images and videos. Brayns offers programming interfaces in C++ and Python that make it highly customizable, while its web-based interfaces allow the interactive exploration of the scene and make the platform accessible to a wider community (no programming skills needed). In order to create high-quality, high-resolution visuals, Brayns makes use of different rendering engines, like Intel OSPRay (CPU-based ray tracing library, https://www.ospray.org/) or NVIDIA Optix (GPU-based ray-tracing framework https://developer.nvidia.com/rtx/ray-tracing/optix).

The capability of Brayns to render large-scale models, like the whole hippocampus circuit, is key to supporting scientific visualization needs and providing insight on scientific aspects that otherwise are very complex to analyze (e.g., signal propagation at circuit level).

The morphology collage images were also produced with Brayns. A set of clipping planes were used to determine the locations of the slices. For each plane, a second, parallel plane was placed 100 μm further from the origin to create a slice of 100 μm thickness. Then, for each slice and for each neuron morphological type, a small set of neurons of the chosen morphological type and physically located inside the volume determined by the slice were picked and

rendered with Brayns together with the polygonal meshes that define the outline of CA1 layers. The set of neurons displayed in every slice was chosen in a way to maximize its physical distribution within the slice (S11A Fig).

Some of the morphology collage images were post-processed with media design tools (e.g., Adobe Illustrator, Adobe Photoshop) to match the visual style of other figures.

**NeuroMorphoVis.**   NeuroMorphoVis (NMV, see Table 1) was used to visualize single morphologies [157]. NMV is a Blender plug-in that allows the visualization and analysis of neuronal morphology skeletons that are digitally reconstructed. NMV presents many features, including the manual repair of broken morphology skeletons and the creation of accurate meshes that represent the membranes of the morphologies.

## Model and code availability

The entire model, its components, and the source data can be explored and downloaded from hippocampushub.eu. The entire circuit can be also downloaded from Harvard Dataverse (doi:10.7910/DVN/TN3DUI). Code and input data to reproduce both main and supplementary figures are also available from Harvard Dataverse (doi:10.7910/DVN/UGOQWE).

## List of assumptions

**General.**

- This list is not exhaustive. We describe the assumptions that are specific to this work and the ones that could be revised in subsequent refinements.

- We do not list here the assumptions that are already included in and derive from the adoption of specific other models (e.g., [27]; animal models).

- Certain assumptions involve the exclusion of specific features. For example, gap junctions, rare cell types, glial cells, vasculature, are excluded from the current datasets and model.

- In addition to the preceding point, certain limitations were imposed by both data quality and accessibility.

**Data.**

- Most of the data came from the dorsal CA1. We approximated the entire CA1 using data from dorsal CA1.

- We mixed data from different labs, experimental conditions, and animal models. We assumed that the inter-data sets variability is less than the intra-individual variability.

**Volume.**

- We approximated the layer anatomy by dividing the overall CA1 volume into parallel layers with a fixed ratio between their thicknesses. The ratio was taken from the analyses of slice reconstructions from dorsal CA1 of adult rats. We presumed that any potential error introduced by this approximation was comparatively smaller than the inherent noise present in the original atlas.

**Morphologies.**

- We considered the reconstructed cell morphologies to be fairly complete.

- One PC reconstruction showed an axon with a length of 3,646 μm and an extension of 1,325 μm along the transverse axis. We deemed this axon to be relatively complete within the CA1 and therefore used it as a prototype for all the PCs. We presumed that using other axons would introduce a larger problem into connectivity with relatively little gain in diversity.

- We treated PCs as 1 homogeneous cell type.

- SP_PVBC and SP_CCKBC were classified as 2 separate classes.

- We defined PV+ cells as SP_PVBC, SP_BS, SP_AA, CCK+ cells as SP_CCKBC, SR_SCA, SLM_PPA and SOM+ (somatostatin) cells as SO_OLM, SO_BS, SO_BP, SO_Tri.

- Scaling and cloning compensated for the small sample size [27].

### Electrical types.

- We assumed only 4 e-types: classical accommodating (cAC), bursting accommodating (bAC), and classical non-accommodating (cNAC) for interneurons, and classical accommodating for pyramidal cells (cACpyr).

### Cell composition.

- For a given cell type, we assumed that all the expected cells in CA1 are located in the layers for which we had the corresponding morphological reconstructions.

- Trilaminar cells and radiatum-retrohippocampal cells in SO had the same proportion.

- To compensate for the lack of some inhibitory types, we increased the interneurons to match the expected E/I ratio of 11:89 [5]. In doing so, we assumed that matching E/I ratio was more important that maintaining the expected number of cell types present in the model.

### Cell positioning.

- The cell somas could have been located in any point of the layers (random placement).

- Cells had a principal axis that was parallel to the radial axis and perpendicular to the layers.

- Lacking precise evidence, we allowed random rotations around the y-axis for the interneurons.

### Connectome.

- Axo-axonic cells contacted only pyramidal cells.

- We did not allow SCA, Ivy, and BS to form synapses on PC somas to prevent the number of somatic synapses from far exceeding expected values. In doing so, we assumed that the exclusion of few synapses would result in a smaller error compared to the one generated by a large number of somatic synapses.

### Single-cell models.

- The considered electrical features were the ones that described most of the behavior of the cells [35].

- We considered channels to be uniformly distributed in all dendritic compartments except KA and Ih.

- We applied the same currents of pyramidal cells to interneurons with exceptions supported by experimental evidence.

### Synapse model.

- Synapses between A and B had the same parameters.

- Synapses between m-types A and B had parameters extracted from the same distributions (truncated Gaussian).

- Generalization: the synapse type and dynamics were defined, first of all, by pre- and postsynaptic m-type.

- The NMDA/AMPA ratio was kept constant through all compartments of a given neuron.

### Network.

- The action potential was propagated stereotypically from AIS to synapse with a fixed velocity of 300 mm/s [27].

- In slices, neurons preserved their integrity. We did not model cut neurons.

### Schaffer collaterals (SC).

- SC synapses were uniformly distributed along the transverse and longitudinal axes.

- SC synapses could have been placed at all locations on the target neuron (except on the soma), including apical tuft dendrites.

- While [54] measured EPSC only on basket cells, then divided into cannabinoid receptor type 1 negative (CB1R-) and positive (CB1R+), we applied the experimental measurements to all interneurons, dividing them into 2 categories according to their positivity/negativity to CB1 markers, as reported in https://hippocampome.org/php/markers.php.

- STP parameters (i.e., $U$, $D$, $F$, in the Tsodyks–Markram model [23]) were uniform for all interneurons, and they followed the values identified by [53].

### Acetylcholine (ACh).

- The dose-response curves for cholinergic modulation was homogeneous for all cell and synapse types.

- The dose-response was well described using the Hill equation.

- ACh, CCh, and muscarine had the same effects on neuronal excitability and synaptic transmission.

- The change in membrane excitability caused by ACh was assumed to be equivalent to a tonic current injection at the soma.

- Perisynaptic effects of volumetric transmission (i.e., non-synaptic release) of ACh was neglected.

### Software used

Table 1 lists the software used in the paper.

## Supporting information

**S1 Appendix. Example of model improvement: steps to add a new cell type to the model.**
(PDF)

**S1 Fig. Workflow for creating new single-cell models.** Black rectangles represent the different building blocks, and the blue labels are the processes between blocks.
(PDF)

**S2 Fig. Workflow for adding new single-cell models to the network model.** Black rectangles represent the different building blocks, the green boxes are the configuration files, and the blue labels are the processes between blocks.
(PDF)

**S3 Fig. Scientific publications on hippocampus progressively increase.** Counts per year of the number of publications based on a Pubmed search for "hippocampus—cornu Ammonis—CA1—CA2—CA3" from 1900-present indicates the large size of the neuroscientific community researching hippocampus.
(PDF)

**S4 Fig. Circuit building workflow.** Simplified workflow of the circuit building. Boxes represent the different building blocks, while blue labels are the operations between blocks.
(PDF)

**S5 Fig. Validation of cloning.** Validations of cloning methods with a similarity metric (differences between median values divided by variance). (A) The mean scores for 21 unique morphometrics averaged across m-types shown for basal and apical dendrites, and axons in each row. (B) The same scoring grouped by m-types instead of metrics for basal and apical dendrites. Since inhibitory neurons did not have apical dendrites, we show its score only for pyramidal cells. (C) Similar to B, but calculated for axons. Excitatory and inhibitory axons are grouped separately.
(PDF)

**S6 Fig. Increasing morphological diversity.** (A) Persistence diagrams of original (blue), repaired (green), and cloned (red) dendrites. Persistence diagrams encode the start (y-axis) and end (x-axis) radial distances of all dendritic trees in the respective populations of neurons. (B) Persistence diagrams of normalized radial distances (to one) from original (blue), repaired (green), and cloned (red) dendrites. (C) Persistence images are the Gaussian averages of the respective persistence diagrams (B) from original (top), repaired (middle) and cloned (bottom) dendrites. Calculations are computed for the all cell groups. For more detailed analyses, see S7 and S8 Figs.
(PDF)

**S7 Fig. Persistent images for the basal and apical dendrites of each m-type.** (A) Persistent images averaged across each m-type for original morphologies and for their cloned counterparts with respect to the basal dendrites, and (B) their differences. (C) The same process applied for apical dendrites which was only present in the excitatory cell group (e.g., pyramidal cells). Note that the radial distance is synonymous with Euclidean distance and should not be mixed with hippocampal radial axis.
(PDF)

**S8 Fig. Persistent images for the axons of each m-type.** Persistent images averaged across each cell type for (A) original axons, (B) cloned versions, and (C) their difference.
(PDF)

**S9 Fig. Validation of single-cell models.** (A) Graphical illustration of the back-propagating action potential protocol. In a pyramidal cell model, the soma is stimulated to elicit an AP, which is then measured at different distances from the soma. (B) In silico measurements (black dots) are reported and compared with experimental data from Golding and colleagues (red dots and whiskers, indicating mean and standard deviation). (C) Graphical illustration of the postsynaptic potential attenuation protocol. Dendrites on the apical trunk of a pyramidal cell are stimulated with bi-exponential currents; then, PSP's amplitudes are measured as it travels toward the soma. (D) In silico measurements of PSP attenuation (black dots) are expressed as a ratio between the PSP measured in the dendrite and the one measured in the soma. Pyramidal cell models are compared with experimental data from Magee and Cook. The 2 distributions have been fitted with exponential equations, resulting in the following space constants: $\tau$ model = 155.6 μm and $\tau$ experiment = 235.2 μm.
(PDF)

**S10 Fig. CA1 atlas overview.** (A) Original atlas. (B) Smoothed atlas. (C) Upper and lower shells. (D) Centerline through the volume. (E) Planes normal to the centerline. (F) Longitudinal, transverse, and radial coordinates. (G) Orientation vectors. (H) Layer assignment.
(PDF)

**S11 Fig. Cell placement.** A pyramidal cell (PC) is used to illustrate the cell placement in CA1 volume. (A) We set a density profile for each cell type. In the case of PC, we set an uniform density in stratum pyramidal (SP). (B) We randomly identify soma positions matching the given cell density. (C) For each soma position, we assign a morphology and orient it using the vector fields. For PC, we align the main axis of the dendrites with a vector (red arrow) parallel to the radial axis, while we align the axon to a vector (green arrow) parallel to the transverse axis. Furthermore, the more complex branch of the axon points to the Subiculum. (D) We select a morphology that respects a set of rules. In the case of PC, we specify 2 rules regarding the dendrites.
(PDF)

**S12 Fig. Validation of cell placement.** (A) Validation of the cell placement. A subset of cells from each m-type is displayed within each of the 100 slices of 100 μm thickness equally spanned along the longitudinal axis. (B) Cell composition is sampled in different subvolumes (9 cylinders of 300 μm of radius equally spanned along the longitudinal axis) and compared with desired composition from Bezaire and Soltesz (R = 0.999998, $p < 0.0001$). (C) Total cell density in CA1, in the layers SLM + SR, SP, pyramidal cell density in SP, total cell density in SO. Neuron density validation is intrinsic (PC in SP) and extrinsic (the rest). The density is sampled in different subvolumes (9 cylinders of 300 μm of radius equally spanned along the longitudinal axis). Experimental values can be found in S5 Table.
(PDF)

**S13 Fig. Prediction of the number of synapses per connection from the number of oppositions per connection.** We compare the available data on the number of synapses per connection of a given pathway to the corresponding number of appositions per connection. Data can be grouped in 2 sets and fit separately (purple line $y = 0.1096x$ for I-I, red line $y = 1.1690x$ for the rest). The fitting lines can be used to predict how much the appositions should be pruned

to match or predict synapses per connection. Experimental values can be found in S9 Table. E: excitatory neuron, I: inhibitory neuron.
(PDF)

**S14 Fig. Bouton density and synapses per connection.** (A) Mean and STD of bouton density values per individual m-type. (B) Mean synapses per connection for each m-type pair. Experimental values are given in square brackets. (C) Comparisons of mean and STD of synapses per connection for experimental data points and corresponding values in the model. Experimental values in panels A and C can be found respectively in S7 and S9 Tables.
(PDF)

**S15 Fig. Connection probability.** (A) Connection probability with respect to the intersomatic distance within and between Excitatory (E) and Inhibitory (I) groups. (B) Mean connection probability for each type pair. Non-existing connections are left blank and experimental observations are shown with square brackets. (C) Comparison of experimental and model connection probabilities within E/I group pairs. Only pairs of neurons with a maximum intersomatic distance of 500 μm were considered. Experimental values can be found in S8 Table.
(PDF)

**S16 Fig. Convergence on neurons and neuron groups.** (A) Indegree distribution for neurons in this model. The inset shows the distribution on logarithmic scale. (B) Number of synapses made on an average postsynaptic cell from each afferent m-type group. Colorbar in log-normal scale. (C) Mean and STD of synapses made onto an average neuron for each m-type. (D) Number of synapses made onto each neurite type for excitatory and inhibitory classes (Megias and colleagues).
(PDF)

**S17 Fig. Divergence of neurons and neuron groups.** (A) Outdegree distribution for neurons in this model. The inset shows the distribution on logarithmic scale. (B) Number of synapses made by an average neuron from presynaptic m-type to postsynaptic group. (C) Comparison of divergence per m-type with the available experimental values. (D) Prediction of synapse divergence broken down into efferent synaptic group. SO_Tri divergence differs significantly from the experimental data. This could be explained by a morphological reconstruction which is not representative of the entire class or other factors that favor the connections between SO_Tri and interneurons beyond chance. (E) Percentage of synapses made onto individual layers or outside the CA1 mesh. For SR_SCA, the discrepancy between model and experiment can be explain by different morphological subtypes used in the 2 cases. In the experiment of Pawelzik and colleagues, the morphology is well confined in SR, while in the morphology used in the model also invades SP and SO. For D and E, hatched bars indicate the experimental values. Experimental values in panels C–E can be found respectively in S11 and S12 Tables.
(PDF)

**S18 Fig. Prediction and validation of synapses.** (A) Example of in silico paired recording experiment between a PC (purple morphology) and a PVBC (light blue morphology), synapses between PC axon and PVBC dendrites in SO are depicted as yellow spheres. The presynaptic neuron is stimulated with a step current to elicit an action potential. Somatic recordings of EPSP and EPSC are depicted on the right. Each gray line represents one of the 35 trials, while the average response is shown in light blue. (B) Prediction of the PSP amplitudes (B) and PSC CVs (C) for the 130 possible pathways. (C) The combinations with bold borders indicate pathways that have been validated (panels C and D). Validation of PSP amplitudes (C) and PSC CVs (D). Dots represent mean values and whiskers the standard deviation of experimental

(horizontal, in red) and model (vertical, in black) data. The dashed gray line indicates the diagonal (i.e., the target). Experimental values can be found in Ecker and colleagues.
(PDF)

**S19 Fig. Schaffer collaterals anatomy and physiology.** (A) The distribution of the number of synapses per connection (1.0 ± 0.2 synapses/connection), the y-axis has a logarithmic scale. (B) The distribution of efferent synapses made by a single SC (34,135 ± 185 synapses). (C) Scheme of the workflow used for the fitting of SC → PC and SC → INT synapses. On the right, targets that are used for the 2 steps for the fitting of SC → PC and SC → INT. (D) Fitting results of SC → PC synapses. Each plot reports the distribution of one PSP feature (rise time, tau decay, and half-width in D1, D2, and D3, respectively) computed over the 10,000 pairs of pre and postsynaptic neurons. On top, experimental (in red) and model (in black) mean and standard deviation values are reported with a dot and a bar, respectively. (E) Fitting results of SC → INT synapses. E1 shows the average PC EPSPs in control conditions (black line) and when gabazine is applied (no GABA, gray line). The difference between the 2 is the IPSP induced by the feedforward inhibition (blue line). The inset shows the EPSP-IPSP latency, which is the difference between the onset of the IPSP and of the EPSP. E2. The EPSP-IPSP latency distribution of the 1,000 randomly selected PCs. Experimental values for panels D and E2 can be found respectively in S17 and S18 Tables.
(PDF)

**S20 Fig. Spontaneous synaptic release alone did not generate sustained theta oscillations in the CA1 model.** (A) Relationship between spontaneous presynaptic release and postsynaptic events rates for pyramidal cells EPSPs (left) and IPSPs (right). (B) Relationship between calcium level and spontaneous presynaptic release rate for stratum pyramidale LFP responses peak frequency (left) and theta band power (right) shows weak theta power across all simulation experiments. (C–H) Example: 0.001 Hz presynaptic spontaneous release rate (cylinder circuit). (C–E) 2 mM calcium. (C) LFP and theta-band filtered LFP extracellular recordings from stratum pyramidale show irregular activity. (D) PSD shows multiple noisy peaks 1–20 Hz with the highest peak just below theta range. (E) Morlet complex wavelet spectrogram shows intermittent episodes of theta-band activity but these were associated with wide-range frequency response. (F–H) 1 mM calcium. (F) LFP and theta-band filtered LFP extracellular recordings show irregular activity but much smaller amplitude than for 2 mM calcium. (G) PSD shows multiple noisy peaks across a wider frequency range several orders of magnitude less than for 2 mM although the highest peak is just within theta-band. (H) Wavelet spectrogram shows a slightly more sustained period of theta and delta-band (1–3 Hz) activity but with more irregular, higher frequency events than 2 mM calcium.
(PDF)

**S21 Fig. Extrinsic random synaptic activity at low levels generated noisy beta but not theta oscillatory activity in the CA1 model.** For each panel: LFP and theta filtered LFP traces (far left), PSD (middle left), wavelet spectrogram (middle right), CSD (far right) (A). Poisson rate 0.05 Hz. PSD shows broad and noisy peak power located in beta frequency band (13–25 Hz) (B). Poisson rate 0.10 Hz. (C). Poisson rate 0.60 Hz. PSD shows little power within theta and beta band.
(PDF)

**S22 Fig. Extrinsic random synaptic activity fails to induce oscillatory spiking response in the CA1 circuit.** (A) Sample histograms of CA3 input spike time input (top) and output post-stimulus response of CA1 pyramidal and interneurons for a range of CA3 Poisson rates. (B) Spiking input–output relationship between CA3 input spike rate and mean output spiking

rates of pyramidal cells (left) and interneuron types (right). (C) Pyramidal cell cross-correlation histograms (CCH) for different CA3 input spike rates both lack evidence of oscillatory response to randomly timed extrinsic afferent EPSPs.
(PDF)

**S23 Fig. Tonic depolarisation induces spiking oscillations at 2 mM but not 1 mM calcium for a narrow percentage of relative rheobase across circuit scales.** (A) At 120% rheobase, for 1 mM calcium sparse and uncoordinated spiking but at oscillatory peaks occur at 2 mM in pyramidal cell cross-correlation histograms (CCH) in all sizes of circuit. (B) For 2 mM calcium, increasing level of tonic depolarization degrades the strength of oscillation for all sizes of circuit suggesting the resonance effect is limited to a narrow range of tonic depolarization.
(PDF)

**S24 Fig. Tonic depolarisation generates theta-band oscillations across circuit scales at 2 mM extracellular calcium concentration.** Example: 2 mM calcium, 120% rheobase depolarization (recording electrode in SP). For each panel: LFP and theta filtered LFP traces (far left), PSD (middle left), Wavelet Spectrogram (middle right), CSD (far right). (A) Cylinder circuit. (B) Slice circuit. (C) Full circuit.
(PDF)

**S25 Fig. LFP theta waves are highly asymmetric for an extrinsic excitatory oscillatory stimulus.** Top panel: 1 s example traces of 1–80 Hz filtered LFP from stratum pyramidal SP (3) electrode with estimated locations of theta peaks and troughs shown. Bottom panel: estimated rise times (trough to peak time) (far left), estimated decay times (peak to trough time) (middle left), calculated theta wave asymmetry index (middle right), and scatter plot of rise and decay times ($n = 71$ total waves). Example: 8 Hz signal frequency, 0.4 Hz cell frequency, 2 mM calcium, full circuit.
(PDF)

**S26 Fig. Population synchrony of pyramidal cells does not match experimental theta trough ("theta-") and fast-spiking interneurons recruitment lower than experimental levels although better for SP_AA than SP_PVBC independent of circuit scale.** Example: 2 mM calcium, 0.4 Hz cell frequency, and 8 Hz modulation frequency.
(PDF)

**S27 Fig. LFP theta waves are nearly symmetric for an extrinsic inhibitory oscillatory stimulus.** Top panel: 1 s example traces of 1–80 Hz filtered LFP from str pyramidal SP(3) electrode with estimated locations of theta peaks and troughs shown. Bottom panel: estimated rise times (trough to peak time) (far left), estimated decay times (peak to trough time) (middle left), calculated theta wave asymmetry index (middle right), and scatter plot of rise and decay times ($n = 79$ total waves). Example: 8 Hz signal frequency, 1 μM ACh, depolarisation = 120%, 2 mM calcium, cylinder circuit.
(PDF)

**S28 Fig. Population synchrony of pyramidal cells matches experimental theta trough ("theta-") and fast-spiking interneurons SP_AA and SP_PVBC experimental matches theta peak ("theta+") for increased stimulus amplitudes.** Example: 120% depolarisation, 1 μM ACh.
(PDF)

**S29 Fig. Propagation of gamma oscillation from CA3 to CA1.** Gamma oscillation (31 Hz) robustly propagates to CA1 when there is a sufficient amount of SC input. (A) Shows wavelet

spectrograms of LFPs recorded in simulations where CA1 neurons were driven by oscillating input via different numbers of Schaffer collaterals, but no CCh was applied. (B) Shows the wavelet spectrograms of the same simulations, but with simulating the effect of 10 μM CCh. A wide range of the number of SC inputs is able to induce strong gamma oscillation in CA1 both in the case of CCh application and without CCh, but CCh seems to increase the number of inputs needed for stable gamma oscillation, probably due to its weakening effect on synapses. (PDF)

**S1 Table. List of abbreviations and acronyms.**
(PDF)

**S2 Table. Key feature comparison of realistic large-scale hippocampal network models with multicompartmental HH model neurons.** For reasons of space, this is a non-exhaustive list of features.
(PDF)

**S3 Table. Cell composition, counts, and densities.** Values were obtained combining multiple data sets. See section Cell composition of Methods for details.
(PDF)

**S4 Table. Morpho-electrical composition.** Percentage of electrical types (e-types) for each morphological type (m-type). Last column indicates the number of traces used to estimate the percentages. See section Morpho-electrical compositions of Methods for more details. BS indicates both SP_BS and SO_BS.
(PDF)

**S5 Table. Neuron density validation.**
(PDF)

**S6 Table. Placement "optional" rules.**
(PDF)

**S7 Table. Bouton density.**
(PDF)

**S8 Table. Connection probabilities per m-type pair and other related parameters.**
(PDF)

**S9 Table. Number of synapses per connection.**
(PDF)

**S10 Table. Experimentally available data for divergence of synapses per m-type.**
(PDF)

**S11 Table. Available data on the divergence of synapses for different m-types to excitatory and inhibitory groups within CA1.**
(PDF)

**S12 Table. Laminar distribution.** Data for SO_Tri comes from the Sprague Dawley rat data set (see section Experimental procedures).
(PDF)

**S13 Table. Presynaptic dynamics parameters.**
(PDF)

**S14 Table. Postsynaptic dynamics parameters.**
(PDF)

**S15 Table. Schaffer collaterals anatomy experimental data.** N.: number of animals, n.: number of synapses.
(PDF)

**S16 Table. Schaffer collaterals layer profile.** N.: number of animals.
(PDF)

**S17 Table. Schaffer collaterals physiology experimental data for SC→Exc synapses. UoM: Units of Measurement, R.: region, n.: number of cells.**
(PDF)

**S18 Table. Schaffer collaterals physiology experimental data for SC→Inh synapses. UoM: Units of Measurement, R.: region, n.: number of cells.**
(PDF)

**S19 Table. Curated data set on neuronal excitability changes caused by cholinergic modulation.**
(PDF)

**S20 Table. Curated data set on synaptic transmission changes caused by cholinergic modulation.**
(PDF)

**S21 Table. Network effects of ACh.**
(PDF)

**S22 Table. Summary of estimated rates of spontaneous synaptic release in rat CA1.** TTX used to inactivate fast sodium channels in all experiments except Hajos and Mody. D: Days, M: Months.
(PDF)

**S23 Table. Phase tuning of rat CA1 m-types.**
(PDF)

**S24 Table. Long-term discharge rates of rat CA1 neurons in vivo during theta periods.**
(PDF)

**S25 Table. Resources and simulators used in validating the current CA1 model.** cav: calcium-voltage scan. osc control: oscillatory, ms: medial septum, SC: Schaffer collateral, N: Neuron, CN: CoreNeuron.
(PDF)

## Acknowledgments

The authors would like to thank all the people involved in the rat CA1 hippocampus project over the last years, in particular Attila Gulyás, Luc Guyot, and Arseny Povolotsky. We would also like to thank those who offered valuable advice or data: Giorgio Ascoli, Norbert Hájos, Jesse Jackson, Corette Wierenga, and Sylvain Williams. Experimentalists who generated data for this project or who recorded, dye-filled and/or reconstructed neurons: A.B. Ali, A.P Bannister, R. Begum, N. Botcher, J. Deuchars, K. Eastlake, D. I. Hughes, M. Ilia, J. Kerkhoff, S. Kirchhecker, H. Pawelzik, and H. Trigg. D.C. West designed data collection and analysis software. We thank those who worked on the Explore feature of the Hippocampus hub: Anil

Tuncel, Pavlo Getta, Caitlin Monney, Stefano Antonel, Alexander Dietz, and Liviu Soltuzu. Finally, we are indebted to Karin Holm for copy editing, and publication advice and support.

## Author Contributions

**Conceptualization:** Armando Romani, Michele Migliore, Szabolcs Káli, Henry Markram.

**Data curation:** Armando Romani, Alberto Antonietti, Davide Bella, Julian Budd, Kerem Kurban, Thomas Delemontex, Joanne Falck, Lida Kanari, Anna-Kristin Kaufmann, Sigrun Lange, Huanxiang Lu, Ying Shi, Mohameth François Sy, Audrey Mercer, Alex M. Thomson.

**Formal analysis:** Armando Romani, Alberto Antonietti, Davide Bella, Julian Budd, Elisabetta Giacalone, Kerem Kurban, Sára Sáray, Alexis Arnaudon, Cristina Colangelo, Thomas Delemontex, András Ecker, Lida Kanari.

**Funding acquisition:** Tamás F. Freund, Audrey Mercer, Alex M. Thomson, Michele Migliore, Szabolcs Káli, Henry Markram.

**Investigation:** Armando Romani, Alberto Antonietti, Davide Bella, Julian Budd, Elisabetta Giacalone, Kerem Kurban, Sára Sáray, Carmen Alina Lupascu, Rosanna Migliore, Michael W. Reimann, Werner Van Geit, Liesbeth Vanherpe.

**Methodology:** Armando Romani, Alberto Antonietti, Julian Budd, Henry Markram.

**Resources:** Henry Markram.

**Software:** Armando Romani, Alberto Antonietti, Julian Budd, Kerem Kurban, Thomas Delemontex, Michael Gevaert, Juan B. Hernando, Joni Herttuainen, Genrich Ivaska, James Gonzalo King, Pramod Kumbhar, Huanxiang Lu, Juan Luis Riquelme, Vishal Sood, Werner Van Geit, Liesbeth Vanherpe.

**Supervision:** Armando Romani, Jean-Denis Courcol, James Gonzalo King, Judit Planas, Srikanth Ramaswamy, Michael W. Reimann, Mohameth François Sy, Werner Van Geit, Tamás F. Freund, Audrey Mercer, Eilif Muller, Felix Schürmann, Alex M. Thomson, Michele Migliore, Szabolcs Káli, Henry Markram.

**Validation:** Armando Romani, Alberto Antonietti, Davide Bella, Julian Budd, Elisabetta Giacalone, Kerem Kurban, Sára Sáray, Cristina Colangelo, András Ecker, Lida Kanari, Pranav Rai.

**Visualization:** Armando Romani, Alberto Antonietti, Davide Bella, Julian Budd, Kerem Kurban, Marwan Abdellah, Elvis Boci, Cyrille Favreau, Juan B. Hernando, Fabien Petitjean, Judit Planas, Nadir Román Guerrero, Henry Markram.

**Writing – original draft:** Armando Romani, Alberto Antonietti, Davide Bella, Julian Budd, Elisabetta Giacalone, Kerem Kurban, Sára Sáray, Cristina Colangelo, Lida Kanari, Nadir Román Guerrero, Henry Markram.

**Writing – review & editing:** Armando Romani, Alberto Antonietti, Davide Bella, Julian Budd, András Ecker, Sigrun Lange, Audrey Mercer, Alex M. Thomson, Michele Migliore, Szabolcs Káli, Henry Markram.

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
