## [Editor Report · Decision Letter 0]

23 Jan 2024

Dear Dr Romani, 

Thank you for submitting your manuscript entitled "Community-based Reconstruction and Simulation of a Full-scale Model of Region CA1 of Rat Hippocampus" for consideration as a Methods and Resources by PLOS Biology.

Your manuscript has now been evaluated by the PLOS Biology editorial staff and I am writing to let you know that we would like to send your submission out for external peer review.

Once your full submission is complete, your paper will undergo a series of checks in preparation for peer review. After your manuscript has passed the checks it will be sent out for review. To provide the metadata for your submission, please Login to Editorial Manager (https://www.editorialmanager.com/pbiology) within two working days, i.e. by Jan 25 2024 11:59PM.

Kind regards,

Christian

Christian Schnell, PhD

Senior Editor

PLOS Biology

cschnell@plos.org

---

## [Decision Letter · Decision Letter 1]

12 Mar 2024

Dear Dr Romani,

Thank you for your patience while your manuscript "Community-based Reconstruction and Simulation of a Full-scale Model of Region CA1 of Rat Hippocampus" was peer-reviewed at PLOS Biology. It has now been evaluated by the PLOS Biology editors, an Academic Editor with relevant expertise, and by several independent reviewers. 

In light of the reviews, which you will find at the end of this email, we would like to invite you to revise the work to thoroughly address the reviewers' reports.

As you will see below, the reviewers agree that the presented model is comprehensive and technically well done. However, they seem less sure about whether the model covers all the cases mentioned in your manuscript and they also mention several limitations. While we would not expect you to incorporate more cell types or data on the theta oscillations, we think that the potential use cases and limitations of your model need to more carefully described and discussed. If you have more data available, you can of course include these and expand the model instead. 

Please do not hesitate to let me know if you have any question about the revision.

Given the extent of revision needed, we cannot make a decision about publication until we have seen the revised manuscript and your response to the reviewers' comments. Your revised manuscript is likely to be sent for further evaluation by all or a subset of the reviewers.

**IMPORTANT - SUBMITTING YOUR REVISION**

*Re-submission Checklist*

*Published Peer Review*

*PLOS Data Policy*

*Blot and Gel Data Policy*

Sincerely,

Christian

Christian Schnell, PhD

Senior Editor

PLOS Biology

cschnell@plos.org

Reviewer #1: This tour de force work of Romani et al. develops a 3D model of the CA1 region of the hippocampus, based on experimental data by the authors and others. This model includes morphologically and biophysically detailed models of CA1 pyramidal cells as well as several types of interneurons and ensures diversity by a carefully curated cloning procedure of the available morphologies. In addition, connections (including synapse number, synapse strength and short-term plasticity properties) between different types of neurons are extensively validated. Schaffer collateral inputs and cholinergic modulation were also included in the model. The authors have written a very well documented and transparent materials and methods section and comment throughout the results section what results the model succeeds or fails to reproduce. Importantly, the authors have developed a web-based resource to share the model and experimental data (hippocampushub.eu) and organize an open course for its usage, to facilitate its uptake by the research community. All in all this is a very important and detailed methodological work. Nevertheless, and although I highly appreciate the difficulty of the task the authors undertook and its challenges, I have major concerns regarding the conceptual advance of the presented model, as well as it general applicability (with respect to its flexibility). Following are my major concerns, followed by minor comments. 

Major: 

1. In several points throughout the text, the authors claim that the main aim of this work is to be used as a tool to address a wide range of research questions (for example, abstract, line 30: "The unique flexibility of the model allows scientists to address a range of scientific questions.", discussion: Lines 507-509: "Importantly, the reference circuit model is, by definition, general: it was not created to reproduce a narrow spectrum of use cases but to be capable of addressing a wide range of research questions." and lines 591-594: "Third, while the traditional approach of constructing circuits for a specific use-case has short-term advantages for demonstrating proof-of-concept, in the long-term a community reference model must be valid across a range of use-cases.").

Yet, it is hard to conceptualise how the model can support the investigation of other CA1 functions, apart from oscillations and rhythmic activity that the authors are using in this study. For example, one of the well-studied CA1 functions is spatial navigation in general, and place field formation in particular, that depends both on EC input (Brun VH, et al., Neuron 2008) and burst firing of pyramidal neurons (Bittner et al., Nature Neuroscience 2015). How can this model be used to study this CA1 function?

2. The above is relevant also to the conceptual advance of this work. Although this model includes several additional features that were not present in previous models of the CA1 region (e.g. of Bezaire et al. 2016, as also described in the discussion section), no additional features are incorporated that would allow for the general usability of the model, going beyond the rhythmic activity (also studied in the previous modeling works). Such advances could be, for example, burst firing of pyramidal neurons; in this work all pyramidal neurons are classic accommodating, whereas it is known that CA1 pyramidal cells support burst firing (e.g., Jarsky et al., The Journal of Comparative Neurology 2008). This is an important issue, as burst firing is an important mechanism exploited for CA1 function (e.g., Bittner et al., Nature Neuroscience 2015). Another such example would be the incorporation of the known dorso-ventral differences along the CA1 region (Fanselow, Dong, Neuron 2011). 

3. Finally, and also relevant with the general or specific applicability of the model of the previous comments, the model does not capture some features of CA1 oscillatory activity (also commented by the authors). Specifically, Goutagny et al. (2009) have experimentally shown that CA1 theta oscillations can arise independent of CA3 inputs in the isolated hippocampus in vitro. This is in contrast with the results of the authors (Section CA1 generation, lines 336-350). Along the same lines, both for MS and CA3 input, phase locking or firing rate considerably deviates from the experimentally reported values for most types of interneurons (Figure 7B, D, Fig. 9B, D). This points that, despite the detailed nature of this model, essential components are missing to reproduce experimental results with respect to theta oscillations. Contrary, the author's model can reliably sustain gamma oscillations, initiated by the SC input (Fig. S27) and Figure 10 is very informative regarding the properties of the simulated CA1 circuit. 

Minor Comments:

Figure S3 B-C: Please use the same color for all the same cell types (e.g. this is not the case for the cells SP_Ivy and SP_OCKBC) to improve the information content of the figure.

Figure S4: Can the authors elaborate why there are many dendrites with the same birth/death distance (diagonal in plots)? Also what is the physiological meaning of the points above the diagonal?

Figure S7: There is a discrepancy between the experimental and modeling data both for the bAP amplitude and the PSP attenuation for CA1 pyramidal neurons, can the authors please comment on the reasons for this? 

Figure S8, Figure Legend: Panel description seems mixed up (e.g. "F. Layer Assignment" should be "H. Layer Assignment".) Same applies in text references of this figure (e.g. line 851, correct ref. is maybe S8F?).

Line 368: "Figure 6A-C, left columns" maybe the authors mean "Figure 6C-E, left columns"

Fig 7B. Leftmost panel: Please add in figure legend what the black arrows and shaded regions show. For all results in figure 7, are they generated under in vivo like conditions (1mM Ca+2) or in vitro like conditions (2mM Ca+)?

611-612: "… solution, young adult rats (Sprague Dawley, 90-180 g) rats…" please delete the second "rats".

Section 4.7/ Table S4: What are the respective percentages for the 154 recordings of the authors (and how many recordings per morphological type)? 

Fig. S13C: What are the different green or blue dots? If they correspond to different m-type pairs, please indicate the respective pairs. This is of importance, as for some data points, the experimental and model connection probability is very different.

S14B, S15B. These panels are identical (repetitions). I would also guess that they should follow the connection probability, described in S13B (if there is no connection, the number of synapses should be zero), and this is not the case. 

Fig. S15D-E: It would be useful to comment on the reasons for the discrepancy with the experimental data for SO_tri (panel D) and for SR_SCA (panel E).

Line 1127, please correct typo: Mg2+ 2.4mM to 1.4mM. 

Line 1246: Referenced panel Figure 4D does not exist. 

Reviewer #2 (Frances Skinner): In this 'Methods and Resources' submission, the authors present a full-scale atlas-based reference model of the rat CA1 hippocampus. This massive undertaking clearly required input and expertise from many people. It clearly presents improvements and extensions over existing methods, specifically in its inclusion of the curved shape of CA1 and cell placements within, while taking advantage of previously developed methods and algorithms (e.g., connectome determination and STP). Indeed, the curvature and electrode placement matter a lot for the specific electrical recordings obtained (as we have learned from the many LFPy related papers from Einevoll's group).

While it is clear that this is solid work, I think that there are various aspects that need to be addressed and/or clarified to make it clear to the community how this reference model could best serve as a resource moving forward.

1) "Community-driven" and "community-based" are used heavily, but it is not clear to me that this is a specific aspect of this work that should be emphasized. That is, many (most?) models use experimental data (as published) from many labs (i.e., community-based). Indeed, the recent publication of Hippocampome.org2.0 (elife) allows leveraging of published experimental data to create Izhikevich type network models. Also, whether this reference model is 'community-driven' is not clear. Indeed, as the authors clearly would know, it has been discussed many times in various contexts (e.g., see Eriksson et al, eLife 2022 and Levenstein et al., J Neurosci 2023 for some viewpoints), that there are many different goal(s) that affect the type and extent of models developed in the community, and approaches that can be used. It seems to me that the community aspect is more about presenting a very detailed CA1 model (reference model) to the community to use as they state. As such, I think it would more helpful and clear for the community if the work was presented in this context, as well as actually specifying questions it could address, rather than just saying that there is a 'wide range of research questions' (Discussion), or a 'range of scientific questions' (Abstract). As presented, it is about not having the need to adjust parameters (line 79) and having a unifying, multiscale model (line 52) etc.? If so, then the model goal is mainly (only?) about reproducing the biology, but not trying to understand it per se? For example, we clearly know that there is degeneracy in biological systems (consider the extensive work in STG system). In essence, it is in the authors' interest to be clear and specific about what they intend with their reference model, thus providing guidance for potential users and audience etc.

2) An advantage of their model is that they can largely reproduce experimental conditions (recording sites, extracellular concentrations etc.), and this may be the main usage of their reference model ('reproduce and reconcile')? However, when they state in the Abstract they 'reproduce and reconcile experimental findings', I am not quite sure which part of the paper they intend this to refer to (SC, theta, etc.). I felt confused in how the work was presented. For example, in reconstructing the SC, they reproduce Sasaki et al - so is considered a validation? Or is this a 'reproduce and reconcile experimental findings' or ?? - later (line 573) they refer to a poor result that was rectified by less sampling(?) to match Sasaki et al. Is it that they heterogeneous cell population (in constructing the model) was too diverse to match experiments? Could the authors explain/expand on this? For Ach, the authors refer to the emergence of three activity regimes and note that consistency is unclear. Is this something their model could perhaps reconcile? However, in general, the organization and flow of the paper was hard to grasp in their presentation of the overall development and using of the model.

3) The theta section (2.21) was confusing in its intent - intended to be 'reproduce and reconcile'? They initially state that we do not know the trigger that generates theta oscillations in CA1 due to conflicting evidence, and go on to examine intrinsic generation and extrinsic via CA3 and MS (of course, they could all contribute in the biological system in different conditions). For intrinsic, the experimental work as cited is from Goutagny et al, but then they go on to compare with Bezaire et al (model work), and conclude that they could not generate theta in their reference model. Rather, it IS the case that theta occurs intrinsically in the biological system), so because their reference model could not show this, indicates that there is something lacking in the reference model it would seem? In the CA3 input part they say 'contrary to Goutagny…', and talk about circuit scales, and justify only using the cylinder circuit. There is a confusing flow/rationale/justification of what the authors intend in their presentation of the work. At the end of the CA3 part, it is about in vivo and in vitro differences. Perhaps they can present one of their model uses/questions to address as representing in vitro and in vivo differences. Along those lines, is it only extracellular Ca that is changed for these differences? The MS section seems to be about illustrating the theta rhythm via disinhibition can occur in their model? The section showing that delta to low gamma propagated best was interesting - were different Ach levels used? - a potential model usage for experimental interfacing? - are there a specific experiments one would do/point to? Again, in general, the organization/flow/rationale of the work presented is somewhat confusing. I think this all stems back to the first point above.

4) The website and education MOOC is commendable and addressing the above points would be helpful for usage goals. The authors refer to a 'smaller version' (line 120) - it would be helpful to specify what they mean by this - the cylinder or slice circuit? Or something else? If something else, what could one investigate (relative to the full model)?

5) The authors refer to simulating a "scalable and reproducible circuit automatically". This sounds excellent but it would be helpful for potential users to provide some more details/examples for what is automated? Adding another cell type? 

6) To help for usage and moving forward, is the authors' intent to keep adjusting model parameters to reflect biology (e.g., dorsal/ventral aspects, deep/superficial pyramidal cell differences and so on that do not currently exist in the model). Maybe focusing on reproducing an experimental population spike or LFP recording output would be helpful to interface with experiment? Especially given the curvature that is modelled. Given that the existing reference model can (line 318) "simulate slices of a certain thickness, change the extracellular concentration of ions, change temperature, and enable spontaneous synaptic events", maybe one should could design such experiments to go hand in hand with these variations? to help reconcile experimental differences maybe? Again, this all reflects back to the first point.

MINOR

- Bistratified cells seem to be a distinct type (e.g., see Harris et al. Plos Biology 2018 and Gouwens et al. Cell 2020), so not sure why two types of bistratified cells used?

- Table S3 - based on what refs? Estimated from where?

- Line 190 onward: "Predictions" of pathways stuff - what experiments are needed? Please clarify, as they seem to say they match data in other stuff later on in same paragraph.

Could you be specific about discrepancies?

- line 957 - not sentences

- Fig 5 - caption typo ('back')

Reviewer #3 (Daniele Linaro): In this paper, the authors present a large-scale model of the CA1 region that incorporates a significant number of key features of this brain region. The effort is monumental and follows in the footsteps of Markram et al., 2015. I particularly appreciated the collaborative nature of this project, which will allow the community to operate as "beta-testers" of the model and its many components: indeed, as presented here, it is my opinion that it is very difficult for a reviewer to assess whether this model will be able to accurately predict experimental results outside those that have been described here. For this reason, I have no specific comments or recommendations on the manuscript, except to congratulate the authors for their important work.

---

## [Decision Letter · Decision Letter 2]

8 Jul 2024

Dear Armando,

Thank you for your patience while we considered your revised manuscript "Community-based Reconstruction and Simulation of a Full-scale Model of Region CA1 of Rat Hippocampus" for publication as a Methods and Resources at PLOS Biology. This revised version of your manuscript has been evaluated by the PLOS Biology editors, the Academic Editor and the original reviewers.

Based on the reviews, we are likely to accept this manuscript for publication, provided you satisfactorily address the remaining points raised by the reviewers. Please also make sure to address the following data and other policy-related requests:

* We would like to suggest a different title to improve accessibility: "Community-based reconstruction and simulation of a full-scale model of the rat hippocampus CA1 region"

* Please add the links to the funding agencies in the Financial Disclosure statement in the manuscript details.

* Please include the full name of the IACUC/ethics committee that reviewed and approved the animal care and use protocol/permit/project license. Please also include an approval number.

* DATA POLICY:

Regardless of the method selected, please ensure that you provide the individual numerical values that underlie the summary data displayed in the following figure panels as they are essential for readers to assess your analysis and to reproduce it: 3CDEFGH, S3, S7B, S10C, S12A, S14CD, S15C, and S18A

* CODE POLICY

Per journal policy, if you have generated any custom code during the course of this investigation, please make it available without restrictions. Please ensure that the code is sufficiently well documented and reusable, and that your Data Statement in the Editorial Manager submission system accurately describes where your code can be found. As the code that you have generated is important to support the conclusions of your manuscript, its deposition is required for acceptance.

* Please note that per journal policy, we do not allow the mention of "data not shown", "personal communication", "manuscript in preparation" or other references to data that is not publicly available or contained within this manuscript. Please either remove mention of these data or provide figures presenting the results and the data underlying the figure(s).

We expect to receive your revised manuscript within two weeks. 

*Published Peer Review History*

*Press*

Sincerely,

Christian

Christian Schnell, PhD

Senior Editor

cschnell@plos.org

PLOS Biology

Reviewer remarks:

Reviewer #1: I appreciate the additions/clarification made by the authors, and I look forward to seeing an efficient uptake of this model by the research community. I only have one minor comment, that does not affect my recommendation for acceptance.

Fig 7B. This previous minor comment was referring to the panel B (not A). It would be informative to add in the figure legend what the black arrows and shaded regions show (for the plots below "Simulation vs in vivo" subtitle in panel B).

Reviewer #2 (Frances Skinner): accept

---

## [Editor Report · Decision Letter 3]

11 Sep 2024

Dear Armando,

Thank you for your patience while we considered your revised manuscript "Community-based reconstruction and simulation of a full-scale model of the rat hippocampus CA1 region." for publication as a Methods and Resources at PLOS Biology. 

We have now completed the editorial checks and most things have been addressed in the revised manuscript. However, there are still a few points that are still open:

* Please note that per journal policy, we do not allow the mention of "data not shown", "personal communication", "manuscript in preparation" or other references to data that is not publicly available or contained within this manuscript. Please either remove mention of these data or provide figures presenting the results and the data underlying the figure(s). There is currently one instance of "not shown" in the figure legend of Figure S19.

* We could not access the repository that contains the code and data. Could you please make this available, so we can check it before publication of the paper?

* We did also ask for source data for the summarizing figures in your manuscript (for example for the panels with box plots). The source data file should contain not only the average/mean and SEM or SD, but the individual data points. Here is the a paper where this has been done: https://journals.plos.org/plosbiology/article?id=10.1371/journal.pbio.3002660

I understand that some of the data you show are obtained from previous publications where these data are not available or from modelling, where it is not feasible to provide the individual data points. For the figures with the first case, could you please mention this in the figure legend? For the figures with the second case, please refer to the code in the repository that was used to generate this figure, so our readers can reproduce the findings and re-use the data. 

I am pasting the full descriptions here again, but this refers to figures 3CDEFGH, S3, S7B, S10C, S12A, S14CD, S15C, and S18A

* DATA POLICY:

Regardless of the method selected, please ensure that you provide the individual numerical values that underlie the summary data displayed in the following figure panels as they are essential for readers to assess your analysis and to reproduce it: 3CDEFGH, S3, S7B, S10C, S12A, S14CD, S15C, and S18A

We expect to receive your revised manuscript within two weeks. 

*Published Peer Review History*

*Press*

Sincerely,

Christian

Christian Schnell, PhD

Senior Editor

cschnell@plos.org

PLOS Biology

---

## [Editor Report · Decision Letter 4]

24 Sep 2024

Dear Armando,

Thank you for the submission of your revised Methods and Resources "Community-based reconstruction and simulation of a full-scale model of the rat hippocampus CA1 region" for publication in PLOS Biology. On behalf of my colleagues and the Academic Editor, Jozsef Csicsvari, I am pleased to say that we can in principle accept your manuscript for publication, provided you address any remaining formatting and reporting issues. These will be detailed in an email you should receive within 2-3 business days from our colleagues in the journal operations team; no action is required from you until then. Please note that we will not be able to formally accept your manuscript and schedule it for publication until you have completed any requested changes.

PRESS

Sincerely, 

Christian

Christian Schnell, PhD

Senior Editor

PLOS Biology

cschnell@plos.org